Manuscript prepared for Clim. Past
with version 2015/09/17 7.94 Copernicus papers of the LaTeX class copernicus.cls.
Date: 4 July 2016

# Mid-to-late Holocene Temperature Evolution and Atmospheric Dynamics over Europe in Regional Model Simulations

Emmanuele Russo[1] and Ulrich Cubasch[1]

Institute of Meteorology - FU-Berlin Karl-Heinrich-Becker-Weg 6-10, 12165 Berlin (DE)

*Correspondence to:* Emmanuele Russo (emmanuele.russo@met.fu-berlin.de)

**Abstract.** The improvement in resolution of climate models has always been mentioned as one of the most important factors when investigating past climatic conditions, especially in order to evaluate and compare the results against proxy data. Despite this, only a few studies have tried to directly estimate the possible advantages of highly resolved simulations for the study of past climate change.

Motivated by such considerations, in this paper we present a set of high-resolution simulations for different time slices of the mid-to-late Holocene performed over Europe using the state-of-the-art Regional Climate Model COSMO-CLM.

After proposing and testing a model configuration suitable for paleoclimate applications, the aforementioned mid-to-late Holocene simulations are compared against a new pollen-based climate

reconstruction dataset, covering almost all of Europe, with two main objectives: testing the advantages of high-resolution simulations for paleoclimatic applications, and investigating the response of temperature to variations in the seasonal cycle of insolation during the mid-to-late Holocene, with the aim of giving physically plausible interpretations of the mismatches between model and reconstructions.

Focusing our analysis on near surface temperature, we can demonstrate that concrete advantages arise in the use of highly resolved data for the comparison against proxy-reconstructions and the investigation of past climate change.

Additionally, our results reinforce previous findings showing that summertime temperatures during the mid-to-late Holocene were driven mainly by changes in insolation and that the model is too

sensitive to such changes over Southern Europe, resulting in drier and warmer conditions. However, in winter, the model does not correctly reproduce the same amplitude of changes, even if it captures the main pattern of the pollen dataset over most of the domain for the time periods under investigation. Through the analysis of variations in atmospheric circulation we suggest that, even though the wintertime discrepancies between the two datasets in some areas are most likely due to high pollen

uncertainties, in general the model seems to underestimate the changes in the amplitude of the North Atlantic Oscillation, overestimating the contribution of secondary modes of variability.

# 1 Introduction

Climate has a direct effect on all living organisms and so has always, and always will have an influence on human affairs (Wigley et al., 1981). From antiquity to the present day, human life and civilization have been affected by the availability of natural resources such as water, food, construction materials, etc. Under the current threat of global warming, understanding how climate will change in the next century has become of fundamental importance for the impact it could have on the life of our planet. Useful instruments for the study of climate change and its possible consequences are climate models. In general terms, a climate model can be defined as a mathematical representation of the climate system based on well-established physical principles (Randall et al., 2007).

Many uncertainties still affect climate models, particularly regarding their sensitivity to changes in the external forcings (Collins and Allen, 2002; Yip et al., 2011). To improve our predictions of the future climate it is necessary to better understand such a response: this can be accomplished through the application of climate models for the study of changes in past climatic conditions.

An important case of study is represented by the evolution of European climate during the mid-to-late Holocene (from 6000 years ago to present day). The large number of proxy data available and the particular configuration of the Earth astronomical parameters, make it a useful period for the evaluation of the models' response to changes in insolation (De Noblet et al., 1996; Kutzbach et al., 1996; Masson et al., 1999; Vettoretti et al., 2000; Bonfils et al., 2004; Braconnot et al., 2007a, b; Mauri et al., 2014). During the mid-to-late Holocene, over northern latitudes in general, changes in the total amount of insolation during the year (with respect to present day conditions) were negligible ($\leq 4.5\ W/m^2$) when compared to the seasonal variations (up to more than $30\ W/m^2$ for summer insolation at high latitudes) (Fischer and Jungclaus, 2011). Indeed, relevant variations in the seasonal values of surface variables would be expected. However, evidence shows that reconstructed climatic parameters, such as surface temperature, over Europe, did not always follow directly the astronomical forcings (Cheddadi et al., 1997; Davis et al., 2003; Bonfils et al., 2004; Braconnot et al., 2007a, b; Mauri et al., 2014). Their signals seem to have also been influenced by other complex processes such as atmospheric circulation, geography, or land-surface interactions with the atmosphere.

Different studies have been conducted in order to understand the mechanisms driving the seasonal behaviour of European surface variables during the mid-to-late Holocene. Cheddadi et al. (1997) showed that the results of a pollen-based reconstruction dataset constrained by lake-level data, indicated that summer and winter temperatures were different over Northern and Southern Europe at the mid Holocene in comparison to present-day values: winters, in particular, were warmer over Northern Europe even if the insolation was reduced, while summers were colder over Southern Europe, despite the higher insolation. Similar results were obtained by Davis et al. (2003) who proposed an updated database of European pollen reconstructions for the entire Holocene. Bonfils et al. (2004), within the PMIP (Paleoclimate Model Intercomparison Project (Joussaume and Taylor, 1995)) collaboration, hypothesized that winter atmospheric patterns and summer soil conditions had an impor-

tant influence on seasonal changes of temperature and precipitation. This has also been highlighted
by a study from Starz et al. (2013) who performed a simulation for the mid-Holocene with a cou-
pled soil-ocean-atmosphere circulation model and dynamic vegetation, better reproducing soil water
storage and heat fluxes. They found that changes in the soil's physical properties of the model led
to improved model results and hampered anomalies in surface variables, with respect to proxy-data.
Fischer and Jungclaus (2011) studied the evolution of the European seasonal temperature cycle in a
transient mid-to-late Holocene simulation with an ocean-atmosphere global climate model, although
they were unable to reproduce correctly the reconstructed data over the entire region of study. In par-
ticular, their results presented only a weak shift to a positive phase of the NAO at mid Holocene in
winter, resulting in colder conditions over Northern Europe and warmer over Southern Europe, with
respect to the values of reconstructions. In summer, again, the signal seemed to be mainly driven by
changes in insolation, resulting in generally warmer conditions over the entire domain and period
of study. Conversely, in their recent work, Mauri et al. (2014) suggested that the different response
of surface variables at the mid Holocene was highly related to changes in atmospheric circulation
both in winter and in summer. Specifically, they proposed that in summer a major incidence of the
"Scandinavian High" was most probably the reason for colder temperatures over Southern Europe
6000 years ago. In winter, on the contrary, a more positive phase of the North Atlantic Oscillation
would have been responsible for warmer and wetter conditions over Northern Europe and an oppo-
site behaviour in the South. Although these interpretations are all physically plausible, still general
consensus is still missing on the correct explanation of the response of the climate system to changes
in insolation for this period. Within the mentioned studies, all the climate model applications have
been conducted with transient simulations or considering a single time slice with Global Circulation
Models. In many cases the resolution of these simulations was not high enough to allow for an as-
sessment of the climate behaviour on a regional scale. As suggested by Renssen et al. (2001), if we
want to evaluate the data against climatic reconstructions based on pollen data or any other record,
an improvement in the resolution is required (Bonfils et al., 2004; Masson et al., 1999). Additionally,
higher resolution is expected to lead to an improvement of the results (Fischer and Jungclaus, 2011),
allowing the representation of small-scale processes and more detailed information on surface and
soil features (Feser et al., 2011).

Bearing this in mind, in recent years the application of regional climate models for paleoclimate
studies has become more frequent. For example, Prömmel et al. (2013) used the COSMO-CLM in
order to address the effect of changes in orography and insolation on African precipitation during
the last interglacial. Fallah et al. (2015) investigated precipitations and dry periods during the Little
Ice Age and the Medieval Warm Period over central Asia. Wagner et al. (2012) compared the mid
Holocene and pre-industrial climate over South America, while Felzer and Thompson (2001) evalu-
ated a regional climate model for paleoclimate applications in the Arctic.

In several studies, regional simulations of European climate during different times of the mid-to-late

Holocene have been performed (Gómez-Navarro et al., 2011),(Gómez-Navarro et al., 2012, 2013, 2015; Schimanke et al., 2012; Renssen et al., 2001; Strandberg et al., 2014). Nevertheless, they either focused on a singular time-slice, or covered a more recent period of time, for which changes in insolation due to astronomical forcings were negligible.

In this paper we employ for the first time a regional climate model, the COSMO-CLM (**CCLM**), for the investigation of the main climatic changes that characterized Europe during multiple time-slices of the Mid-to-Late Holocene, with three main objectives:

  – Propose and test a model configuration suitable for paleoclimate studies

  – Investigate the possible added value of highly resolved simulations arising in the comparison
against proxy-reconstructions

  – Analyse proxy and model mismatches, providing plausible physical interpretations of the dynamical processes responsible for them

Our discussion is structured as follows: in section 2 the employed methodology, including a brief description of the models and the proxy datasets, is presented. Results are illustrated and discussed
in section 3: first a validation of the data for present-day conditions is conducted in order to test the performances of the model with the changes necessary for paleoclimate applications; then the mid-to-late Holocene simulations are compared against pollen-based reconstructions, trying, in a first instance, to highlight the advantages of the performance of highly resolved simulations specifically for this case of study; finally, physically plausible interpretation of the mismatches between the
CCLM results and the reconstructions are proposed; the results of other studies are additionally discussed.

## 2    Methods

### 2.1    Experimental Setup

In this work we perform a set of climate simulations, covering several time slices of mid-to-late
Holocene, employing models at different resolution.

The modus operandi consists of three parts and is based on the so-called *time-slice* technique (Cubasch et al., 1995):

  1. First a transient continuous simulation is performed with the coupled atmosphere-ocean circulation model ECHO-G, composed by the ECHAM4 (Roeckner et al., 1996) and the ocean
model HOPE (Wolff et al., 1997), at a spectral resolution of T30 ($\sim 3.75^o \times 3.75^o$). Further information on the simulation realization are provided in Wagner et al. (2007).

  2. We then select seven different time slices, at a temporal distance of approximately 1000 years from each other, from 6000 years ago down to the pre-industrial period, 200 years before

present, in accordance to the time slices for which the pollen reconstructions are available. For every time slice, a simulation is conducted, for a 30-year period, with the atmosphere-only global circulation model ECHAM5 (Roeckner et al., 2003) at a spectral resolution of T106 ($\sim 1.125^o \times 1.125^o$), using prescribed sea ice fraction and sea surface temperatures derived from the ECHO-G continuous run.

3. Finally the ECHAM 5 outputs are further downscaled with the regional climate model COSMO-CLM model version 4.8 clm 19 at a horizontal resolution of 0.44 longitude degrees, using 40 vertical levels. The CCLM model is a non-hydrostatic RCM with rotated geographical coordinates and a terrain following height coordinate (Rockel et al., 2008), developed from the COSMO model by the German weather service (DWD) (Doms and Schättler, 2003).

In a first step we want to test whether the RCM setup and the applied model's code modifications, required for implementing values of GHGs and astronomical forcings, are suitable for paleoclimate studies. In order to set the values of astronomical parameters for the corresponding investigation periods, we apply the routine of Prömmel et al. (2013) that allows the estimation of latitudinal and seasonal insolation at the top of the atmosphere based on Earth's astronomical parameters calculated by Berger (1978). In Fig.1 the anomalies of zonal mean insolation on top of the atmosphere (**TOA**) between the pre-industrial period **PI** and 6000 years BP are presented. Additionally, the winter and summer mid-to-late Holocene evolution of TOA insolation for 60 and 30 latitudes North are also shown in the same figure (*Right*). Additional changes to the original model code are required in order to set the values of equivalent $CO_2$ concentration, representing variations in $CH_4$, $CO_2$ and $N_2O$. These data are deduced from air trapped in ice cores (Flückiger et al. (2002)). The contribution of the mid-to-late Holocene changes in GHGs concentration to the radiative balance is negligible (less than 2W/$m^2$) in comparison to the effects of changes in insolation, and only the latter are considered in our discussion.

The setup of the COSMO-CLM is based upon the work of Hollweg et al. (2008) within the Euro-CORDEX Downscaling experiment (Jacob et al., 2014). A more detailed description of the model configuration used is provided in Table 1. For this study the model has been employed coupled to a Soil Vegetation Atmosphere Transfer scheme, the **TERRA ML**, a multi-layer model with a constant temperature lower boundary condition that allows to reproduce the fluxes of heat, water and momentum between the soil-surface and the atmosphere. Recent data of the physical parameters of the Earth's surface (e.g., orography, land use, vegetation fraction, and land-sea mask) are employed for the simulations. The model domain, shown in Fig.2, is the one used for the Euro-CORDEX simulations (Jacob et al., 2014), extending from Southern Greenland to Western Russia in the North and from the Western Atlantic coast of Morocco to the Red Sea in the South. Each simulation includes a 5-year spinup period used to let the model reach a semi-equilibrium state as suggested by Hollweg et al. (2008) .

## 2.2 Observations

For the model validation for present climate, the **E-OBS** (Haylock et al., 2008) and the Climate Research Unit (**CRU**) (Harris et al., 2014) observational datasets are used as benchmarks for the comparison with the results of a COSMO-CLM control run covering the period 1991-2000 and driven by the ERAInterim (**ERAInt**) dataset (Dee et al., 2011). The validation is conducted with respect to the total precipitation and 2 meter temperature winter and summer seasonal means. Additionally, CCLM heat fluxes and evapotranspiration values, from the same simulation, are validated against the **GLDAS** (Global Land Data Assimilation System Version 1 Products) dataset.

## 2.3 Proxy-Reconstructions

Subsequently, the results of the mid-to-late Holocene simulations are compared against the dataset of Mauri et al. (2015). This is the latest updated pollen-based climate reconstruction dataset for Europe and constitutes an upgrade of the results of Davis et al. (2003). It is derived with the same methodology, but with a wider number of fossil and surface-samples, following a more rigorous quality control. The data cover a time slice every millennium for the entire Holocene and are derived through a 4-dimensional spline-interpolation in time and space. They are deduced with an analogue transform method and corrected with postglacial isostatic readjustment. Along with the data, a standard error estimate derived from the transform and the interpolation methods is also provided. Reconstructions contain information on seasonal (winter and summer) and annual values of precipitation and temperature, as well as a measure of moisture balance and of growing degree days over 5 degrees, and are provided on a regular grid with a resolution of $1 \times 1$ longitude degrees.

The choice of the dataset of Mauri et al. (2015) has been done for several reasons. First of all, it allows us to perform a comparison against the model results over most of the simulation domain, considering different variables (even if we only focus on temperature in our discussion). Then, it covers exactly the same time slices of our model simulations: no other dataset has this temporal and spatial coverage at such high spatial resolution. Additionally, the robustness of the data has been thoroughly tested, in Mauri et al. (2015), against other proxies (including chironomids, $\delta 18O$ from speleothems and lake ostracods, bog-oaks, glacio-lacustrine sediments, wood anatomy and other pollen reconstructions based on different reconstruction methods) leading to satisfactory results. Nonetheless, similar pollen-based climatic reconstructions have been extensively employed in other data-model comparisons, and, most recently, for the evaluation of the PMIP3/CMIP5 climate models included in the last IPCC report ((Stocker et al., 2013; Harrison et al., 2015)).

## 3    Results and Discussion

### 3.1    Model Validation and Evaluation for Present Day

As a first step a control simulation has been performed with present values of orbital parameters and greenhouse gases (sec.2), in order to test the ability of the CCLM, modified accordingly to our purposes, to properly reproduce present-day climate. Additionally, this provides further knowledge about the spatial distribution of the model performances.

The simulation covers a 10-year period, between 1991 and 2000. Even if the length of this simulation can be considered as "critical" for the model's validation, we want to acknowledge that, due to computational reasons, it was not possible to cover a longer period.

In Fig.3 and Fig.4, winter and summer seasonal means of temperature (left panel) and precipitation (right panel) from the CCLM simulations are compared against the CRU and the E-Obs observational datasets. In the first column of each panel, the climatology of the different datasets is shown: the model is able to correctly reproduce, within a certain degree of accuracy, the climatology of the observations for both temperature and precipitation in winter and in summer.

In the right column of every panel, Temperature and Precipitation values from the present-day control run are directly validated, through a Student's T-test, against the CRU and the E-Obs datasets. The same test is conducted for evaporation and heat fluxes but against the GLDAS dataset in Fig.5. In these figures the black dots represent the grid cells where the null hypothesis of the T-test, assuming that the data being sampled could be drawn from the same underlying distribution, is not rejected at a significance level of $5\%$. The biases between the CCLM results and the observations are represented with different colours. The results show that, for temperature, the model performs well over Northern Europe in both winter and summer. Winter-time results are in particularly good agreement with observations over Northeastern Europe and Scandinavia (Fig.3II). However, larger deviations (up to $4^oC$ in some cases) are present over Central Europe, Turkey and Northern Africa. In particular the model tends to simulate generally colder conditions over these regions. Winter precipitation results seem to be in good agreement over a major part of the domain, with some deviations from the observations over regions with particularly complex orography, in regions that are normally highly affected by westerlies and in the Northern African coasts of the Mediterranean Sea (where the biases are particularly pronounced, and the model results diverge by almost $100\%$ from the values of the observations) (Fig.3IV). In summer, instead, the main discrepancies are found over Southern Europe both for temperature and precipitation (Fig.4). In particular the temperature anomalies reach $4^oC$ over most of the mentioned area. It has been shown in previous works (Hagemann et al., 2004; Christensen et al., 2008; Kotlarski et al., 2014; Jerez et al., 2010, 2012) that, in general, regional climate models poorly simulate southern European summer conditions. This seems to be most likely related to deficiencies in soil-atmosphere coupling (Seneviratne et al., 2006; Fischer et al., 2007; Seneviratne et al., 2010). In soil moisture-controlled evaporative regimes, such as the Mediterranean

basin, low soil moisture contents (due probably to an underestimation of spring-time precipitation or badly represented soil properties in consequence of complex orography) limit the amount of energy transferred by the latent heat flux. This increases the sensible heat flux, ultimately leading to an increase of air temperature, on the one-hand, and to a decrease of local precipitation on the other (Zveryeav and Allan, 2010).

Based on these considerations, we suggest that the model reproduces anomalously warm and dry conditions over a wide part of Southern Europe and the Mediterranean basin, during summer, as a consequence of a wrong conversion of energy towards latent heat in these regions. This hypothesis is supported by the heat fluxes and evapotranspiration maps (Fig5) presenting a spatial distribution of the anomalies resembling the ones of temperatures and precipitation. In particular, the model underestimates latent heat flux and evapotranspiration, while overestimating sensible heat over corresponding area.

Nevertheless the performances of the model with the applied changes are in good agreement with the results of other works focusing on the same region ((Hollweg et al., 2008; Kotlarski et al., 2014; Schimanke et al., 2012; Gómez-Navarro et al., 2011, 2013), having in general the same features and spread of the anomalies. Indeed the applied changes and configuration appear to be exploitable for paleoclimate applications.

### 3.2  Possible added Value of Highly Resolved Simulations for Paleoclimate Studies

In a successive step, we conduct a comparison of the three models at different resolution in order to estimate possible advantages in the use of highly resolved simulations for paleoclimate studies. According to Solomon et al. (2007): "Paleoclimate data are key to evaluating the ability of climate models to simulate realistic climate change". In particular, since the details added by high resolution models can help in the interpretation of proxy data that are often influenced by processes taking place on smaller scales than the ones resolved in coarser models, they are supposed to be a particularly suitable tool for paleoclimate studies. Within this context, in our discussion we try to highlight the importance of using high resolution models, and in particular Regional Climate Models, for the simulation of past climate change.

Aiming at investigating the value added by highly resolved simulations for the comparison of changes in near surface temperatures against proxy-reconstructions, we follow a two steps approach:

1. Firstly, we conduct a qualitative analysis of the simulations performed with three models at different resolution in order to detect visible differences in the reproduced signals.

2. Secondly, we employ a quantitative approach in order to estimate the skills of the RCM, in comparison to the driving GCM, in reproducing the same mid-to-late Holocene changes in temperature as derived from proxy-reconstructions.

As a benchmark for such comparison we use the pollen-based temperature reconstructions of Mauri et al. (2015). In this way, we aim at establishing whether the representation of smaller scale processes and improved orographic features of the region of study, could lead to results that are in better agreement with the mentioned proxy-based reconstructions.

In Fig.6 we present the anomalies of temperature summer and winter seasonal means between 6000BP and the Pre-industrial period, as reproduced by the different models and the pollen-based reconstructions. From these maps we first notice, in both the seasons, that a similar signal of climate change is present in all the simulations. This is expected, being, in every case, the data constrained by the coarser resolution models. Nevertheless, while the highly resolved simulations allow us to detect

a warmer bias over Northern Europe in winter, also present in the proxy data, the ECHO-G does not present such behaviour. Additionally, the land-sea area in the ECHO-G is considerably different than the ones of the other models. Regions such as Southern Spain, the Black Sea area, Southern Italy and Scandinavia are partly or completely masked-out in this case.

Consequently, we focus further analysis on the comparison between the ECHAM5 and the CCLM

results. In both seasons additional details are easily detectable in the CCLM pattern. The coastline is also better reproduced in this case, resulting in a better detailed representation of the land-sea contrast, a more precise reproduction of surface processes and, consequently, leading to more suitable information for possible comparison against proxy-data. Nonetheless, the CCLM shows better defined patterns as a consequence of higher resolution, being able to discriminate higher spatial

variability.

On the basis of such analysis, in the successive step, we try to quantify how better the CCLM reproduces the reconstructed temperatures in comparison to the ECHAM5. Under the mentioned considerations, we use an approach similar to the one employed by Zhang et al. (2010) and based on the work of Goosse et al. (2006). After regridding, by bilinear interpolation, the CCLM and the

295 ECHAM5 results on the reconstructions grid, we introduce a Cost Function defined as:

$$CF_{mod}^k = \sqrt{\frac{1}{n}\sum_{i=1}^{n}\omega_i{}^k(T_{rec,i}^k - T_{mod,i}^k)^2} \tag{1}$$

where $CF_{mod}^k$ is the value of the cost function for each considered time slice of mid-to-late Holocene $k$ and each model $mod$. The parameter $n$ represents the number of the reconstructions' grid boxes. $T_{rec,i}^k$ is the temperature of the proxy-data at every location $i$, while $T_{mod,i}^k$ is the cor-

300 responding temperature of the model simulation. Additionally, the parameter $w_i^k$ takes into account the uncertainties of the reconstructions at every location and time period. Its value is given by:

$$\omega_i^k = \frac{1}{(SE_i^k)^2 + 1} \tag{2}$$

where $SE_i$ represents the standard error of the reconstructions at every grid box $i$. The corresponding uncertainties of the model results are considerably small ($\sim 0.01^oC$) in comparison to the ones of the reconstructions, similarly to Goosse et al. (2006), and are indeed neglected. In this way reconstructions with higher uncertainties will contribute less in the calculation of the Cost Function. The values of the Cost Function for the two models are provided in Tab.2. Values closer to 0 indicate a better agreement with proxy reconstructions.

As we can notice, even if not particularly large differences are present, the Cost Function computed for the CCLM is in almost all the cases smaller than the one for ECHAM5. In particular the CCLM results are, in some cases, closer by more than $10\%$ to the reconstructions. It is important to mention that the scale of the pollen-based reconstructions, considered for our analysis, is closer to the resolution of the ECHAM5 than of the CCLM. As suggested by Di Luca et al. (2015), given that the main difference between the GCM and the RCM is related to their horizontal resolution, it seems natural that the results depend on the spatial scale of the analysis. Additionally, it is key to state that the evinced results are relative to this case study and other comparisons should be performed, considering different couples of RCM-GCM, in order to derive more robust conclusions on the suitability of higher-resolution models for the comparison against proxy-reconstructions. Nonetheless, the motivation behind producing higher resolution climate simulations is not only related to scientific arguments of the type described above. From a different perspective, such results, due to the greater level of detail, could be preferable for applications in studies in which human adaptation or environmental response to past climatic changes would be investigated. Accordingly, the need for climate information at very fine scales, for applications such as archaeology or vegetation reconstructions, constitutes a strong incentive to perform higher-resolution climate simulations (Di Luca et al., 2015; Rummukainen, 2016). The evinced results and the proposed discussion, give us concrete motivation for the choice of conducting RCM simulations for this particular case study.

### 3.3 The CCLM results and their Anomalies in the Comparison with Reconstructions

Finally we focus on the comparison between the CCLM results and the pollen-based reconstructions. After analysing the differences between the two datasets and their temporal evolution, we propose, by means of correlations with trends of insolation and changes in atmospherical circulation patterns, physically plausible interpretation of the evinced mismatches.

Fig.7 and Fig.8 present the temperature biases between the two datasets for winter and summer seasonal means, respectively. These are calculated, after upscaling the CCLM results on the grid of the pollen-based reconstructions by bilinear interpolation, for every time slice of mid-to-late Holocene. Additionally, they are accompanied by the maps of the corresponding pollen uncertainties.

In winter, generally colder conditions are reproduced by the model over northern continental Europe, with slightly warm biases over most of the South (Fig.7). In Scandinavia a negative bias is present

for the first two millennia, after which the situation then reverses. The largest anomalies (in some cases up to $\sim 4^o C$) are present over Northeastern Europe (likely related to high pollen-data uncertainty partly due to the fact that seasonal values derived from pollen in this area are biased towards the winter season) and Turkey.

In summer, instead, CCLM results present a positive bias over most of the domain, with particularly pronounced anomalies (in some case larger than $\sim 4^o C$) over different parts of Southern Europe and the Mediterranean basin (Fig.8).

In addition to the previous analyses, the maps of temperature temporal evolution are presented in Fig. 9. They show the slope of the mid-to-late Holocene linear trends of temperature anomalies with respect to the pre-industrial period, calculated, at every grid box, by means of a weighted least squares method, taking into account the contribution of the different uncertainties. The points for which the trends are not significant, according to a F-test at a significance level of 10%, are masked out in grey.

From these maps we see that in winter, even if over part of Southern Europe the two datasets present similar trends, thei behaviour is different in the North: CCLM results show no significant trend (Fig.9a), while the pollen-based reconstructions present significantly decreasing temperatures over a considerable part of the domain (Fig.9b). In particular, over Scandinavia, while the pollen-based reconstructions show a strong, significant cooling trend, no significant trend is evident for the model results. Conversely, in summer, the model results are characterized by a negative trend over most of the domain (Fig.9c), highly correlated to changes in insolation. The pollen data, instead, show a significant negative trend similar to the CCLM results over part of Northern Europe only, and an opposite positive trend over most of the South (Fig.9d). Since changes in atmospheric circulation have often been suggested as possible drivers of temperature evolution during mid-to-late Holocene winters and summers (Bonfils et al., 2004; Braconnot et al., 2007a; Fischer and Jungclaus, 2011; Mauri et al., 2014), in order to obtain further insights into the causes of the evinced biases, we conduct a Canonical Correlation Analysis (**CCA**) of model's mean sea level pressure and temperature anomalies, with respect to the pre-industrial period, for winter and summer seasons. The Canonical Correlation Analysis is particularly suitable for our purposes since it helps to identify spatial patterns of maximum correlation between climate variables, indicating potential underlying physical mechanisms (Wilks, 1995; von Storch and Zwiers, 1995). In CCA, according to Gómez-Navarro et al. (2015), "from a physical point of view, the leading patterns should show similar characteristics when the mechanisms leading to the relationships between the climate fields are controlled by the same processes".

In our analysis we adopt the method of Barnett and Preisendorfer (1987) in which a EOF analysis is conducted prior to the CCA, retaining only a few leading EOFs, in order to remove part of the random noise from the data. More specifically, after conducting the EOF analysis on the anomalies, with respect to the pre-industrial period, of MSLP and T2M, we select the first eight principal

components of both the variables in winter, and the first eight and twelve principal components of, respectively, MSLP and T2M in summer. In this way, in both the cases, the selected PCs will explain approximately 80% of the total variance in the original datasets. We then apply the CCA analysis on the retrieved PCs.

Fig.10 and Fig.11 show the first two canonical pairs of patterns with the largest canonical correlation for both winter and summer.

The MSLP pattern explaining most of the variance, in winter, resembles the NAO (Fig.10c). The model seems to reproduce well the spatial pattern of the NAO when compared to other studies (Gómez-Navarro et al., 2015). Nevertheless, the trend of the temporal evolution of its expansion

coefficients (Fig.12c), seems not to be pronounced enough in order to reproduce a response in temperatures comparable with the respective results of pollen data. Additionally, the value of the canonical correlation, even if high, is slightly smaller than the one of a secondary mode of atmospheric variability, in this case represented by a blocking system centered over the Baltic Sea. The trend of the expansion coefficients of this pattern is slightly positive but again not particularly pronounced.

As a result of the combined effects of the evinced patterns of atmospheric variability, the CCLM temperature trends will be significant only over part of Southern Europe.

In summer, instead, the first CCA pair (Fig.11 a,b) seems to be highly related to changes in insolation (Fig.13 a,b). It is key to note that, the first canonical pattern of summer MSLP anomalies and its structure, seems to be a proper product of this particular case of study. Even if it implies changes

in circulation, we do not see any particularly prominent dipole structure characteristic of other well-known circulation patterns for the region. Its effects on temperature are particularly high on the Atlantic coast of continental Europe, resulting in a smoothing of the trend of summer temperature over this region.

In the second CCA pair, the pattern of the mean sea level pressure (Fig.11 c) resembles the positive

phase of the Summer North Atlantic Oscillation (SNAO) (Folland et al., 2009). The trend (Fig.13 c) of its expansion coefficients is again not particularly pronounced. As a consequence, the changes in the corresponding temperature pattern (Fig.13 d), are also not remarkable.

Consequently, we suggest that in summer, during mid-to-late Holocene, the changes in circulation alone would not have been enough to justify the variations in surface temperature, as recon-

structed from the proxies. While over Northern Europe the relatively good agreement between the temperature of the two datasets over part of the domain suggests that for this region the insolation is probably the main driver of changes, for Southern Europe, however, the role of land-atmosphere coupling needs to be considered (Seneviratne et al., 2006). According to Bonfils et al. (2004) and Starz et al. (2013), over Southern Europe, the presence of more moisture in the soil during mid-

Holocene summer, due probably to more winter and early spring precipitation, is responsible, as a direct effect of higher insolation, for cooler conditions due to stronger latent heat transfer. According to the mentioned studies and to the previously presented analyses of model's heat-fluxes, we sup-

port this interpretation and suggest that the reason why the model does not manage to capture the reconstruction temperature trend, could most probably be related to a wrong reproduction of soil-
atmosphere heat exchanges. As previously discussed, the scarce ability of the model to correctly reproduce the soil-atmosphere fluxes for this area, leads to an underestimation of evaporation and, consequently, to drier and warmer conditions. Further experiments, with improved soil properties, are indeed necessary in order to better reproduce soil moisture content, and to obtain more robust results for the comparison with reconstructions.

It is important to mention that the behaviour of mid-to-late Holocene's summer temperature over Europe has been highly debated during recent years. While a dipole behaviour has been suggested by several studies based on pollen analyses (Huntley and Prentice, 1988; Cheddadi et al., 1997; Prentice et al., 1998; Davis et al., 2003; Mauri et al., 2015) and others relying on a combination of different proxies, such as the one of Magny et al. (2013), which suggested a North-South paleohydrological contrast in the central Mediterranean during the Holocene, other studies argued against such hypothesis. In particular Osborne et al. (2000) proposed that reconstructions of summer temperature based on pollen could be erroneous for the Mediterranean region, since here the vegetation distribution is mainly limited by effective precipitation, rather than by summer temperature. The latest hypothesis should be taken into account for the comparison between pollen-based reconstructions and model simulations. Nevertheless, additional investigations have shown that, when directly compared to the pollen record, the mid-Holocene vegetation simulated from the output of climate models is way too dry over Southern Europe, with an expansion of Mediterranean and steppe/desert vegetation and contraction in forest cover, a direct consequence of simulated warmer conditions (Prentice et al., 1998; Wohlfahrt et al., 2004; Gallimore et al., 2005; Benito Garzon et al., 2007; Kleinen et al., 2010).

Based on these considerations, recognizing the dataset of Mauri et al. (2015) as a valuable source for the investigation of European temperature evolution during mid-to-late Holocene, we acknowledge the fact that joint efforts from specialists of different disciplines are still required in order to further clarify possible uncertainties.

### 3.3.1 Other Modelling Studies

An important benchmark for the comparison of our results against other modelling studies is represented by the outcomes of the PMIP3 experiment (Braconnot et al., 2011), for which several simulations have been performed, with different coupled circulation models, for the mid-Holocene and the pre-industrial time. Here we focus on the results of twelve of the PMIP3 simulations. Specifically, we perform a direct comparison of the regional mean of winter and summer near surface temperature calculated for Northern and Southern Europe for the PMIP3 simulations as well as each of ours The results are presented in two tables, provided as supplementary material, in which the corresponding values derived from the pollen-based reconstructions are also included. Two main features arise from such analysis: first of all a common positive bias ($\sim +1^{o}C$) over Southern Europe in summer for all

the models is evident, while the reconstructions present a negative value ($\sim -1.2^oC$). This indicates that the temperature differences are positive in the model simulations as a result of the higher summer insolation at mid-Holocene than at the pre-industrial period. Additionally, another feature that seems to be common to all the models is represented by the failure in representing winter anomalies in both the regions and it is attributable to a wrong reproduction of changes in the amplitude of NAO (Fischer and Jungclaus, 2011; Strandberg et al., 2014). While some models present a value similar to the one of reconstructions for Southern Europe ($\sim +0.5^oC$), in the North the differences are significant, with the pollen-based reconstructions presenting a warm bias ($\sim +2.5^oC$), and the models having slightly positive values (between 0 and $+1^oC$) in some cases, and negative (up to $\sim -1^oC$) in the others.

## 4 Summary and Conclusions

In this work we performed for the first time a set of highly resolved climate simulations over Europe for different time-slices of mid-to-late Holocene, by means of the state-of-the-art regional climate model COSMO-CLM.

As a first step, using the CRU and the E-OBS observational datasets as benchmarks, a model setup suitable for paleoclimate investigations has been tested for the reference period 1991-2000. The results show that the RCM is able to reproduce realistic climatology with respect to the observations. The largest biases arise in summer over Southern Europe where the model reproduces warmer and drier conditions ($\sim +4^oC$ for temperature and $< -50\%$ for precipitation), likely related to a wrong conversion of energy towards latent heat over this area. Nevertheless, the results are in good agreement with the ones of other studies for the same region, and the employed configuration can be considered a valid reference for future applications.

Successively, the results of mid-to-late Holocene simulations have been compared against a new pollen-based climate reconstruction dataset. Winter and summer seasonal means of near surface temperature have been considered for our analysis.

To begin with, the possible advantages of higher resolution models for paleoclimate applications have been investigated. The RCM seems to better reproduce the signal of the climate-reconstruction when compared to the driving GCMs, with a more detailed reproduction of the coast-line and better defined patterns. Additionally, using a quantitative approach, we have demonstrated that the results of the RCM are closer to the values of the reconstructions in comparison to the driving GCM, in some cases by more than 10 %. Considering also the final user perspective, the evinced results gave us concrete reasons for choosing to conduct highly resolved simulations for this particular case study.

Finally, the CCLM results are used in order to investigate the response of the climate system to changes in the seasonal cycle of insolation, with the aim of proposing plausible physical interpretations of the mismatches arising in the comparison against the reconstructions.

The results show that, in winter, over Southern Europe temporal behaviour and spatial distribution of temperature in the two datasets are comparable. Conversely, the model tends to reproduce generally colder conditions over central and northern continental Europe. Through the analysis of atmospheric circulation patterns we argue that this bias is due to a different representation by the model of the expected changes in circulation, as a result of reduced influence of westerly winds and an increased importance of secondary modes of atmospheric variability. Additionally, larger differences are present in Northeastern Europe, likely related to high uncertainties of pollen data over this area.

In summer, the simulated northern conditions agree well with the proxy data over part of the domain. Their behaviour seems to be a direct response to insolation changes. Conversely, while the model produces warmer summer conditions over Southern Europe at mid-Holocene, in comparison to pre-industrial times, again mainly due to insolation changes, the pollen data exhibit an opposite trend. According to the results of previous works and to the analysis of atmospheric dynamics, we suggest that this behaviour is mainly due to a higher partition of radiation towards latent heat, resulting in a cooling effect of the surface that the model is not able to reproduce due to deficiencies in the representation of soil-atmosphere heat fluxes over this area.

This paper sets the basis for further investigations: in particular a set of new simulations with improved radiation schemes, soil properties and land use, could lead to important contributions to climate modelling and, consequently, to the improvement of future climate change projections.

*Acknowledgements.* The authors are grateful to the two anonymous referees for their constructive comments that helped to considerably improve the manuscript. This paper was supported by the Cluster of Excellence "Topoi - The Formation and Transformation of Space and Knowledge in Ancient Civilizations". The computational resources were made available by the German Climate Computing Center (DKRZ) and the Freie Universität Berlin (ZEDAT). We would like to thank Achille Mauri and Basil Davis for providing the pollen-based reconstructions and for their continuous support and constructive discussions. We would also like to express our sincere appreciation to Janina Körper for designing and conducting the ECHAM5 climate simulations. A particular acknowledgment goes to Edoardo Mazza for his continuous support and intellectual debate. We would also like to thank Ingo Kirchner, Bijan Fallah, Nico Becker, Alexander Walter and John Walter Acevedo Valencia for the fruitful and interesting discussions. Additionally, we are particularly greatful to Jacqueline Harvey for proofreading the manuscript.

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

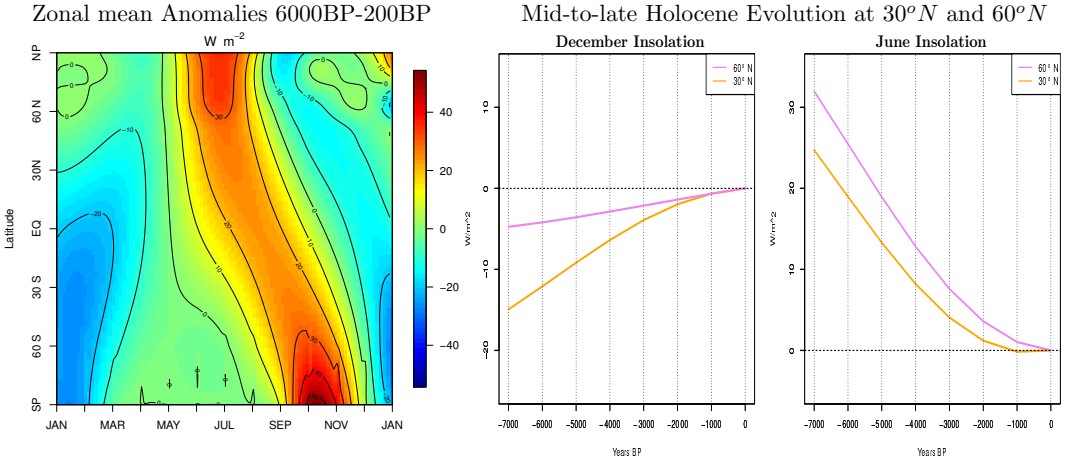

**Figure 1.** *(Left) Anomalies of zonal mean insolation on top of the atmosphere (TOA) between pre-industrial period (PI) and 6000 years BP. (Right) Mid-to-late Holocene trends of the anomalies, with respect to present-day values, of December and June TOA incoming insolation, calculated, according to Berger (1978), for 30 and 60 degrees North. Units are $W/m^2$.*

**Table 1.** COSMO-CLM Main model configuration parameters

| | |
|---|---|
| Convection | Tiedke |
| Time Integration | Runge-Kutta, $\Delta T=240s$ |
| Robert-Aselin time filter (alphaas) | 0.53 |
| Lateral Relaxation Layer | 500Km |
| Radiation | Ritter and Geleyn |
| Turbulence | Implicit treatment of vertical diffusion using Neumann boundary conditions |
| Rayleigh Damping Layer (rdheight) | 11Km |
| Soil Active Layers | 9 |
| Active Soil Depth | 5.74m |

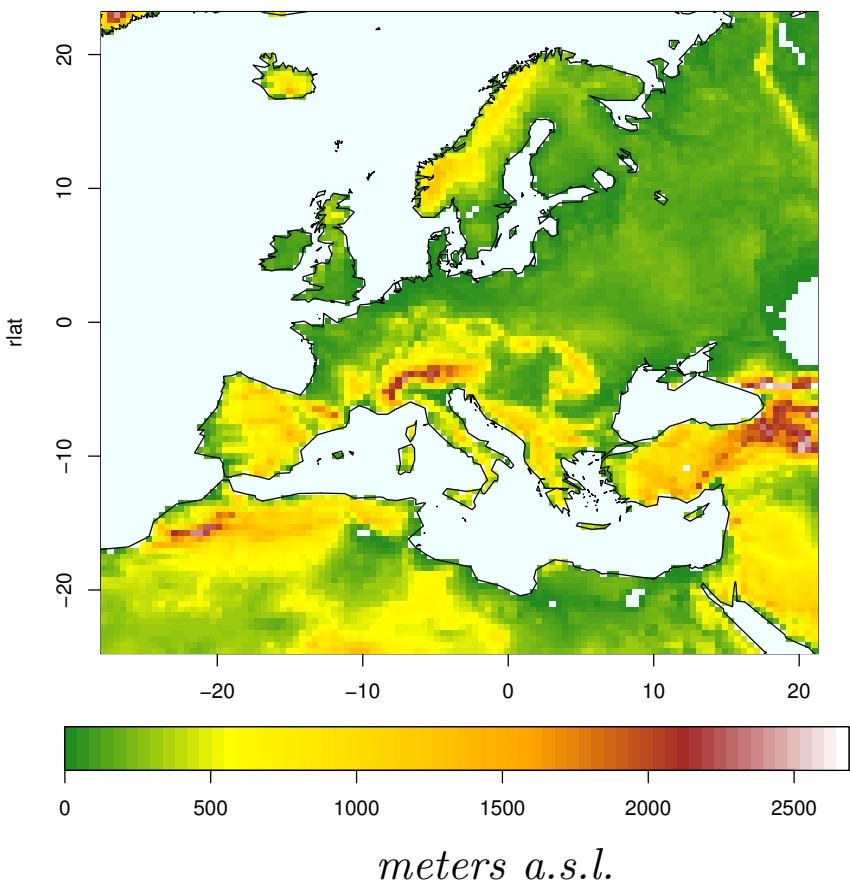

**Figure 2.** *Orography map of the COSMO-CLM simulation domain in rotated coordinates.*

**Table 2.** *Winter (left) and summer (right)Temperature Cost Function estimates for the CCLM and the ECHAM5 models compared to the Proxy reconstructions for each time slice of mid-to-late Holocene. Values closer to 0 indicate a better agreement with proxy reconstructions.*

| Time Slice | Winter | | Summer | |
|---|---|---|---|---|
| | **CCLM** | **ECHAM5** | **CCLM** | **ECHAM5** |
| 6000BP | 0.87 | 0.92 | 0.93 | 0.96 |
| 5000BP | 0.88 | 0.92 | 0.72 | 0.72 |
| 4000BP | 0.77 | 0.84 | 0.65 | 0.67 |
| 3000BP | 0.78 | 0.82 | 0.63 | 0.71 |
| 2000BP | 0.77 | 0.79 | 0.48 | 0.54 |
| 1000BP | 0.61 | 0.61 | 0.43 | 0.48 |

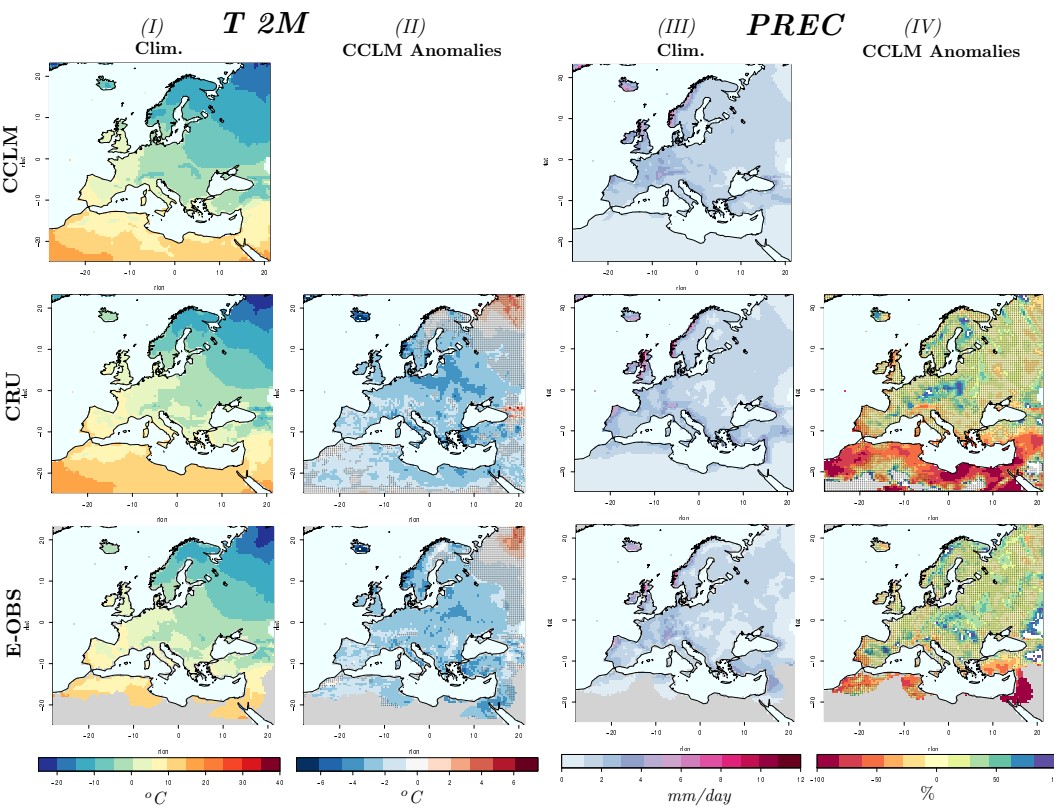

**Figure 3.** *Analysis of Winter seasonal means of 2 meter temperature (left panel) and Precipitation (right panel) for the period 1991-2000. The first column of each panel (I,III) shows the mean climatology for the investigated period as represented in the three considered datasets: the CCLM in the first row, the CRU in the second and the E-OBS at the bottom. The second columns show (II,IV), instead, the biases between the CCLM results and the respective observational datasets. The area with a point represent the grid cells where the anomalies between the two datasets are not significant, according to a Student's T-test, at a significance level of 5%.*

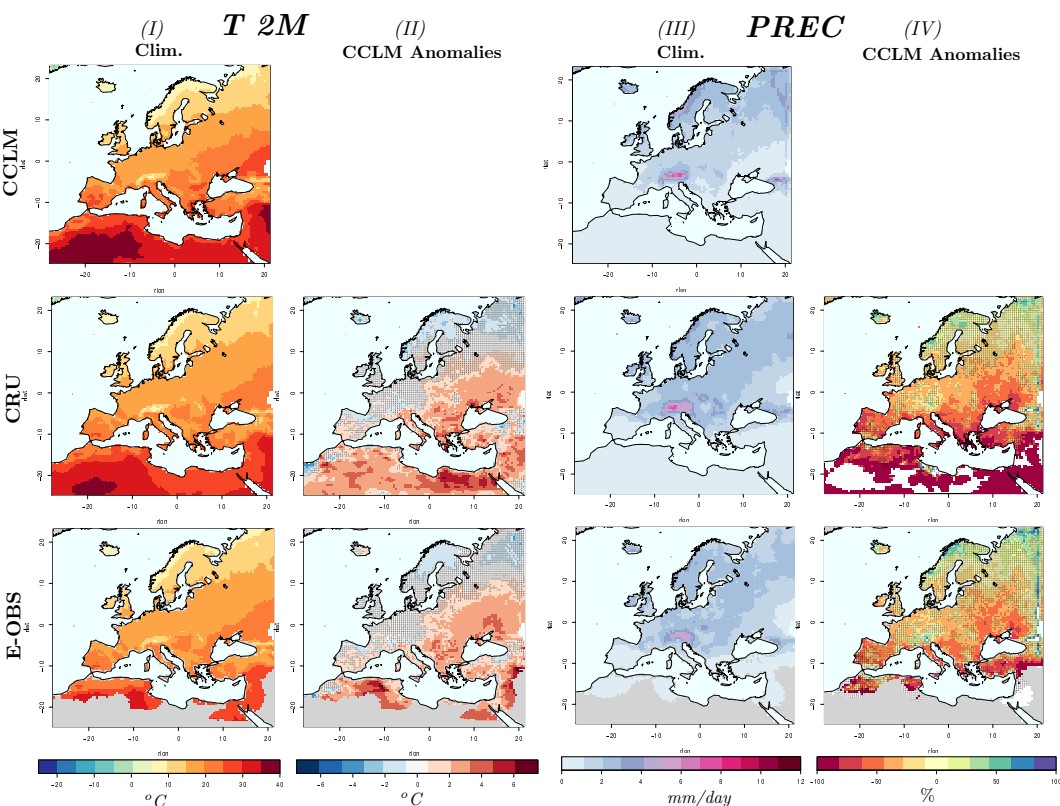

**Figure 4.** *As Fig3 but for Summer.*

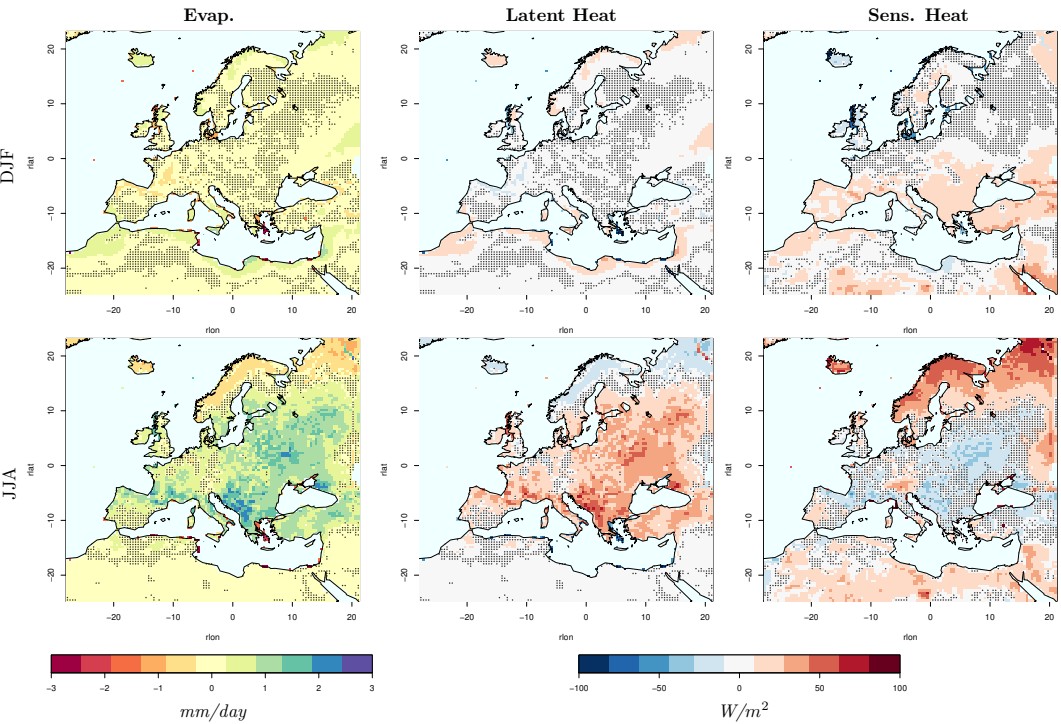

**Figure 5.** *Biases of seasonal means of Evapotranspiration (left), Latent (center) and Sensible Heat (right) fluxes, between the CCLM simulations and the GLDAS dataset, calculated for the reference period 1991-2000. As in the previous figures, the area with a point represent the grid cells where the anomalies between the two datasets are not significant, according to a Student's T-test, at a significance level of 5%. Winter results are presented in the first row, and Summer results in the second.*

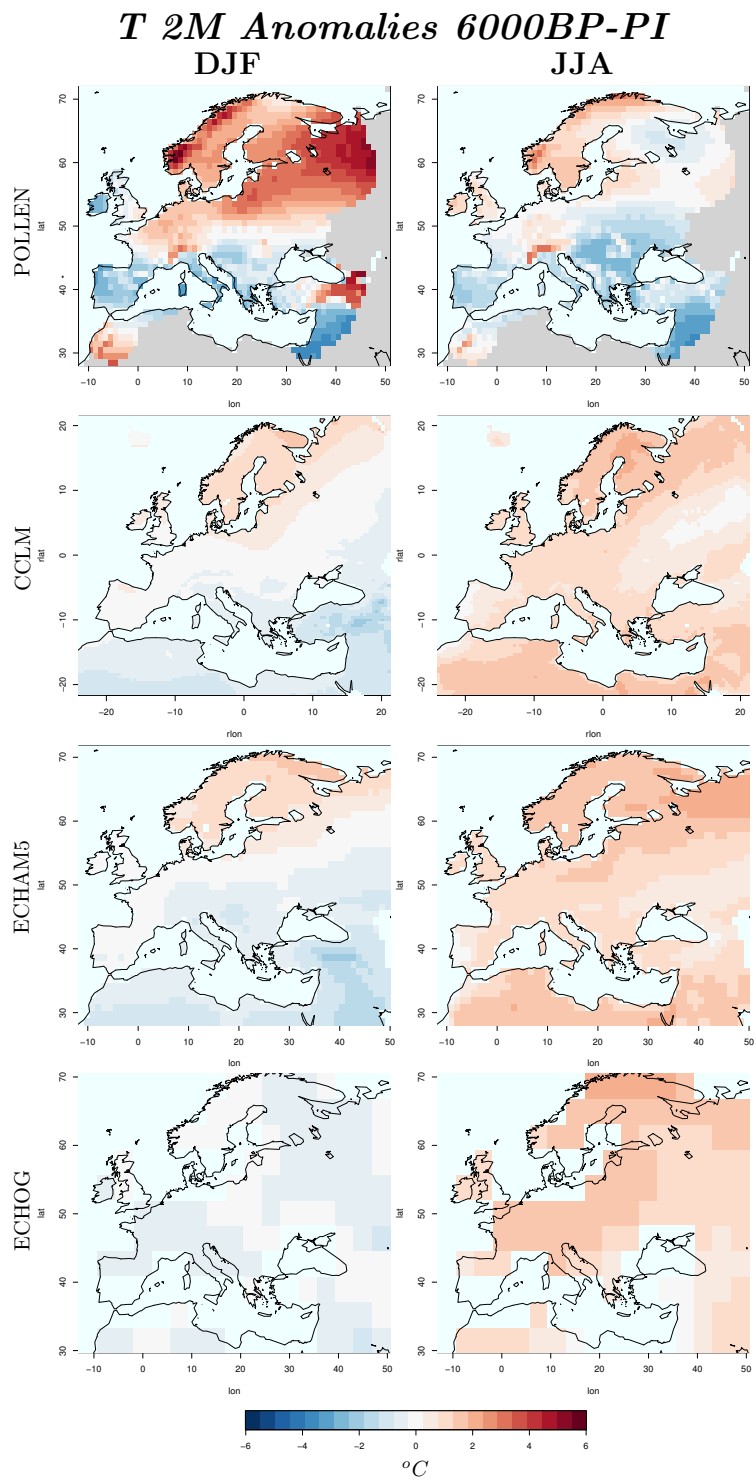

**Figure 6.** *Maps of the anomalies between 6000BP and the preindustrial period of Winter (left) and Summer (right) seasonal means of 2 meters temperature, calculated over a 25-year period. The results of the three different models and the pollen-based reconstructions are presented. From top to bottom: POLLEN-based reconstructions, CCLM, ECHAM5, ECHO-G. The results are presented on each dataset original grid: the CCLM data, in particular, are shown in rotated geographical coordinates.*

# Winter

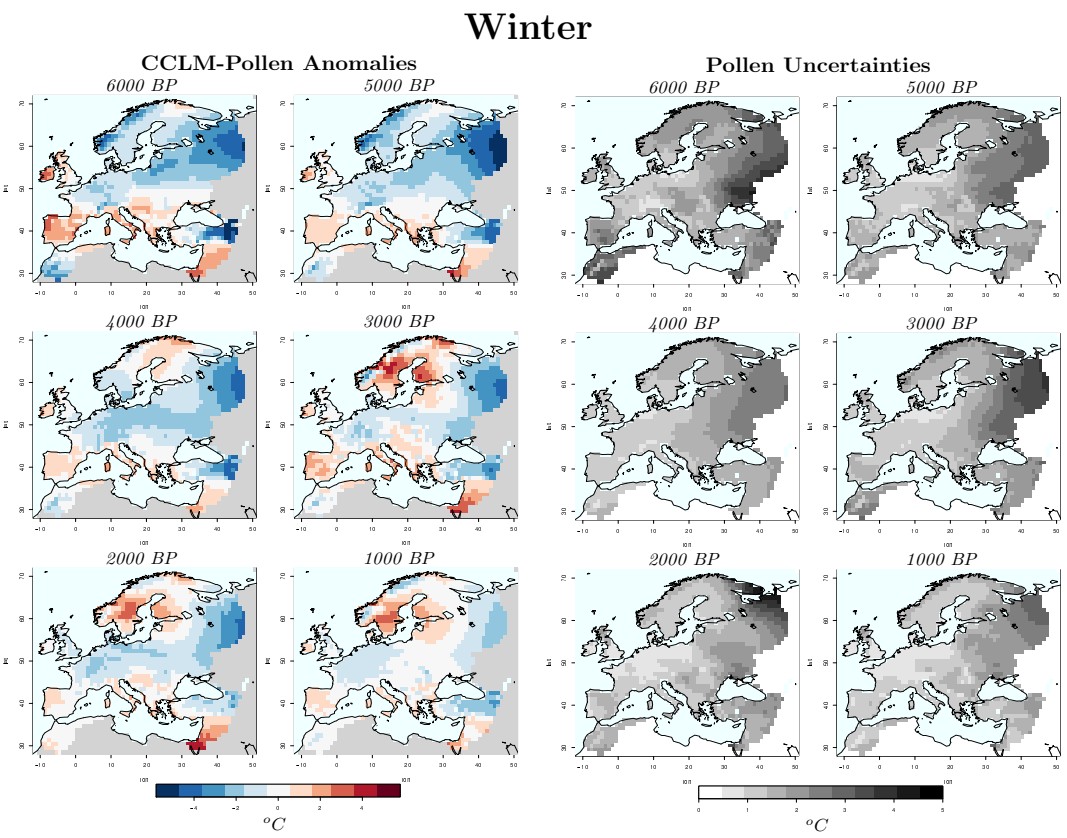

**Figure 7.** *Left: Maps of Winter 2 meters temperature anomalies between CCLM and Pollen-based Reconstructions for the different time slices of mid-to-late Holocene. Right: Standard error of winter temperature seasonal mean derived from the pollen-based reconstructions for each time slice of mid-to-late Holocene.*

# Summer

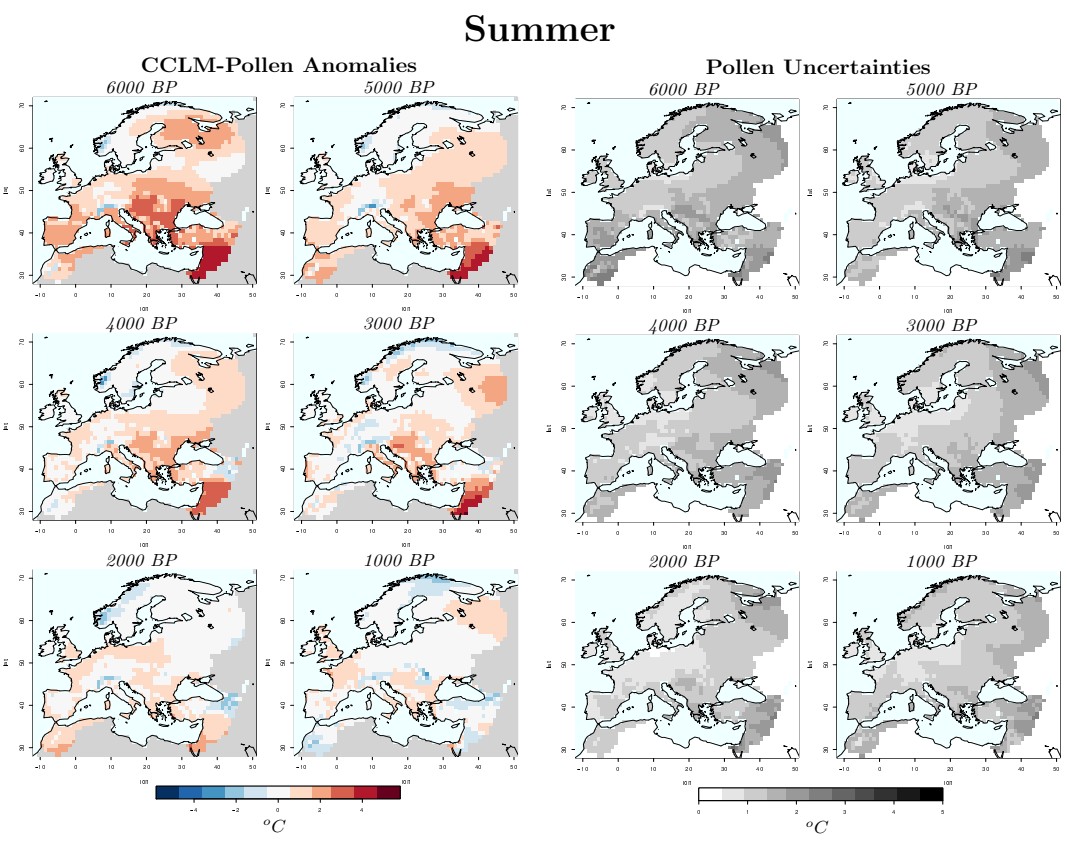

**Figure 8.** *As in Fig.7 but for Summer seasonal means.*

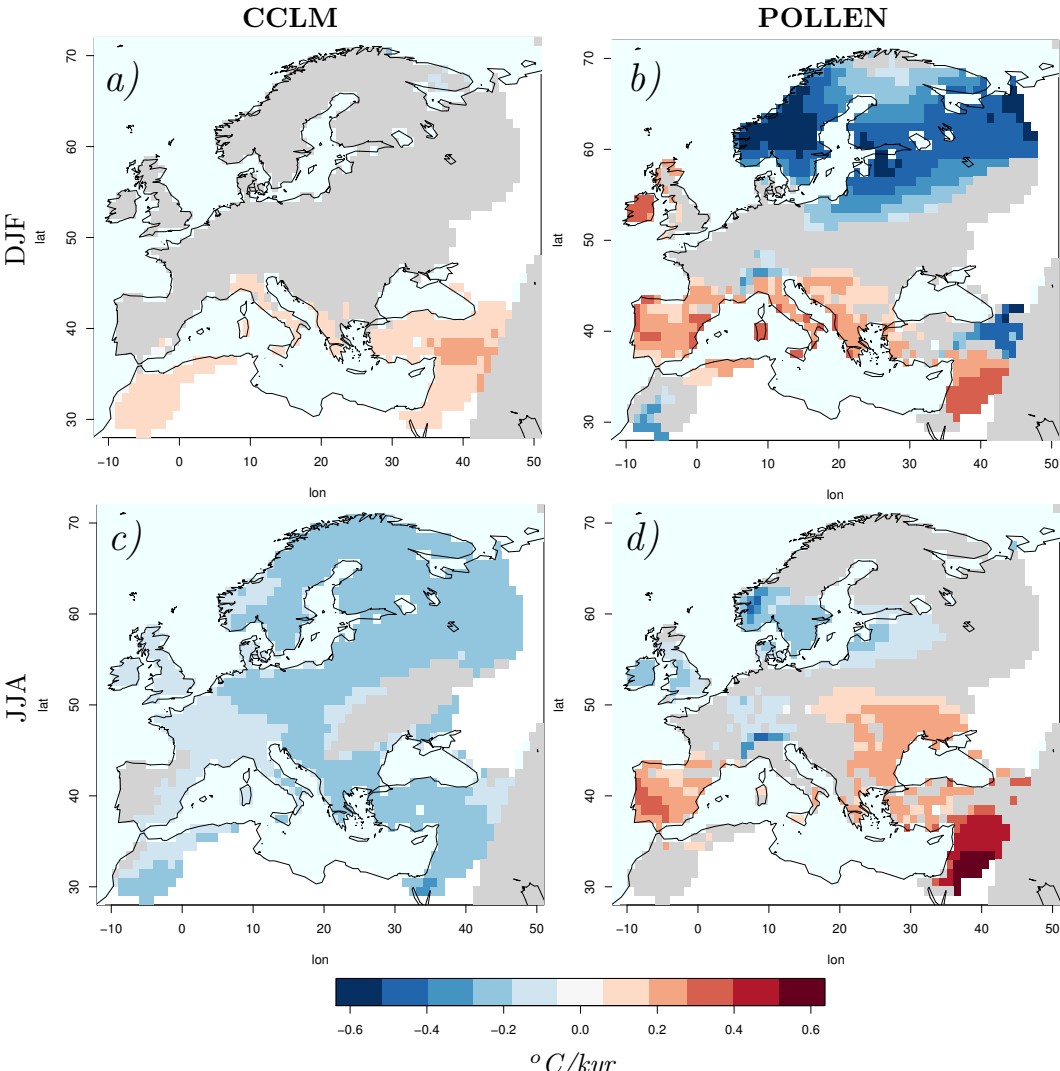

**Figure 9.** *Mid-to-late Holocene temporal Evolution of the anomalies, with respect to the pre-industrial period, of near surface temperature winter (first row) and summer (second row) seasonal means, derived from the CCLM simulations (left) and the pollen-based reconstruction (right). The maps show the slopes of the linear trends calculated, for every grid box, taking into consideration the uncertainties associated to the two datasets, by means of a weighted least squares method. The area masked out in grey, are the area where the trends are not significant, according to a F-test at a significance level of 10%.*

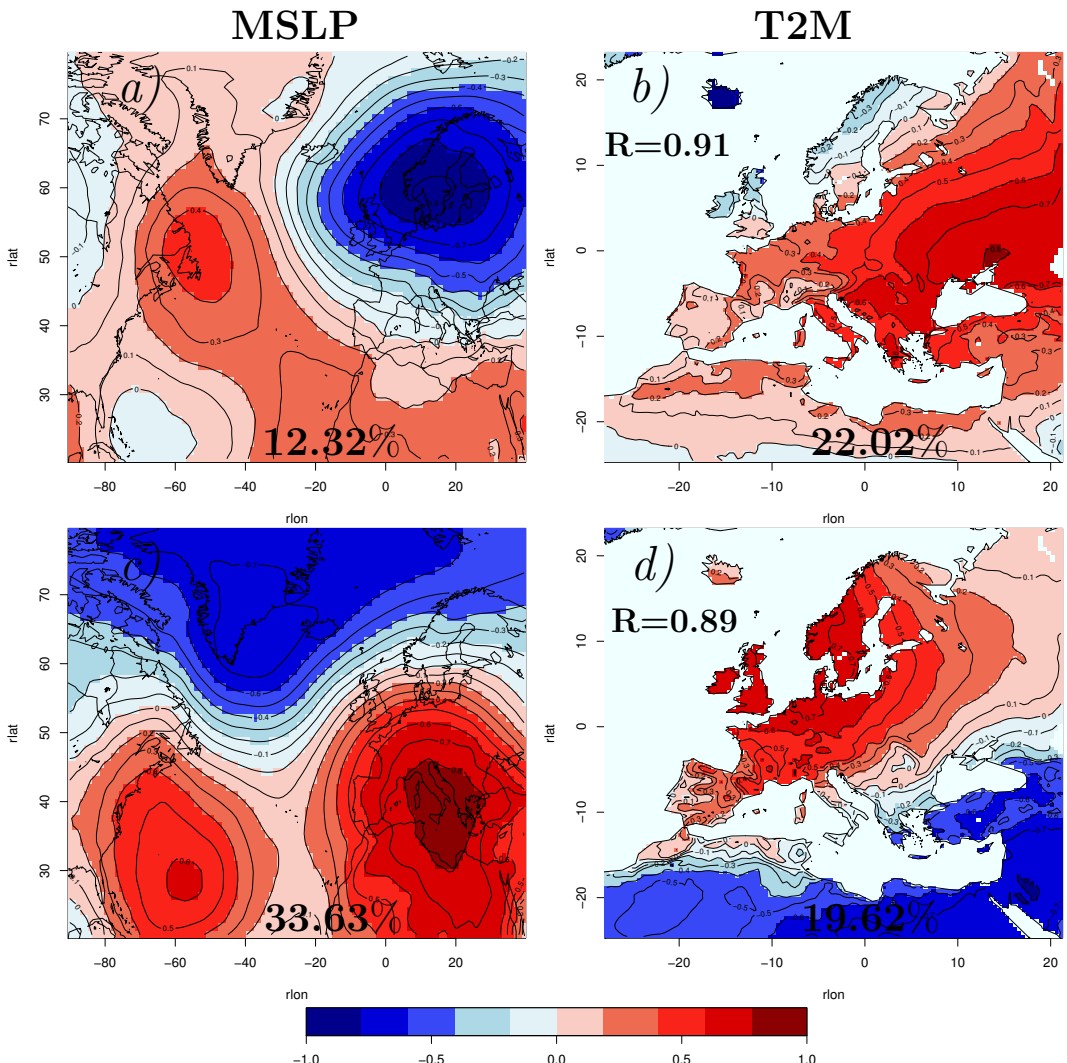

**Figure 10.** *Canonical correlation pattern pairs of MSLP (left) and T2M (right) in Winter, calculated accordingly to the Barnett and Preisendorfer (1987) method. Each panel illustrates the percentage of variance explained by the patterns and the canonical correlation associated with the pair. The results are calculated for the mid-to-late Holocene, from 6000BP to Pre-industrial times. Note that the MSLP has been obtained directly from the driving GCM, since the window of interest lies outside the RCM domain. For both the variables the analysis has been conducted on the standardized anomalies with respect to the pre-industrial period. Red (blue) areas indicate positive (negative) correlations, for each grid point, between the data and the corresponding canonical score series.*

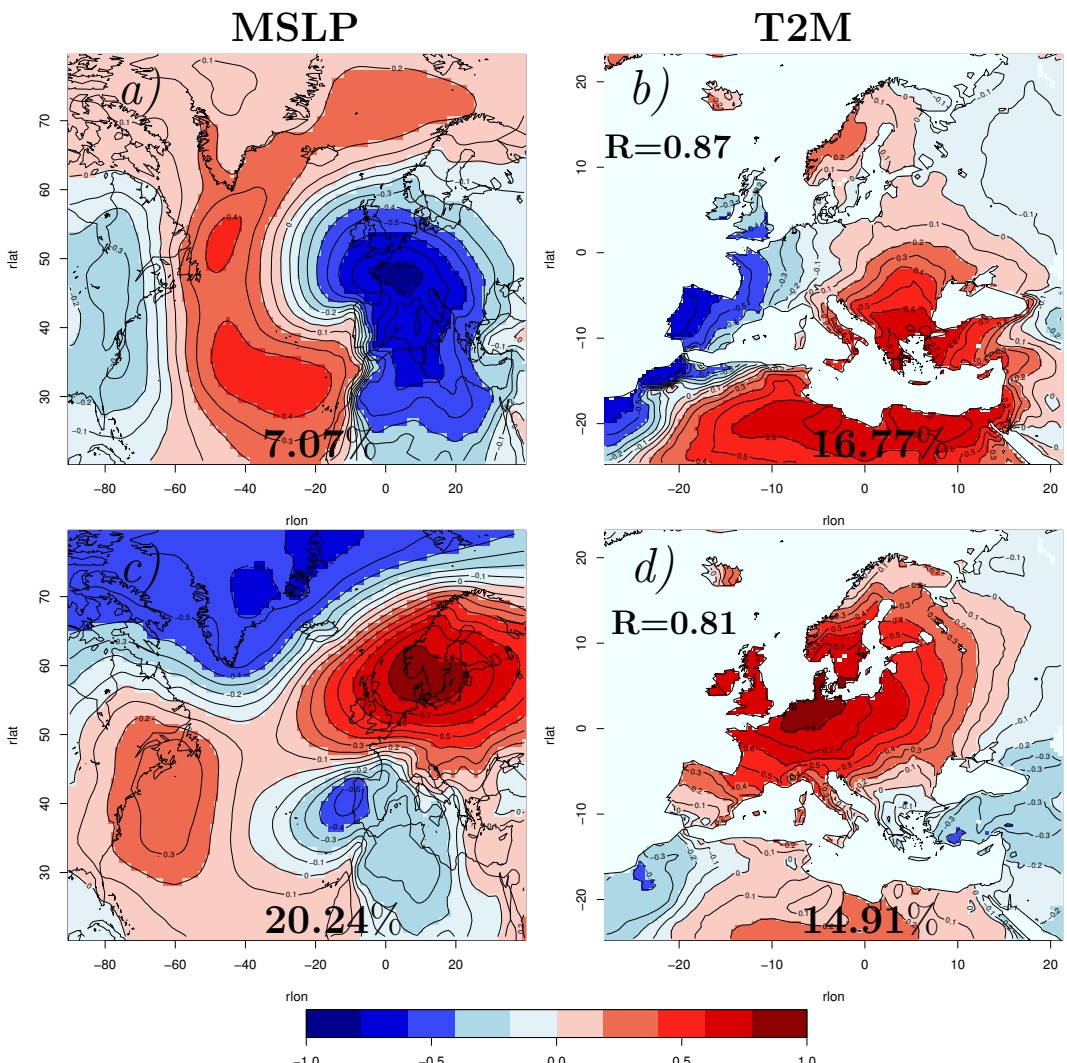

**Figure 11.** *As in Fig.10 but for Summer season.*

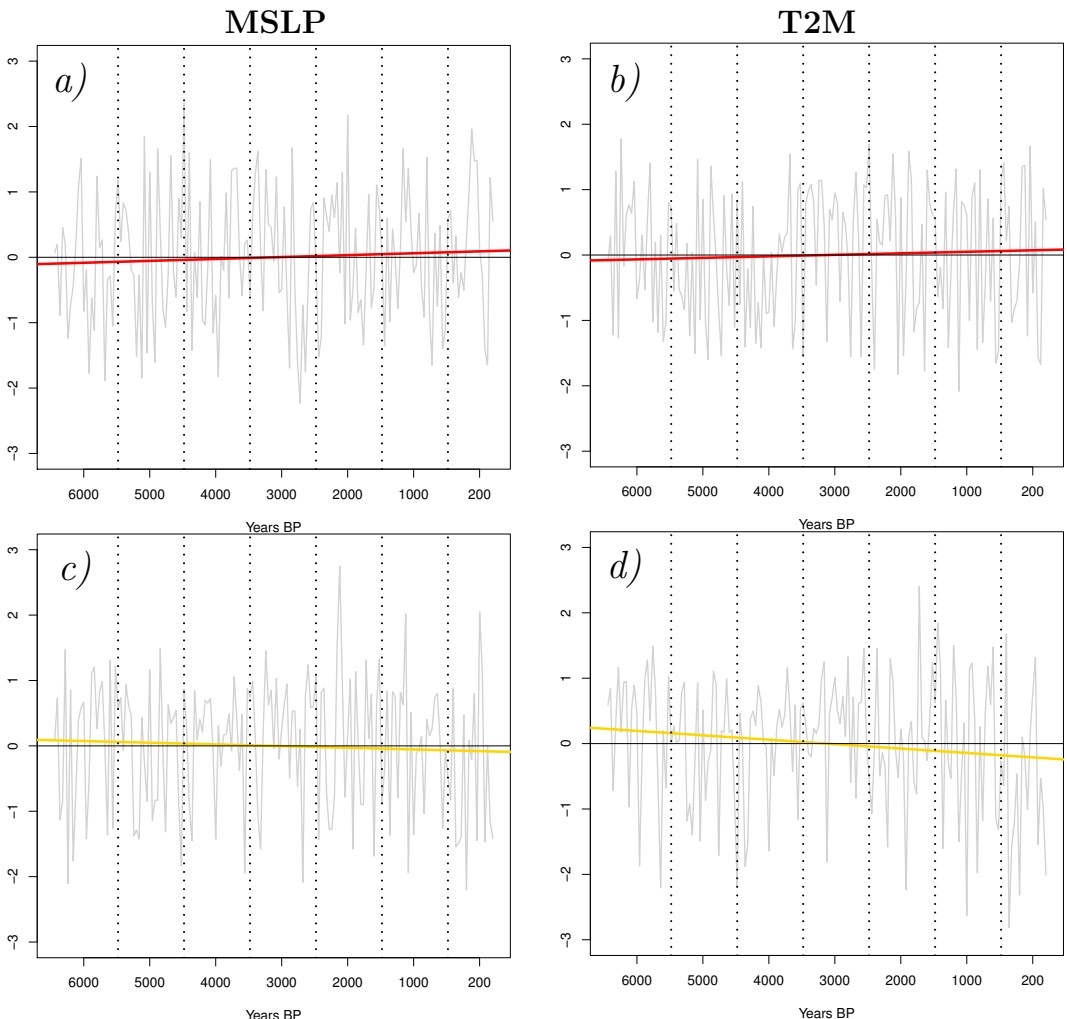

**Figure 12.** *Canonical score series of the first two pairs of Canonical Correlation patterns of, respectively, MSLP (left column) and 2 meter temperature (right column) winter seasonal mean anomalies*

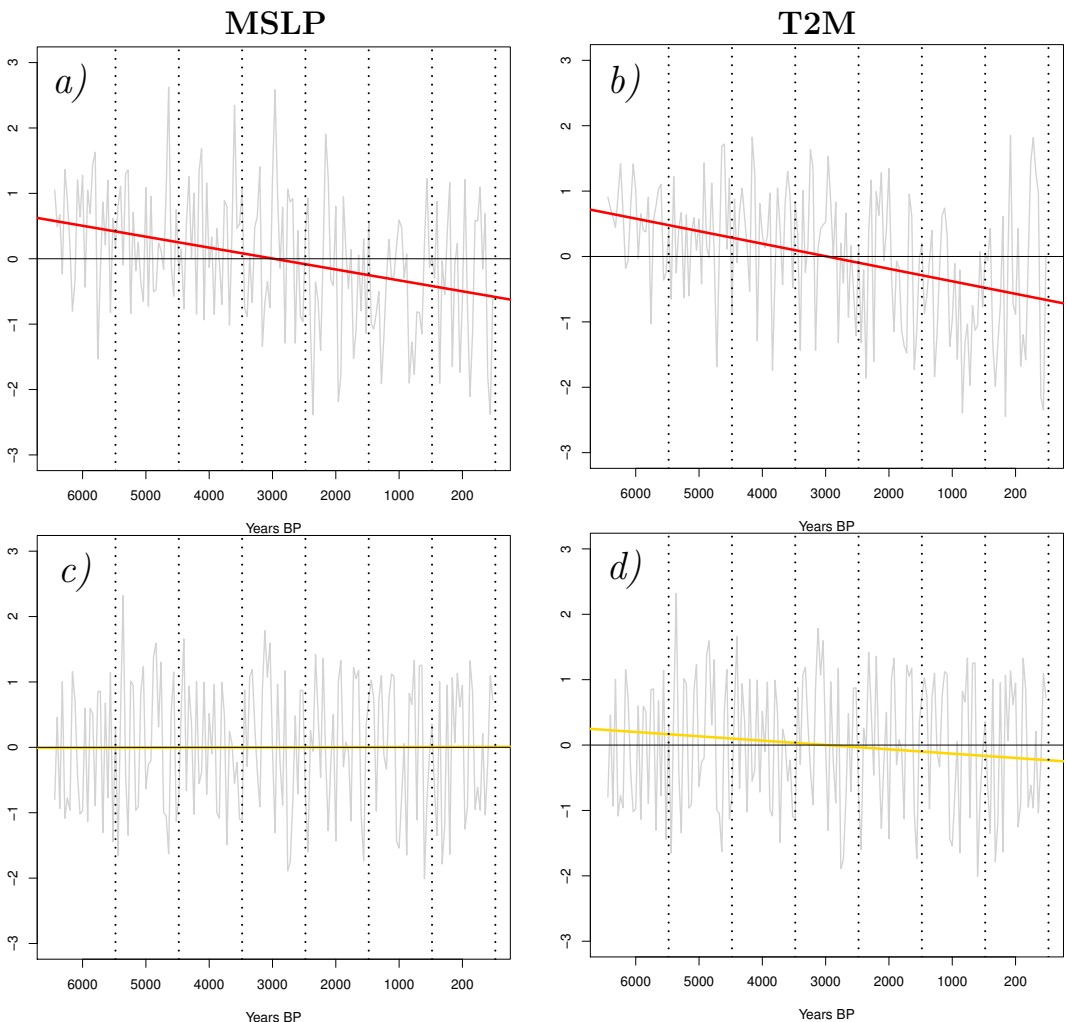

**Figure 13.** *As in Fig.12 but for summer*