# Peer review of "Correlation maps MSLP Time Expansion (PCs) and PREC"

_Climate of the Past, 2016_

## Referee Comment (RC1) · Anonymous Referee #1 · 8 Feb 2016

Review of the article entitled: "Mid-to-late Holocene Temperature Evolution and Atmospheric Dynamics over Europe in Regional Model Simulations" by E. Russo and U. Cubasch

**Summary:**

The manuscript presents a set of time slide simulations with a RCM for the mid-to-late Holocene. The skill of the simulations is stabilised comparing 2 m temperature and precipitation to other datasets for a reference period, and the output of the simulations for the past periods is compared to temperature reconstructions derived from pollen archives. Finally, potential sources for the disagreements identified and sought.

**General comment**

The paper can be broadly divided into three parts: i) validation of the simulation, ii) comparison with reconstructions iii) search for explanations of the disagreements. In all these three elements, I can identify caveats that in my opinion should be improved with different/complementary analyses. I have tried to review them in a comprehensive, yet constructive way, as detailed below.

Besides the technical aspects, I think there is room in the manuscript for improvement regarding writing style. It was challenging for me to read and understand many parts of the paper. This is in part due to incomplete information in the captions and the main text, wrong labelling in the figures, and the misleading use of some concepts as "observation" or "validation". The internal structure in the paragraphs is confusing: paragraphs loosely connected, overly short, or in a misleading order respect to the panels in the figures. These issues add complexity that makes the lecture of the paper uncomfortable. Despite this rather negative view, I try to be constructive giving a list of points that develop the aspects that in my opinion can be improved in the manuscript. Note however that this list is not comprehensive.

**Comments regarding the abstract/introduction**

In the first line, "is always been mentioned" is grammatically wrong. Despite that, it sounds a bit loose, almost sceptical. Is it an important factor or not? This ambiguous tone of the first sentence of the abstract is manifest through the whole manuscript. By the way, the authors do not make an attempt to demonstrate that this is indeed the case for these simulations. More on this below.

The paper is somewhat optimistic regarding the use of "for the first time". It is true that, as far as I know, there are no other set of time slide simulations. However there are various high-resolution simulations for the last millennium for Europe. Actually there exists at least one transient simulations for the last two millennia, in fact driven by the same ECHO-G run used by the authors of the manuscript. The authors should not ignore such previous, yet scarce efforts in this topic in the intro, but also the discussion of the results.

In line 6, "validation" is used in a wrong context. The model is validated normally against observations. But you can not validate the model looking at a reconstruction. Neither you validate a reconstruction looking at a simulation. You can only compare them, and try to gain insight through the disagreements. The use of "validation" in this wrong context is spread through the manuscript and should be avoided.

I think at least the first four paragraphs can be safely merged.

Lines 39 to 41 it is argued that then changes in solar irradiation were "negligible", and latter than "we expect that such changes would imply relevant variations...". It sound contradictory.

The paragraph in lines 42 to 45 is made out of a single sentence, which is too long. Still, the paragraph itself is short and can be merged with the former. Further, such sentence demands references.

In the paragraph starting in line 89, some examples of RCM simulations in palaeoclimate applications are outlined. It's strange to see that no simulation for Europe is referred. Examples of such simulations are: Gómez-Navarro et al. (2011, 2012, 2013, 2015a, 2015b) and Schimanke et al. (2012).

**Comments regarding the model validation**

It is not clear how long is the control period used in section 3.1. The only hint is the label in Figure 2, 1990-2000. Is that the case? It should be clearly stated, not only in the main text, but also in the caption of the figure. Actually, the length of this period is CRITICAL for the model evaluation, a fact that is not acknowledged in the discussion of the results. A 10-year period of a GCM simulation is strongly populated with internal variability. Under this scenario, a comparison with observations is tricky. The model could be "by chance" going through a cold or warm phase, which would have a strong impact in the validation, at least in the way it has been established in the paper, focused on mean values. In this sense, the validation does not look at important aspects such as the variability. How is the variance reproduced by the model? I'm not sure due to the short length of the simulation, but it could make sense to look at the variability modes of temperature and precipitation.

I do not think the choice of target for the validation is the best one. Using ERA Interim for the validation of precipitation in particular is a bad idea, since it is not constrained by precipitation observations, so there is no warranty that this dataset is bias free. I think it would be wiser to use the E-OBS dataset, which was developed specifically for the validation of RCMs in Europe (Haylock et al. 2008).

The way the similarity between the model and the observations is presented is a bit confusing to me. When the difference between two normally distributed variables is shown, the standard and intuitive approach, which steams from the application of the Central Limit Theorem, is to apply a t-test. The KS test is more suited for testing the shape of PDFs when the mean is know to be the same. For example if two dataset have the same mean, but different variance, the figure would show null bias (yellow colour here), but still the test would produce significant differences, which is misleading for the reader.

I think the maps showing precipitation difference are not very useful. A difference of 5 mm/day might be huge or tiny depending on the mean precipitation. I think changes in precipitation are more meaningful when shown as perceptual deviations with respect to the mean.

It is mentioned that the dots indicate grid where differences are "significantly not different". That's not exactly true. They indicate areas where the null hypothesis of the data being sampled from the same underlying distribution could not be ruled out, i.e. where they are "not significantly different".

I agree with the hypothesis used to explain the model deficiencies in Souther Europe regarding soil-atmosphere feedbacks. The particular role of these processes in RCM simulations in areas with strong water deficit was investigated in detail by Jerez et al. (2010, 2012).

Line 206 reads "These findings CONFIRM that… are MOST PROBABLY...". This is an example of doubtful and confusing sentence that should be avoided.

Maybe is worth to mention that generally the model skill resembles that identified in similar

simulations for Europe (Schimanke et al. 2012, Gómez-Navarro et a. 2011, 2013).

**Comments regarding the comparison with pollen reconstructions**

As pointed out above, this comparison is by no means a model validation. This should be made clear in the wording. As such, all sentences like "CCLM performs well" should be modified. The maps in Figure 5 are calculated as means with respect to which period?

In Figure 6, error bars are provided for the pollen data, but not for the simulation. I'm aware it is not easy to stablish them. However, such errors/uncertainties should not be neglected in the discussion of the results. The model has deficiencies that introduce systemic biases. But on top of then, there are non systematic biases introduced by unpredictable internal variability. This factor might lower or rise mean temperature in the simulation quite significantly, as pointed out by Gómez-Navarro et al. (2012) in a very similar scenario. Thus, this should be discussed at least qualitatively in this part of the text.

Many conclusions are drawn from Figure 6 regarding matchings of trends. I'm not sure at what extent such conclusions have any statistical significance, since in almost all cases the simulation lies within the uncertainty of the reconstruction. Having an almost perfect match between the reconstruction and the simulation is still perfectly possible within the range of uncertainty of the reconstructions.

Something I miss in this analysis here is the GCM simulations used to drive COSMO. I wonder how the ECHO-G and later ECHAM5 compares also with reconstructions. Is the RCM adding anything relevant to these simulations? If the answer is "certainly yes", then the use of the RCM is fully justified and the paper would gain interest. If the answer is "mostly no", it would be still interesting, since it would imply that the many GCM simulations available for the last millennia are still relevant at rather regional scales. I'm sure the PMIP community would be very interested in answering this question.

**Comments regarding the interpretation of the paleo records**

Generally it was difficult to follow the arguments in this section. It would significantly help to label the maps as Fig 6b, Fig 6c, etc. and use such labels extensively through the manuscript. In this regard, the discussion of the results starts with summer, whereas the first row shows winter. Small inconsitences like this, although non critical for the scientific message, have a dramatic impact in the reading pace.

The EOFs for MSLP are shown and used in the discussion. They are used to argue regarding NAO and SNAO, for instance. I'm not totally comfortable with that, since the NAO is defined as the leading pattern for a spatial window that is not that of the RCM. This explains in my opinion why the NAO pattern does not stand out as the leading mode in winter, and second mode in summer just "resembles" the SNAO pattern. I think a more orthodox approach would be to calculate the EOFs within the GCM, in a window that properly encompasses the North Atlantic. This is justified since the large scale circulation is fixed by the GCM, and thus the NAO simulated should be consistent with the climate variability within the RCM domain. Hence, such patterns could still be used to discuss about regional variability within the RCM domain.

Line 255 reads "In summer the first EOF shows that the model reproduces similar conditions in atmospheric circulation between the mid-Holocene and pre-industrial times". I do not understand how that conclusion is drawn from the map in Figure 8.

In page 8 the wording "observed" is used in various sentences, and it's not fully clear what is meant (most likely respect to the simulation, but it could also be the reconstruction). I think "simulated" is more appropriate and precise.

Some inferences about the "clearness" of the sky are made which are based in indirect evidence such as EOF analysis. I think it is not necessary to make such risky affirmations. We have direct information that can tell us exactly how cloudy the simulated climate was. After all, in the simulation we can check directly variables such as cloud cover, which give a direct measure of what is being argued. I would go for a direct measure whenever possible, as it is the case. Similarly, in the paragraph between lines 277 and 279 (and Fig. 11) the more pronounced positive phase of the NAO can be directly tested within the GCMs, rather than indirectly inferred through a map of temperatures.

Finally, I think there are more powerful statistical tools than the one used here to study the co-variability between temperature and MSLP. Canonical Correlation Analysis could be used to derive relations between the variability of MSLP and temperature, and it would produce a picture of such co-variability more robust that the one provided by maps in Figure 10, for instance. An example of the application of such a tool in a very similar context is Gomez-Navarro et al. (2015b)

**Comments regarding Figures**

Figure 1: The colour scale shows everything below 1000 meters as green. I think a palette with stronger contrast could be chosen.

Figure 2: The reference period should be stated in the caption. I think the limits of the palette can be adjusted to better span the range of temperatures.

Figure 3: The colour palette provides barely any contrast all. Everything is yellow in the maps.

Figures 4 and 5: Same comments as in former figures

Figure 6: Please label panels as 6a, 6b, etc. I do not think using colour in the caption is an orthodox approach. Note that the caption does not agree with the order of panels. First row does not show North, but it is the first column which does, etc.

Figure 7: I can barely see the numbers and labels in the figures in the right.

Figure 8: I think the label with the loading can be moved to inside the maps. This would allow to put the maps closer together, which would allow to make maps larger and more readable. The latter comment can be applied to almost all figures.

Figure 9: Please label panels to indicate which represent EOF1 etc. Where are the units? Either the EOF or the PC carries the units, in this case pressure. I guess they are included in the EOF patterns in Figure 8. If so, please label the palette accordingly.

**References**

Gómez-Navarro, J. J., Montávez, J. P., Jerez, S., Jiménez-Guerrero, P., Lorente-Plazas, R., González-Rouco, J. F., and Zorita, E.: A regional climate simulation over the Iberian Peninsula for the last millennium, Clim. Past, 7, 451–472, doi:10.5194/cp-7-451-2011, 2011.

Gómez-Navarro, J. J., Montávez, J. P., Jiménez-Guerrero, P., Jerez, S., Lorente-Plazas, R.,

González-Rouco, J. F., and Zorita, E.: Internal and external variability in regional simulations of the Iberian Peninsula climate over the last millennium, Clim. Past, 8, 25–36, doi:10.5194/cp-8-25-2012, 2012.

Gómez-Navarro, J. J., Montávez, J. P., Wagner, S., and Zorita, E.: A regional climate palaeosimulation for Europe in the period 1500–1990 – Part 1: Model validation, Clim. Past, 9, 1667–1682, doi:10.5194/cp-9-1667-2013, 2013.

Gómez-Navarro, J. J., Werner, J., Wagner, S., Luterbacher, J., and Zorita, E.: Establishing the skill of climate field reconstruction techniques for precipitation with pseudoproxy experiments, Climate Dynamics, 1–19, doi:10.1007/s00382-014-2388-x, 2015a.

Gómez-Navarro, J. J., Bothe, O., Wagner, S., Zorita, E., Werner, J. P., Luterbacher, J., Raible, C. C., and Montávez, J. P: A regional climate palaeosimulation for Europe in the period 1500–1990 – Part 2: Shortcomings and strengths of models and reconstructions, Clim. Past, 11, 1077-1095, doi:10.5194/cp-11-1077-2015, 2015b.

Haylock, M. R., N. Hofstra, A. M. G. Klein Tank, E. J. Klok, P. D. Jones, and M. New (2008), A European daily high-resolution gridded data set of surface temperature and precipitation for 1950–2006, J. Geophys. Res., 113, D20119, doi:10.1029/2008JD010201.

Jerez, S., J. P. Montavez, J. J. Gomez-Navarro, P. Jimenez-Guerrero, J. Jimenez, and J. F. Gonzalez-Rouco (2010), Temperature sensitivity to the land-surface model in MM5 climate simulations over the Iberian Peninsula, Meteorol. Z., 19(4), 363–374.

Jerez, S., J. P. Montavez, J. J. Gomez-Navarro, P. A. Jimenez, P. Jimenez-Guerrero, R. Lorente, and J. F. Gonzalez-Rouco (2012), The role of the land-surface model for climate change projections over the Iberian Peninsula, J. Geophys. Res., 117, D01109, doi:10.1029/2011JD016576.

Schimanke, S., Meier, H. E. M., Kjellström, E., Strandberg, G., and Hordoir, R.: The climate in the Baltic Sea region during the last millennium simulated with a regional climate model, Clim. Past, 8, 1419–1433, doi:10.5194/cp-8-1419-2012, 2012.

---

## Short Comment (SC1) · 12 Feb 2016

I would like to draw the authors' attention to a study (Strandberg et al., 2014) that simulates 6k BP and 0.2k BP climate in Europe with a RCM. Although it only consists of two time slices I think it qualifies as "high resolution simulations for different time slices of mid-to-late Holocene performed over Europe using a Regional Climate Model" (perhaps the first such simulations).

Furthermore, since Strandberg et al. (2014) use boundary data from ECHO-G and compare the results with the reconstructions from Mauri et al. (2014) it should be of

interest for Russo and Cubash.

I know that it is a characteristic of modellers to exaggerate the uncertainties in the models and downplay the uncertainties in the reconstructions, but I would be careful to "validate" the model against one set of reconstructions alone since they may be of equally good/poor quality as the model simulations.

When considering astronomical forcing alone (see Fig. 2 in Wagner et al., 2007), we would expect 6k to be warmer than 0.2k and the temperature difference to be largest in summer in northern Europe. This is the signature we see in the model simulations of Strandberg et al. (2014). The non-pollen proxy based palaeoclimatic data presented in Strandberg et al. (2014) and the pollen based reconstruction of Peyron et al. (2013) rather support the differences in summer temperatures simulated by Strandberg et al. (2014) than the reconstruction of Mauri et al. (2014), in particular for southern and eastern Europe.

References

Mauri, A., Davis, B., Collins, P., and Kaplan, J.: The influence of atmospheric circulation on the mid-Holocene climate of Europe: a data-model comparison, Climate of the Past, 10, 1925–1938, 2014.

Peyron, O., Magny, M., Goring, S., Joannin, S., de Beaulieu, J.-L., Brugiapaglia, E., Sadori, L., Garfi, G., Kouli, K., Ioakim, C., and Combourieu-Nebout, N.: Contrasting patterns of climatic changes during the Holocene across the Italian Peninsula recon-structed from pollen data, Clim. Past, 9, 1233–1252, doi:10.5194/cp-9-1233-2013, 2013.

Wagner, S., Widmann, M., Jones, J., Haberzettl, T., Lücke, A., Mayr, C., Ohlendorf, C., Schäbitz, F., and Zolitschka, B.: Transient simulations, empirical reconstructions and forcing mechanism for the Mid-holocene hydrological climate in Southern Patagonia, Clim. Dynam., 29, 333–355, 2007.

---

## Referee Comment (RC2) · Anonymous Referee #2 · 24 Feb 2016

This study presents novel paleoclimate modelling results obtained with a high-resolution regional climate model (COSMO-CLM) that is nested in an atmospheric general circulation model (ECHAM5). These modelling results span the past 6000 years and are compared to pollen-based temperature reconstructions. The topic fits very well within the scope of Climate of the Past.

To my knowledge, this is the first study on the mid-to-late Holocene that has been performed with a regional climate model. Many modelling studies with a focus on this same time period have been published with GCMs or EMICs, and it will be interesting

to see how the results of this regional climate model compare to these studies and to investigate whether the higher spatial resolution produce a better match to proxy-based reconstructions.

In my opinion, the innovative results do merit publication, but the presentation of the results and the discussion should be considerably improved. As detailed below, this manuscript requires major revisions before it can be accepted.

Main comments

- The grammar and spelling can be much improved. There are many long sentences that are hard to read. I have indicated a few below. I strongly suggest to have the text thoroughly checked by a native English speaker.

- I propose to compare the results of COSMO-CLM to the results of ECHAM5. The latter results have already a relatively high spatial resolution (T106 or 1.125x1.25 degr) compared to previous GCM studies. This resolution is actually close the resolution of the reconstructions (1x1 degr). In the manuscript, the authors have regridded (up scaled) their regional climate model results from 0.44x0.44 degree resolution to 1x1 degree to make the comparison in Fig 5. It would be interesting to see to what extent the COSMO-CLM produces a better match. Is it, from a paleoclimate perspective, worthwhile to make the considerable effort to nest the regional model in the high-resolution GCM results? Or do both models produces very similar results? In my view, addressing these questions would strengthen the paper. To make room for such a comparison, Figures 2, 3 and 4 could be moved to the supplementary information, as these figures do not directly concern the core topic of this study (mid-to-late Holocene temperatures and atmospheric dynamics).

- The left column of Fig 5 presents maps of the winter and summer temperature anomalies (model minus reconstructions), "averaged over all the mid-to-late Holocene time slices". It is not clear to me what the authors have actually done here. Have they first averaged the maps of the different time-slices for the model and the data, and then

calculated the model-data anomaly? Or have they calculated the trend between 6000 and 200 BP in both model and data, and then made a map of the difference between the two methods? The caption suggests that they have applied the first method, but in my view this would only be meaningful if the anomalies are more less constant through time, which is clearly not the case (see Figure 6). Since the trends from 6000 to 200 BP seem approximately linear in both model and data, it would make more sense to compare maps of these trends or to show maps for different time slices. Figure 11 actually shows linear trend maps for both the model and the reconstructions, but only for DJF. It is unclear to me how to relate Figure 11 to Figure 6. Figure 11 seems to indicate a pollen-based linear warming in Southern Europe of mostly less than 0.4°C, while Figure 6 shows a warming trend for the pollen-based reconstructions of 1°C for Southern Europe. In addition, the pollen-based cooling trend in Figure 6 of more than 2°C does not match Figure 11 which shows a much smaller cooling trend. Is there an inconsistency between Figure 6 and 11, or have I missed something? Please clarify.

- The right column of Fig. 5 shows the uncertainties in the pollen-based temperature reconstruction. How were these maps constructed? According to Fig. 6, these uncertainties are not constant through time, so simply averaging the errors for the different time slices is not informative here either. Please clarify.

- For the summer in Southern Europe, the model and the reconstructions show opposite trends: cooling in the model and warming in the reconstructions. The authors provide an explanation for this model-data mismatch that is based on the warm bias of the model in S Europe due to the underestimation of evaporation in summer. However, the mismatch may also be explained by uncertainty in the pollen-based reconstructions in S Europe. Paleoclimate reconstructions based on pollen rely on the assumption that changes in the vegetation were driven by the parameter to be reconstructed (i.e. summer temperature). In the Mediterranean region, vegetation distribution is mainly limited by effective precipitation, rather than by summer temperature (e.g. Osborne et al. 2000). It would therefore be good to discuss the associated uncertainties in the

methodology of the pollen-based reconstructions and to mention Holocene temperature reconstructions that are based on other proxies. For instance, summer temperature reconstructions from the S Europe domain based on Chironomids, show a clear Holocene cooling (Heiri et al. 2015; Toth et al. 2015) that actually support the presented modelling results. In addition, Holocene SST reconstructions from the Mediterranean Sea show a similar cooling trend (e.g. Marchal et al. 2002). The discussion section should be extended accordingly.

- In the discussion, the results should also be compared to other modelling studies that focus on the mid-to-late Holocene climate. Do the new results presented here confirm earlier findings? How do the seasonal trends and 6k-0k anomalies compare to that of other models (e.g., PMIP3)? What do other Holocene modelling studies say about changes in atmospheric circulation over Europe and the North Atlantic basin?

- Conclusions: The conclusions should be made less descriptive / more quantitative. The paragraph starting on line 296 does not contain conclusions and can be removed. Please explain on Line 310 what atmospheric circulation configuration is meant here.

Minor comments

Line 26: I suggest providing a more accurate definition of climate models

Line 34: "orbital parameters". I propose to use astronomical parameters instead, since obliquity is not a parameter of the Earth's orbit.

Line 37: Please rephrase this sentence, as it is not easy to read

Line 43: "solar forcing". Usually, "solar forcing" is used to describe changes in solar activity as opposed to astronomical forcing that reflects changes in insolation due to changes astronomical parameters. To avoid confusion, I suggest using astronomical forcing here.

Line 46: In my view, this sentence does not introduce the reader to the paragraph, so I propose using a different topic sentence.

Line 57: It is not clear to me what is meant by "hampered climate anomalies"

Line 60: typo, atmopshere

Line 60: "not being able to reproduce correctly the reconstructed data over the entire region". Please clarify. Was the model too cold or too warm? What was the bias?

Line 63: Please rephrase the sentence starting at this line.

Line 72: " In many cases" What cases, please elaborate.

The objectives of the paper should be explained more clearly. On page 3, two objectives are provided. The first objective is to "obtain a better interpretation of the new pollen database..." Why better? What problems have been encountered in the interpretation?

Line 105. This first sentence of Section 2 does not provide information on the applied methods. I suggest moving this sentence to Section 1 and to replace it with a topic sentence that introduces the methodology used.

Line 128: Berger and Loutre (2002) do not calculate astronomical parameters and is not the appropriate reference here. In their figure they show the values of such parameters, but these are based on Berger (1978), so I suggest to use this reference here.

Line 133: "only the latest ones". I am not sure what is referred to here. The latter effects?

Line 175: "while coloured are the anomalies". Please rephrase and clarify.

Line 194: I propose to use "anomalously warm conditions" here.

Line 195: " as a consequence of a wrong conversion of energy towards latent heat." This suggests to me that there is an error in the model code that described this conversion. Is that the case, or is the conversion in principle correct and does the model

have a bias in S Europe?

Line 205: typo "teperature"

Line 213: I suggest replacing "Pollen" by "pollen-based temperatures"

Line 214: Please rephrase, as this sentence is confusing. The sentence suggests that Section 3.2 will discuss the results after the validation against Mauri et al's data has taken place, while in fact the next paragraph deals with this validation. Besides, I would prefer using evaluation instead of validation here.

Line 216: I suggest referring to Figure 1, as this figure shows the boundaries of the two domains.

Line 220: I assume that the model results are up-scaled and regridded on a 1x1 degree grid before the anomalies are calculated. Please clarify this here

Line 231: I propose replacing "Paleo-Results" by Paleoclimate results.

Line 237: Figure 7 shows the insolation changes over the mid-to-late Holocene. This is the main radiative forcing for the model experiments, so I suggest to show it already in Section 2 where the experimental design is discussed.

Line 250: what other cases?

Caption Figures 8 and 9: The captions are not consistent with the figures. Are summer results plotted at the upper or the lower row?

Figure 8: How is Figure 8 constructed? On what timeslice is it based, or is it based on results from several time slices?

Line 268: "scarce ability" Replace by poor ability?

Line 276: "showing instead low correlation over the South". This is a confusing statement. Figure 10 shows that over most of the Mediterranean, the correlation in winter is strongly negative for the 1st EOF and strongly positive for summer.

Line 284: "the model simulates a lower weight of the NAO (∼40%) for mid-to-late Holocene in comparison to present-days conditions (∼55%)". How can we reconcile this with the notion of a "more pronounced positive phase of the NAO during the mid-Holocene" as stated on line 277?

Additional references Heiri, O. et al. The Holocene 25, 137-149 (2015). Marchal, O., et al. Quat. Sci. Rev. 21, 455-483 (2002). Osborne, C.P., et al. Glob. Change Biol. 6, 445-458 (2000). Toth, M., et al. The Holocene 25, 569-582 (2015).

---

## Author Comment (AC1) · 24 Mar 2016

Emmanuele Russo (emmanuele.russo@met.fu-berlin.de)

11

[Figure]

**Mid-to-late Holocene Temperature Evolution and Atmospheric Dynamics over Europe in Regional Model Simulations**

March 24, 2016

Reply to
**G. Strandberg**
*Mid-to-late Holocene Temperature Evolution and Atmospheric Dynamics over Europe in Regional Model Simulations by Russo, Emmanuele; Cubasch, Ulrich cp-2016-10*

Dear G. Strandberg,
thank you very much for your interest in our paper.
We are very thankful to receive the comments of an expert in the field of paleoclimate and regional climate modelling. Below we try to answer to your remarks, and detail how we dealt with your concerns reported in *italic*.

Thank you.

[Figure]

**Main Comments**:

1. *I would like to draw the authors' attention to a study (Strandberg et al., 2014) that simulates 6k BP and 0.2k BP climate in Europe with a RCM. Although it only consists of two time slices I think it qualifies as "high resolution simulations for different time slices of mid-to-late Holocene performed over Europe using a Regional Climate Model" (perhaps the first such simulations). Furthermore, since Strandberg et al. (2014) use boundary data from ECHO-G and compare the results with the reconstructions from Mauri et al. (2014) it should be of interest for Russo and Cubash.*

    Thanks for suggesting the work of Strandberg et al. 2014. It is interesting and gave us the opportunity to consider new proxy-reconstructions for our discussion. Additionally, the paper structure, and in particular the paragraph on the comparison against other PMIP results, makes it a good reference to consider in order to further improve the first draft of our manuscript.

2. *I know that it is a characteristic of modellers to exaggerate the uncertainties in the models and downplay the uncertainties in the reconstructions, but I would be careful to "validate" the model against one set of reconstructions alone since they may be of equally good/poor quality as the model simulations. When considering astronomical forcing alone (see Fig. 2 in Wagner et al., 2007), we would expect 6k to be warmer than 0.2k and the temperature difference to be largest in summer in northern Europe. This is the signature we see in the model simulations of Strandberg et al. (2014). The non-pollen proxy based palaeoclimatic data presented in Strandberg et al. (2014) and the pollen based reconstruction of Peyron et al. (2013) rather support the differences in summer temperatures simulated by Strandberg et al. (2014) than the reconstruction of Mauri et al. (2014), in particular for southern and eastern Europe.*

    The choice of the dataset of Mauri et al. 2014 has been done for many reasons. First of all it allows to perform a comparison with model results over most of the simulations domain, considering different variables (even if we only focus on temperature in our discussion). Then, it covers exactly the same time-slices of the model simulations. No other dataset has this temporal and spatial coverage. Additionally, the robustness of the data has been already tested, in Mauri et al. 2014, against other proxies (including chironomids, $\delta^{18}$O from speleothems and lake ostracods, bog-oaks, glacio-lacustrine sediments, wood anatomy and other pollen reconstructions based on different reconstruction methods). For such reasons we think that the reconstructions of Mauri et al. 2014 are a reliable source for the comparison of model results.

Nevertheless, considering other proxies for our analyses could be an important point. Preliminary qualitative analysis against other reconstructions, such as the ones of Hairi et al. 2014 and Peyron et al. 2013, confirm that the data used in our discussion present a similar behaviour. In our former analysis this was not evident since we considered regional means for the investigation of mid-to-late Holocene temperature evolution. For this reason, we now performed additional analysis accordingly to this point.

With kind regards on behalf of the all authors,
Emmanuele Russo

---

## Author Comment (AC2) · 11 Apr 2016

epsfig color url

Reply to
**2nd Reviewer**
*Mid-to-late Holocene Temperature Evolution and Atmospheric Dynamics over Europe in Regional Model Simulations by Russo, Emmanuele; Cubasch, Ulrich cp-2016-10*

Dear reviewer,

Thank you very much for your effort in reviewing our paper.

Below we go point by point through your technical corrections, detailing how we dealt with your concerns reported in **b**olt.

Thank you.

1. **Main Comments:**
   **The grammar and spelling can be much improved. There are many long sentences that are hard to read. I have indicated a few below. I strongly suggest to have the text thoroughly checked by a native English speaker.**

   We propose to improve the structure and the grammar of the paper in order to make it more easily readable. We also aim at shortening long sentences and express complex periods in a more concise and robust way.

   **I propose to compare the results of COSMO-CLM to the results of ECHAM5. The latter results have already a relatively high spatial resolution (T106 or 1.125x1.25 degr) compared to previous GCM studies. This resolution is actually close the resolution of the reconstructions (1x1 degr). In the manuscript, the authors have regridded (up scaled) their regional climate model results from 0.44x0.44 degree resolution to 1x1 de- gree to make the comparison in Fig 5. It would be interesting to see to what extent the COSMO-CLM produces a better match. Is it, from a paleoclimate perspective, worth- while to make the considerable effort to nest the**

**regional model in the high-resolution GCM results? Or do both models produces very similar results? In my view, address- ing these questions would strengthen the paper. To make room for such a comparison, Figures 2, 3 and 4 could be moved to the supplementary information, as these figures do not directly concern the core topic of this study (mid-to-late Holocene temperatures and atmospheric dynamics).**

According to the IPCC (2007) report: "Paleoclimate data are key to evaluating the ability of climate models to simulate realistic climate change". In particular, since the details added by high resolution models can help in the interpretation of proxy data that are often influenced by processes taking place on smaller scales than the ones resolved in coarser models, they are considered a particularly suitable tool for paleoclimate studies.

Within this context, in our discussion we try to highlight the importance of using high resolution models, and in particular Regional Climate Models, for the simulation of past climate change. Aiming at investigating the value added by highly resolved simulations for the comparison of near surface temperatures against proxy-reconstructions, we follow a two steps approach:

(a) Firstly, we conduct a qualitative analysis of the simulations performed with three models at different resolutions in order to detect visible differences in the reproduced signals.

(b) Secondly, we employ a quantitative approach in order to estimate the skills of the RCM, in comparison to the driving GCM, in reproducing the same changes in temperature during mid-to-late Holocene as derived from proxy-reconstructions.

As a benchmark for such comparison we use the pollen-based temperature re-constructions of ? In this way we aim at establishing whether the representation of smaller scale processes and improved orographic features of the region of study, could lead to results that are in better agreement with the mentioned proxy-reconstructions.

In Fig. 1 we present the anomalies of summer and winter seasonal mean temperatures between 6000BP and the Pre-industrial period, as reproduced by the different models. From these maps we first notice as, in both the seasons, a similar signal of climate change is present for all the simulations. This is expected, beeing, in every case, the data constrained by the coarser resoluted models. Nevertheless, while the higher resoluted simulations allow to catch a warmer bias over Northern Europe in winter, also present in the proxy data, the ECHO-G does not show such behaviour. Additionally, the land-sea area in the ECHO-G is considerably different than the ones of the other models. Regions such as Southern Spain and the Black sea area, Italy and Scandinavia are partly or completely masked-out in this case.

Consequently, we reasonably suggest to focus further analyses on the comparison between the ECHAM5 and the CCLM results. In both seasons additional details are easily detectable in the CCLM pattern. The coastline is also better reproduced in this case, resulting in more suitable informations for possible comparison with proxy-data. Nonetheless, the CCLM shows better defined patterns as a consequence of higher resolution, being able to discriminate higher spatial variability.

In the successive step, we try to quantify how better the CCLM reproduces the reconstructed temperatures in comparison to the ECHAM5. Under the mentioned considerations, we use a similar approach to the one employed by Zhang et al. (2010) and based on the work of Goosse et al. (2006). After upscaling the RCMs results and interpolating the ECHAM5 ones on the reconstructions grid, we introduce a Cost Function defined as:

$$CF_{mod}^k = \sqrt{\frac{1}{n}\sum_{i=1}^{n}\omega_i{}^k(T_{rec,i}^k - T_{mod,i}^k)^2} \qquad (1)$$

where $CF_{mod}^k$ is the value of the cost function for each considered time slice *k* of mid-to-late Holocene, and each model *mod* . The parameter *n* is the number of the reconstructions grid boxes, $T_{rec,i}^k$ the reconstructions temperature at every location *i*, while $T_{mod,i}^k$ is the correspondant temperature of the model simulation. The parameter $w_i^k$ is instead introduced for considering the uncertainties of the reconstructions at every location and time period. Its value is given by:

$$\omega_i^k = \frac{1}{(\sigma_i^k)^2 + 1} \qquad (2)$$

In this way reconstructions with higher uncertainties will contribute less in the calculation of the Cost Function. We neglicted models uncertainties since they are considerably small ($\sim 0.01^o C$) in comparison to the reconstructions ones. The values of the CF for the two models are provided in Tab.1 and in Tab.2.

As we can notice, even if not particularly large differences are present, the Cost Function computed for the CCLM is in almost all the cases lower than the ECHAM5 one. In particular the CCLM results are, in some cases, closer by almost 10% to the reconstructions. It is important to mention that the scale considered in our analysis is closer to the resolution of the ECHAM5 than the one of the CCLM. As suggested by Di Luca et al. (2015), given that the main difference between the GCM and the RCM is related with their horizontal resolution, it seems natural that the results depend on spatial scale of the analysis. Additionally, is key to state that the evinced results are relative to this case of study and other comparisons should be performed, considering different couples of RCM-GCM,

in order to derive more robust conclusions on the suitability of higher resoluted models for the comparison against proxy-reconstructions.

Nonetheless, the motivation behind producing higher resolution climate simulations is not only related to scientific arguments of the type described above. From a different perspective, such results, due to the greater level of detail, could be preferable for applications in studies in which human adaptation or environmental response to past climatic changes would be investigated. The need for climate information at very fine scales, for application such as archaeology or vegetation reconstructions, hence constitutes a strong incentive to perform higher-resolution climate simulations (Di Luca et al. (2015), Rummukainen (2016)).

In conclusion, the evinced results and the proposed discussion, give us concrete motivations for the choice of conducting RCM simulations for this particular case of study. Nevertheless, we aim at keeping Fig.2, Fig.3 and Fig.4 of the discussion paper within the revised version of the manuscript, as representing a satisfactory test for the reliability of the chosen model setup, they could be suitable for other studies conducting paleoclimate simulations for the region.

In the new version of the manuscript we will add a section based on the presented analyses accompanied by detailed and pertinent discussion.

**The left column of Fig 5 presents maps of the winter and summer temperature anomalies (model minus reconstructions), "averaged over all the mid-to-late Holocene time slices". It is not clear to me what the authors have actually done here. Have they first averaged the maps of the different time-slices for the model and the data, and then calculated the model-data anomaly? Or have they calculated the trend between 6000 and 200 BP in both model and data, and then made a map of the difference between the two methods? The caption suggests that they have applied the first method, but in my view this would only be meaningful if the anomalies are more less constant through time, which is clearly not the case (see**

Interactive
comment

**Figure 6). Since the trends from 6000 to 200 BP seem approximately linear in both model and data, it would make more sense to compare maps of these trends or to show maps for different time slices. Figure 11 actually shows linear trend maps for both the model and the reconstructions, but only for DJF. It is unclear to me how to relate Figure 11 to Figure 6. Figure 11 seems to indicate a pollen-based linear warming in Southern Europe of mostly less than 0.4Ůę C, while Figure 6 shows a warming trend for the pollen-based reconstructions of 1Ůę C for Southern Europe. In addition, the pollen-based cooling trend in Figure 6 of more than 2Ůę C does not match Figure 11 which shows a much smaller cooling trend. Is there an inconsistency between Figure 6 and 11, or have I missed something? Please clarify.**

In the previous analysis, Fig.5 was obtained by simply averaging the anomalies over all the time-slices. The same procedure was also applied in order to obtain a map of the average uncertainties. Following the considerations of the referee, we realized that such approach was not totally correct and we re-performed our analysis consequently. In the new case, as shown in Fig.2 and Fig.3, we compute the seasonal anomalies of 2 meters temperature between the CCLM and the pollen-based reconstructions for every single period of time. We additionally provide, together with the anomalies, the respective pollen-reconstructions uncertainties. This choice is reasonable since the uncertainties maps could result useful in the interpretation of the mismatches arising between the two datasets.

Additionally, we are now considering a new approach for the investigation of seasonal trends. We recomputed figure 6 taking into consideration, this time, the uncertainties in both the datasets (for more specifications please refer to the first referee response). Here the new plots are similar to Fig. 11 of the discussion paper, showing this time both winter and summer trends. Only the area where

the trends are significant, according to a F-test at a significance level of 0.1, are shown. Additionally, such trends are calculated by mean of a weighted least squares method, allowing to take into consideration, as said, the uncertainties of the two datasets. Since the changes in both the datasets are not homogeneous over the region, we think that these maps should be more appropriate than the previous ones based on regional means. We want to highlight, relatively to the referee's comment, that the new maps do not show values of changes in temperature. Rather they show the slope of the trend associated to every grid box.

**The right column of Fig. 5 shows the uncertainties in the pollen-based temperature reconstruction. How were these maps constructed? According to Fig. 6, these uncer- tainties are not constant through time, so simply averaging the errors for the different time slices is not informative here either. Please clarify.**

Please refer to the previous point.

**For the summer in Southern Europe, the model and the reconstructions show opposite trends: cooling in the model and warming in the reconstructions. The authors provide an explanation for this model-data mismatch that is based on the warm bias of the model in S Europe due to the underestimation of evaporation in summer. However, the mismatch may also be explained by uncertainty in the pollen-based reconstructions in S Europe. Paleoclimate reconstructions based on pollen rely on the assumption that changes in the vegetation were driven by the parameter to be reconstructed (i.e.summer temperature). In the Mediterranean region, vegetation distribution is mainly limited by effective precipitation, rather than by summer temperature (e.g. Osborne et al. 2000). It would therefore be good to discuss the associated uncertainties in the methodology of the pollen-based reconstructions and to mention Holocene temperature reconstructions that are based on other proxies. For instance, summer**

**temperature reconstructions from the S Europe domain based on Chirono-
mids, show a clear Holocene cooling (Heiri et al. 2015; Toth et al. 2015) that
actually support the presented modelling results. In addition, Holocene
SST reconstructions from the Mediterranean Sea show a similar cooling
trend (e.g. Marchal et al. 2002). The discussion section should be extended
accordingly.**

The choice of the dataset of Mauri et al. 2015 has been done for several rea-
sons. First of all, it allows to perform a comparison against model results over
most of the simulations domain, considering different variables (even if we only
focus on temperature in our discussion). Then, it covers exactly the same time-
slices of our model simulations. No other dataset has this temporal and spatial
coverage at such high spatial resolution. Additionally, the robustness of the data
has been thoroughly tested, in Mauri et al. 2015, against other proxies (includ-
ing chironomids, $\delta 18$ O from speleothems and lake ostracods, bog-oaks, glacio-
lacustrine sediments, wood anatomy and other pollen reconstructions based on
different reconstruction methods) leading to satisfactory results. Nonetheless,
similar pollen-based climatic reconstructions have been extensively employed
in other data-model comparisons, and, most recently, for the evaluation of the
PMIP3/CMIP5 climate models included in the last IPCC report (Stocker et al.
2013, Harrison et al. 2015).

As the referee mentioned, different studies already criticized the use of pollen-
based data for reconstruction of temperature over the Mediterranean region,
claiming that the vegetation distribution is mainly limited by effective precipita-
tion, rather than by summer temperature (e.g. Osborn et al. (2000); Renssen et
al. (2009)). In response to such critiques we want to refer to a detailed comment
provided by Basil Davis, and attached to this discussion.

According to the aforementioned reasons, and additionally supported by the explanations given by B. Davis in his comments, we think that the employed pollen-based reconstructions can be considered a very reliable source for the main goals of our paper.

Nevertheless, in accordance to the referee's comments, in the new version of the manuscript we will provide further discussion on the uncertainties in the methodology of the pollen-based reconstructions and specify more details on the reliability tests conducted by Mauri et al. 2015.

Since the comparison against independent and different proxies has already been performed by Mauri et al. 2015, we feel that such analysis could be omitted from our manuscript.

Additionally, the previous analyses of mid-to-late Holocene temperature evolution were misleading. In fact, simply considering regional means, they did not allow to have a proper overview of the trends at different locations, possibly resulting in a mismatch in the comparison against other proxies. The new maps presented in Fig. 4 show now a more heterogeneous behaviour, and are in better agreement with other independent reconstructions such as the one of Heiri et al. (2015), mentioned by the referee, for which summer temperatures over the Alpine region were characterized by a decreasing trend during mid-to-late Holocene.

**In the discussion, the results should also be compared to other modelling studies that focus on the mid-to-late Holocene climate. Do the new results presented here confirm earlier findings? How do the seasonal trends and 6k-0k anomalies compare to that of other models (e.g., PMIP3)? What do other Holocene modelling studies say about changes in atmospheric circulation over Europe and the North Atlantic basin?**

We agree. We propose to present, in the revised version of the manuscript, a section in which our results are compared against other studies. In particular, we will focus on the anomalies between 6000BP and the pre-industrial period, performing a direct comparison against the outcomes of 12 models from the PMIP3 experiment. We will compute the regional means for two regions over Northern and Southern Europe for al the datasets. We will include such values in two tables, attached to this discussion, that we aim to provide as supplementary material in the revised manuscript. The main features arising from such analysis are, a common positive bias over Southern Europe in summer, and the failure to properly represent winter anomalies in both the regions. We aim to implement and develop our discussion accordingly.

**Conclusions: The conclusions should be made less descriptive / more quantitative. The paragraph starting on line 296 does not contain conclusions and can be removed. Please explain on Line 310 what atmospheric circulation configuration is meant here.**

We agree. We propose to make our conclusion more quantitative. According to the new analysis presented here and as a response to the 1st referee, we aim at extending our discussion and develop our conclusions in a more concise and robust way.

2. **Minor Comments:**

**Line 26: I suggest providing a more accurate definition of climate models**

We agree. We will develop a more detailed description of the climate models. Nevertheless, we will try to be as exhaustive as possible, referring to their thecnical manuals for further details that would not be inherent in the discussion.

**Line 34: "orbital parameters". I propose to use astronomical parameters instead, since obliquity is not a parameter of the Earth's orbit.**

[Figure]

We agree and will change the term "Orbital" in "Astronomical".

**Line 37: Please rephrase this sentence, as it is not easy to read**

We propose to rephrase the highlighted sentence accordingly to the referee's comment

**Line 43: "solar forcing". Usually, "solar forcing" is used to describe changes in solar activity as opposed to astronomical forcing that reflects changes in insolation due to changes astronomical parameters. To avoid confusion, I suggest using astronomical forcing here.**

Aware of the mistake, we will correct the term "solar forcing" with "astronomical forcing".

**Line 46: In my view, this sentence does not introduce the reader to the paragraph, so I propose using a different topic sentence.**

We agree. We propose to modify this part in order to better connect it with the following text.

**Line 57: It is not clear to me what is meant by "hampered climate anomalies"**

We reformulated this sentence. With "hampered anomalies" we wanted to indicate that, the improvement in the reproduction of soil water storage and heat fluxes by climate models, as suggested byStarz et al. (2013), could lead to a reduction of the biases arising from the comparison with observations. We agree with the referee that the former expression was somehow misleading and we will reformulate it in a clearer way.

**Line 60: typo, atmopshere**

Corrected in atmosphere.

**Line 60: "not being able to reproduce correctly the reconstructed data over the entire region". Please clarify. Was the model too cold or too warm? What was the bias?**

We will extend the previous period with further details, referring to the results of Fischer & Jungclaus (2011). In particular, their results presented only a weak shift to a positive phase of the NAO at mid-Holocene in Winter, resulting in colder conditions over Northern Europe and warmer over Southern Europe with respect to the values of reconstructions. In summer, again, the signal seemed to be mainly driven by changes in insolation, resulting in homogenously warmer conditions at 6000 BP.

**Line 63: Please rephrase the sentence starting at this line.**

We reformulated the sentence accordingly to the referee's comment.

**Line 72: " In many cases" What cases, please elaborate. The objectives of the paper should be explained more clearly. On page 3, two objec- tives are provided. The first objective is to "obtain a better interpretation of the new pollen database..." Why better? What problems have been encountered in the inter- pretation?**

We agree. The objectives of the paper should be better explained. We try to do so also based on the referee comments and the additional analyses provided in this revision. Mauri et al. (2015) presented a possible interpretation of the anomalies evinced from their reconstructions between 6000BP and the pre-industrial

period, mainly based on changes in atmospheric circulations. Supported by previous findings, we use our results and the entire mid-to-late Holocene time slices reconstructions of **?**, in order to arise plausible interpretations. In particular, while for winter we agree with their interpretation of a more pronounced positive phase of the NAO at mid-Holocene, our findings support different interpretations for summer temperature behaviour. We will try to improve our discussion accordingly.

**Line 105. This first sentence of Section 2 does not provide information on the applied methods. I suggest moving this sentence to Section 1 and to replace it with a topic sentence that introduces the methodology used.**

We agree. We propose to move this sentence to section 1 and to modify it in order to better introduce the reader to the employed methodology.

**Line 128: Berger and Loutre (2002) do not calculate astronomical parameters and is not the appropriate reference here. In their figure they show the values of such parameters, but these are based on Berger (1978), so I suggest to use this reference here.**

We will change the reference accordingly to the referee's comment.

**Line 133: "only the latest ones". I am not sure what is referred to here. The latter effects?**

In the previous sentence we referred to the changes in insolation due to astronomical forcings. We will try to express the period in a clearer way.

**Line 175: "while coloured are the anomalies". Please rephrase and clarify.**

We agree. We wanted to indicate that biases between the two datasets are represented by a chromoghraphic gradient, from blue (when negative), to red

(when positive). We reformulate the sentence accordingly.

**Line 194: I propose to use "anomalously warm conditions" here.**

We agree. We propose to correct the sentence accordingly to the referee's suggestion.

**Line 195: " as a consequence of a wrong conversion of energy towards latent heat." This suggests to me that there is an error in the model code that described this con- version. Is that the case, or is the conversion in principle correct and does the model have a bias in S Europe?**

Being our results consistent with the ones of previous studies investigating present days conditions (Kotlarski et al. 2014; Jacob et al. 2014; Hollweg et al. 2008), we suggest that the model code describing soil-atmosphere interactions should be reliable. Some biases are present, particularly over Southern Europe, most presumably due to difficulties in properly reproducing soil water storage capacity for this complex orographic area.

**Line 205: typo "teperature"**

corrected in Temperature

**Line 213: I suggest replacing "Pollen" by "pollen-based temperatures"**

We agree. We will replace "Pollen" with "pollen-based temperatures" accordingly to the referee's comment.

**Line 214: Please rephrase, as this sentence is confusing. The sentence suggests that Section 3.2 will discuss the results after the validation against Mauri et al's data has taken place, while in fact the next paragraph deals with this validation. Besides, I would prefer using evaluation instead**

**of validation here.**

We agree. We propose to use "Comparison" as a better suitable word in this case.

**Line 216: I suggest referring to Figure 1, as this figure shows the boundaries of the two domains.**

We agree. Nevertheless we propose to modify Figure 1 accordingly to the new analysis we presented.

**Line 220: I assume that the model results are up-scaled and regridded on a 1x1 degree grid before the anomalies are calculated. Please clarify this here**

The model results are up-scaled to the observations'grid as hypothesized by the referee. We already provided further details within these comment and will do the same within the revised manuscript when necessary.

**Line 231: I propose replacing "Paleo-Results" by Paleoclimate results.**

We agree. We will modify the sentence accordingly to the referee's suggestion.

**Line 237: Figure 7 shows the insolation changes over the mid-to-late Holocene. This is the main radiative forcing for the model experiments, so I suggest to show it already in Section 2 where the experimental design is discussed.**

We agree. We will move the mentioned picture to the second chapter accordingly to the editor's suggestion.

**Line 250: what other cases?**

Realizing that the previous sentence was misleading, we propose to replace it with "other regions".

**Caption Figures 8 and 9: The captions are not consistent with the figures. Are summer results plotted at the upper or the lower row?**

As the referee noticed, in Figures 8 and 9 the upper row represents winter while the lower summer. The captions, instead, were previously inverted. We propose to change the caption accordingly.

**Figure 8: How is Figure 8 constructed? On what timeslice is it based, or is it based on results from several time slices?**

Figure 8 represent the first two EOFs of winter and summer seasonal mean of mean sea level pressure, standardized to the pre-industrial period. We propose to add more details in the caption of this figure, being the previous one not very precise.

**Line 268: "scarce ability" Replace by poor ability?**

We modified "scarce ability" with "poor ability" following the referee's suggestion.

**Line 276: "showing instead low correlation over the South". This is a confusing state- ment. Figure 10 shows that over most of the Mediterranean, the correlation in winter is strongly negative for the 1st EOF and strongly positive for summer.**

We realized that the previous period was not really clear. In fact, with the term SNAO we wanted to refer here to the Summer NAO. The conclusions we were

proposing, were definitely the same as the ones suggested by the referee. For this reason we propose to better express this period in order make it more easily readable.

**Line 284: "the model simulates a lower weight of the NAO ($\sim 40\%$) for mid-to-late Holocene in comparison to present-days conditions ($\sim 55\%$)". How can we reconcile this with the notion of a "more pronounced positive phase of the NAO during the mid- Holocene" as stated on line 277?**

We agree. Nevertheless, we want to highlight the fact that, according to different comments of both the authors, we deeply modified the previous analysis of atmospheric circulation. Based on the new analyses, we suggest that the previous sentence on line 284 needs corrections.

With kind regards on behalf of the all authors,
Emmanuele Russo

**References**

Solomon, S., D. Qin, M. Manning, Z. Chen, M. Marquis, K.B. Averyt, M.Tignor and H.L. Miller (eds.), 2007. *IPCC, 2007: Climate Change 2007: The Physical Science Basis. Contribution of Working Group I to the Fourth Assessment Report of the Intergovernmental Panel on Climate Change*. Cambridge University Press, Cambridge, United Kingdom and New York, NY, US.

Zhang, Q., Sundqvuist, H.S., Moberg, A., Körnich, H., Nilsson, J., Holmgren, K., 2010. *Climate change between the mid and the late Holocene in northern high latitudes - Part 2: Model-data comparisons*, Climate of the Past, 6: 609-626.

Goosse, H., Renssen, H., Timmermann, A., Bradley, R.S., Mann, M.E., 2006. *Using Paleocli-*
*mate proxy-data to select optimal realisations in an ensemble of simulations of the climate of the past millennium*, Climate Dynamics, 27: 165-184.

Tselioudis, G., Douvis, C., Zerefos, C., 2012. *Does dynamical downscaling introduce novel information in climate model simulations of precipitation change over a complex topography region?*, Int. journ. of Climat., 32: 1572-1578.

Rummukainen, M., 2016. *Added value in regional climate modeling*, WIREs Climate Change, 7: 145-149.

Di Luca, A., de Elia, R., Laprise, R., 2015. *Challenges in the Quest for Added Value of Regional Climate Dynamical Downscaling*, Curr. Clim. Change Rep., 1: 10-21.

Mauri, A., Davis, B., Collins, P., and Kaplan, J., 2015. *The Climate of Europe during the Holocene: a gridded Pollen-based Reconstruction and its multi-proxy Evaluation*, Quaternary Science Reviews, 112:109-127.

Peyron, O., Magny, M., Goring, S. Joannin, S., de Balieu, J.-L., Brugiapaglia, E., Sadori, L., Garfi, G., Kouli, K. Ioakim, C., Combourieu-Nebout, N., 2013. *Contrasting Pattern of Climatic Changes during the Holocene across the Italian Peninsula reconstructed from pollen data*, Climate of the Past, 9: 1233-1252.

Heiri, O., Lotter, F.A., Hausmann, S., Kienast, F., 2013. *A chironomid-based Holocene summer air temperature reconstruction from the Swiss Alps*, The Holocene, 13,4:477-484.

Starz, M. Lohmann, G. Knorr, G, 2013. *Dynamic soil feedbacks on the climate of the mid-Holocene and the Last Glacial Maximum*, Climate of the Past, 9:2717-2730.

Renssen, H., Seppa, H., Heiri, O., Roche, D.M., Goosse, H., Fichefet, T., 2009. *The spatial and temporal complexity of the Holocene thermal maximum*, Nat. Geosci., 2:410-413.

Luterbacher, J., Garcia Herrera, R., Allan, R., Alvarez-Castro, B. G., Benito, G., Booth, J., Buntgen, U., Colombaroli, D., Davis, B., Esper, J., Felis, T., Fleitmann, D., Frank, D., Gallego, D., Gonzalez-Rouco, F. J., Goosse, H., Kiefer, T., Macklin, M. G., Montagna, P., Newman, L., Ribera, P., Roberts, N., Silenzi, S., Tinner, W., Valero-Garces, B., van der Schrier, G., Vanniere, B., Wanner, H., Werner, J. P., Willett, G., Xoplaki, C. S., Zerefos, E. and Zorita, E., 2012. *A review of 2000 years of paleoclimate evidence in the Mediterranean*, in: Lionello, P. (ed.) The climate of the Mediterranean region: From the past to the Future. Elsevier Insights. Elsevier, Amsterdam, pp. 87-186.

Osborn,T.J., Briffa, K.R., 2000. *Revisiting timescale-dependent reconstruction of climate from tree-ring chronologies*, Dendrochronologia, 18:9-26.

Fischer, N., Jungclaus, J.H., 2011. *Evolution of the seasonal temperature cycle in a transient*

*Holocene simulation: orbital forcing and sea-ice*, Climate of the Past, 7, 1139-1148.

Heiri, O., Ilyashuk, B., Millet, L., Samartin, S., Lotter, A., 2015. *Stacking of discontinuous regional paleoclimate records: Chironomid-based summer temperatures from the Alpine region*, The Holocene, 25(1):137-149.

Please also note the supplement to this comment:
http://www.clim-past-discuss.net/cp-2016-10/cp-2016-10-AC2-supplement.pdf

———————————————————

**Table 1.** *Winter Temperature Cost Function Estimates for the CCLM and the ECHAM5 models compared to the Proxy reconstructions for each time slice of mid-to-late Holocene. Values closer to 0 indicate a better agreement with proxy reconstructions.*

| Time Slice | CCLM | ECHAM5 |
|---|---|---|
| 6000BP | 0.87 | 0.92 |
| 5000BP | 0.88 | 0.92 |
| 4000BP | 0.77 | 0.84 |
| 3000BP | 0.78 | 0.82 |
| 2000BP | 0.77 | 0.79 |
| 1000BP | 0.61 | 0.61 |

**Table 2.** *As Table 1 but for Summer Temperature*

| Time Slice | CCLM | ECHAM5 |
|---|---|---|
| 6000BP | 0.93 | 0.96 |
| 5000BP | 0.72 | 0.72 |
| 4000BP | 0.65 | 0.67 |
| 3000BP | 0.63 | 0.71 |
| 2000BP | 0.48 | 0.54 |
| 1000BP | 0.43 | 0.48 |

**Table 3.** *Comparison of Winter temperature anomalies between 6000BP and Pre-Industrial times, for different simulations of the PMIP3 experiment and as represented in our simulations. Also the data calculated from the pollen reconstructions are provided at the bottom of the table. The values represent mean values over the regions between 35:50N and -10:40E(South), and 55:72N and -10:40E (North).*

| Model | North | South |
|---|---|---|
| BCC-CSM1-1 | 1.08 | -0.18 |
| CCSM4 | -0.65 | -0.50 |
| CCSM4 | -0.62 | -0.32 |
| CNRM-CM5 | 1.45 | 0.27 |
| CSIRO-Mk3-6-0 | 0.69 | 0.19 |
| FGOALS-g2 | 0.13 | -0.99 |
| FGOALS-g2 | -1.13 | -0.26 |
| GISS-E2-R | 0.39 | -0.01 |
| IPSL-CM5A-LR | 0.91 | 0.03 |
| MIROC-ESM | 0.14 | -0.47 |
| MPI-ESM | -0.48 | -0.35 |
| MRI-CGCM3 | 0.23 | -0.16 |
| CCLM | 0.83 | -0.29 |
| ECHAM5 | 1.1 | -0.33 |
| ECHO-G | 0.21 | -0.11 |
| Pollen | 2.51 | -0.66 |

**Table 4.** *As Table 3, but for Summer Temperature*

| Model | North | South |
|-------|-------|-------|
| BCC-CSM1-1 | 1.52 | 1.21 |
| CCSM4 | 0.81 | 1.15 |
| CCSM4 | 1.06 | 1.36 |
| CNRM-CM5 | 1.29 | 1.21 |
| CSIRO-Mk3-6-0 | 1.25 | 1.69 |
| FGOALS-g2 | 0.53 | 0.76 |
| FGOALS-g2 | 0.89 | 1.29 |
| GISS-E2-R | 1.26 | 0.41 |
| IPSL-CM5A-LR | 1.21 | 1.30 |
| MIROC-ESM | 0.81 | 1.20 |
| MPI-ESM | 1.19 | 1.09 |
| MRI-CGCM3 | 1.01 | 1.22 |
| CCLM | 0.83 | 0.85 |
| ECHAM5 | 1.16 | 0.67 |
| ECHO-G | 1.24 | 0.49 |
| Pollen | 0.64 | -1.17 |

[Figure]

**Fig. 1.** Maps of Winter (left) and Summer (right) 2 meters temperature anomalies between 6000BP and the preindustrial period. The results of the different models are presented: CCLM(top),ECHAM5(center),ECHOG()

**Winter**

[Figure]

**Fig. 2.** Left: Maps of Winter temperature anomalies between CCLM and Pollen Reconstructions for the different time slices of mid-to-late Holocene. Right: Uncertainties in the winter seasonal mean of the pollen

**Summer**

**CCLM-Pollen Anomalies**

*6000 BP* *5000 BP*

*4000 BP* *3000 BP*

*2000 BP* *1000 BP*

$^{o}C$

**Pollen Uncertainties**

*6000 BP* *5000 BP*

*4000 BP* *3000 BP*

*2000 BP* *1000 BP*

$^{o}C$

**Fig. 3.** As in Fig.2 but for Summer seasonal means

**CCLM**

**POLLEN**

DJF

*a)*

*b)*

JJA

*c)*

*d)*

−0.6  −0.4  −0.2  0.0  0.2  0.4  0.6

**Fig. 4.** Mid-to-late Holocene temporal Evolution of 2 meters temperature seasonal mean. The maps show the slopes of the linear trends calculated, for every grid box, taking into consideration the uncertaintie

**Supplement:**

**Comment on Russo & Cubasch, *Mid-to-late Holocene Temperature Evolution and Atmospheric Dynamics over Europe in Regional Model Simulations***

By Basil Davis

I would just like to reply to a number of comments made by Reviewer 2 with respect to the reconstruction of Mauri et al. (2015), and the data-model discrepancies highlighted by Russo & Cubasch in their paper.

In the first instance I would encourage all of the reviewers to read the original Mauri et al. (2015) paper, where we have gone to some lengths to address the kind of concerns expressed by the climate modeling community that are also expressed here.

I will start with a general overview, and then move to the specific comments.

The reconstruction of Mauri et al. (2015) is based on over 800 well-dated pollen-based quantitative climate records from throughout Europe, many hundreds of which come from Southern Europe. This data has been projected onto a uniform spatial grid that is consistent with, and appropriate to, the spatial resolution used by climate models. The method derives a regional climate record from a dense network of local site records, thereby minimizing the role of local climatic factors that may not be represented at the scale of most models. Uncertainties for the reconstruction have been fully documented and the reconstruction successfully evaluated against other reconstructions based on other proxies and methods (Chironomids being a notable and previously documented exception). Similar pollen-based climate reconstructions have been extensively used for data-model comparisons, and most recently for the evaluation of the PMIP3/CMIP5 climate models included in the last IPCC report (Stocker et al. 2013, Harrison et al. 2015). Cooler summer temperatures over much (but by no means all) of Southern Europe in the early-mid Holocene have been a feature of all pollen-based climate reconstructions since the seminal work of Huntley & Prentice (1988) almost 30 years ago.

*Reviewer #2: Paleoclimate reconstructions based on pollen rely on the assumption that changes in the vegetation were driven by the parameter to be reconstructed (i.e. summer temperature). In the Mediterranean region, vegetation distribution is mainly limited by effective precipitation, rather than by summer temperature (e.g. Osborne et al. 2000).*

Effective precipitation is an important factor in determining the distribution of the Mediterranean vegetation, but temperatures are also important through thermic responses such as metabolic processes and frost tolerance. No vegetation model would be able to successfully predict the distribution of the Mediterranean vegetation based on effective precipitation alone.

The reference cited by the reviewer (Osborne et al., 2000) uses a vegetation model to highlight the sensitivity of Mediterranean vegetation to changes in $CO_2$. The effect of $CO_2$, and the relative importance of temperature and precipitation in determining past vegetation change in the Mediterranean have already been comprehensively investigated by Wu et al. (2007) using inverse modeling. Usually, climate is used as

an input into a vegetation model to arrive at a vegetation. In inverse modeling, the vegetation (in this case determined from pollen data) is used as an input to arrive at a climate. This method therefore reconstructs climate from pollen data in an entirely independent way from the modern analogue method used by Mauri et al. (2015). It also means that CO2 can also be included as an independent input and its effects included in the analysis. The results of inverse modeling by Wu et al. (2007) show however the same mid-Holocene summer cooling reconstructed by Mauri et al. (2015) (Figure 1), indicating that irrespective of change in precipitation and CO2, it is still necessary to have cooler summers in order to explain the vegetation shown in the pollen record.

[Figure]

*Figure 1. Comparison of mid-Holocene summer temperature anomalies reconstructed from pollen data using the Modern Analogue Technique (MAT) and Inverse Modeling Method (IVM). Results are shown for all sites in Southern Europe that were used by both Wu et al. (2009/7) and Davis et al. (2003). The comparison shows how similar summer temperature anomalies are reconstructed by both MAT and IVM. MTWA: Mean Temperature of the Warmest Month. TEDE: Temperate Deciduous Forest, WAMX: Warm Mixed Forest, CLMX: Cool Mixed Forest.*

The reviewer appears to suggest that the reconstruction of Mauri et al. (2015) is somehow biased because of the insensitivity of Mediterranean vegetation to temperature. Putting aside whether this is a real problem or not, Mediterranean vegetation is not actually the underlying cause of the reconstructed early-mid Holocene summer cooling that the models are unable to reproduce. The summer cooling comes instead from the expansion of temperature deciduous vegetation south into the Mediterranean, the same temperature deciduous vegetation that also expands north into Scandinavia at this time. In the north of Europe this is the result of warmer summer temperatures, and in the south of Europe the result of cooler summer temperatures. It is the same vegetation in both cases and the reconstruction has the same confidence in both regions. There is therefore no methodological reason to both accept the reconstruction for Northern Europe that agrees with the models, while at the same time rejecting the reconstruction for Southern Europe because it does not agree with the models. It should also be noted that the cooling in summer that is reconstructed over Southern Europe is also independently replicated over the same latitudes of the Southern USA (Viau et al. 2006) (Figure 2), while similarly not being shown in model simulations for the region (eg. Sawada et al. 2004, Renssen et al 2009).

[Figure]

Figure 2. The top panel shows the proportional change in land cover (pollen-biomes) for Southern Europe (south of 45N) for the Holocene (From Davis et al. 2015). Note that the early-mid Holocene is characterized by an expansion of Temperate Deciduous Forest with little change in the area of Xerophytic Mediterranean scrub. The expansion of cooler temperate taxa is what drives the cooler summer temperatures shown in pollen-based climate reconstructions at the same regional scale (in this case from Davis et al. 2003). These cooler summer temperatures are also reproduced in regional pollen-based climate reconstructions for the same latitudes (south of 45N) across the Southern USA (again based on many hundreds of sites), despite being based on an entirely different modern and fossil pollen dataset (Viau et al. 2006).

Even if we reject all pollen-based climate reconstructions, we can still show that the discrepancy between pollen data and models over Southern Europe is real. In this approach the climate model output is used as an input into a vegetation model, and the resulting climate model derived vegetation is compared to the pollen record. Here again, the data-model discrepancy does not go away. The climate models all suggest summer temperatures were even higher than today in the early-mid Holocene, with summer precipitation either similar or less than today. This results in greater summer aridity than today with a prolonged summer drought and a large decrease in effective moisture for plant growth. The vegetation simulated by climate models is much too dry than that suggested by the pollen data, with an expansion of Mediterranean and steppe/desert vegetation and contraction in forest cover (Prentice et al. 1998, Wohlfart et al. 2004, Gallimore et al. 2005, Garzon et al. 2007, Kleinen et al. 2010) (Figure 3)

[Figure]

Figure 3. A comparison of biomes simulated by a vegetation model driven by a mid-Holocene climate model simulation, and biomes reconstructed from pollen data (from Prentice et al. 1998).

The warmer climate simulated by the models also results in a large increase in evaporation in the early-mid Holocene, making it difficult for climate models to explain the near equilibrated hydrological budget of the Mediterranean Sea at this time (Duplessy et al. 2005), which led to stratification and sapropel formation. Similarly, and for the same reason, models have difficulty reproducing evidence of higher lake levels (Harrison et al. 1993, Magny et al. 2013). Higher lake levels suggest a more favourable moisture balance that is entirely consistent with the expansion of temperature deciduous and drought intolerant taxa shown in the pollen data.

*Reviewer #2: For instance, summer temperature reconstructions from the S Europe domain based on Chironomids, show a clear Holocene cooling (Heiri et al. 2015; Toth et al. 2015) that actually support the presented modelling results.*

The reconstruction of Mauri et al. (2015) is based on many hundreds of individual temperature reconstructions from sites across Southern Europe assimilated onto a uniform spatial grid that is consistent with the spatial resolution of a climate model. The reconstruction shows regional and local variability in temperature trends that can be clearly seen in the spatial pattern of anomalies shown in the paper. This includes warming at altitude in the Alps and Carpathian mountains in agreement with the trends at the sites cited by the referees, although it is true that in general Mauri et al. (2015) show that Chironomid records are not generally replicated by pollen-based

records. This is consistent with other studies that have compared pollen and chironomid based reconstructions (Velle et al. 2010). For instance, while hundreds of pollen-based climate reconstructions from sites in Northern Europe show early-mid Holocene warming (Mauri et al. 2015) in agreement with climate models (Renssen et al. 2009), it is also possible to find Chironomid records from the region that show cooling at this time that is contrary to climate model simulations (Figure 4).

[Figure]

*Figure 4. Comparison of a pollen-based reconstruction (A) and chironomid-based reconstruction (B) from Lake Gilltjarnen, northern Central Sweden (from Antonsson et al. 2006). The pollen record shows mid-Holocene warming while the Chironomid record shows mid-Holocene cooling.*

Southern Europe in particular is topographically diverse, with many mountain ranges, islands, inlets and other geographic features that can have an important impact on local climate but which are not represented at the grid box scale of a climate model. Consequently changes in prevailing wind direction, air masses, lapse rates and other dynamic components of the climate system at the regional scale may have local impacts that are recorded at individual sites, but which are not representative of climate change at the regional or grid box scale. For instance, the pollen-based reconstruction by Cheddadi et al. (1998) at the site of Tigalmamine in the Atlas Mountains of Morocco shows early-mid Holocene warming which appears to contradict the general cooling reconstructed by Mauri et al. (2015). However, the study by Mauri et al. (2015) included the same data from this site, and the same warming was reconstructed. What the Mauri et al. (2015) reconstruction shows is that the temperature changes recorded at the Tigalmamine site are not representative at the regional scale when all sites in the region are considered. The regionally anomalous nature of the Tigalmamine site is also clear from the precipitation reconstruction published alongside the temperature reconstruction, which shows early-mid Holocene aridity at a time when most of North Africa was experiencing a pronounced humid period (Figure 5).

[Figure]

*Figure 5. The pollen-based climate reconstruction by Cheddadi et al. (1998) at the site of Tigalmamine in the Atlas Mountains of Morocco shows two regionally anomalous features during the early-mid Holocene. The first is the early Holocene warming shown in the temperature record, which is not reproduced in other sites in the region (Mauri et al. 2015). The second is the early Holocene aridity shown in the precipitation record, which occurs at a time when precipitation increased regionally during the 'African Humid Period' (de Monecal et al. 2000).*

The interpretation of individual sites and small site networks as regionally representative should be treated with caution, especially in heterogeneous regions such as Southern Europe. Even during the widespread warming of the Twentieth Century it is still possible to find sites that show a contrary cooling trend (Figure 6), and small site networks that happen to be based on these sites will provide a very different impression of climate change than a large site network similar to the one we have used in Mauri et al. (2015).

[Figure]

*Figure 6. Even during the widespread warming of the Twentieth Century it is still possible to find a few sites that show a contrary cooling trend.*

*In addition, Holocene SST reconstructions from the Mediterranean Sea show a similar cooling trend (e.g. Marchal et al. 2002).*

As with the pollen-based climate reconstructions, evidence from SST reconstructions in the Mediterranean region show both warmer and cooler temperatures during the Holocene. For instance, cooling has been found in the following studies; Kallel et al. 1997, Emeis et al. 2000, Siani et al. 2001, Sbaffi et al. 2001, Melki et al. 2009, Adloff et al. 2011. Attempts to understand Holocene SST conditions at regional spatial scales have been limited because the multi-site record is more complex than it often appears from individual site records. This complexity has revealed itself when attempts are made to compare closely located sites based on the same proxy, or records from the same or closely located sites based on different proxies, or to establish modern baseline SST's in order to calculate anomalies to show actual warming/cooling as opposed to simple trends (Hessler et al. 2014). The authors of the Marchal et al. (2002) paper cited by the reviewer also present similar conclusions.

For instance the site of D13822 near the coast of Portugal shows strong early-mid Holocene warming, but the nearby site of SU81-18 does not (Figure 7). As we have already mentioned, it is important not to assume that a single site that reflects local conditions is representative of climate change at a regional or grid-box scale, particularly given clear evidence from Mauri et al. (2015) of local climate variability in the Mediterranean region.

[Figure]

*Figure 7. A comparison of the Holocene SST alkenone (UK'37) record from site D13822 off the coast of Portugal (Rodrigues et al.2010) with SST records from the nearby site SU81-18. Both alkenone (UK'37) (Bard et al. 2000) and foraminifera (Waelbroeck et al. 2001) SST reconstructions from site SU81-18 fail to reproduce the strong warming seen in site D13822.*

Many of the marine sites that show the strongest early-mid Holocene SST warming in the Mediterranean region are based on alkenones. Process studies have shown that SST reconstructions based on this proxy can be subject to bias due to sensitivity to changes in river discharges and surface mixing (Versteegh et al. 2007, Abrantes et al. 2009; Grauel et al. 2013). It has also been suggested that alkenones may be less reliable in low salinity conditions (Bendle and Rosell-Mele, 2004).

It is not clear what effects, if any, these possible sources of bias may have had on the SST record, but we do know that the Mediterranean Sea has undergone profound changes during the Holocene. For instance, river discharges into the Mediterranean have varied considerably through the Holocene, being largely responsible for the development of an equilibrated hydrological budget in the early-mid Holocene that decreased salinity and allowed stratification and sapropel formation (Rohling et al. 2015).

Lastly, in considering the reliability of the reconstruction by Mauri et al. (2015) we can also mention plausibility. Importantly, the spatial patterns of anomalies that we reconstruct do not appear to be randomly distributed within their uncertainty bounds, but have a spatial coherence that varies systematically through time. In Mauri et al. (2014) we show that these spatial patterns of temperature changes are entirely plausible based on changes in atmospheric dynamics. These changes in atmospheric dynamics are not shown in climate models, but have been suggested by other authors based on many different lines of evidence. These include changes in the Norwegian and Arctic Ocean currents, Norwegian glacier mass balance, the salinity of the Baltic, isotopic composition of lake sediments in Sweden and groundwater fed lakes in Spain.

**References**

Abrantes, F., Lopes, C., Rodrigues, T., Gil, I., Witt, L., Grimalt, J., Harris, I., 2009. Proxy calibration to instrumental data set: Implications for paleoceanographic reconstructions. *Geochem Geophy Geosy* 10.

Adloff, F., Mikolajewicz, U., Kucera, M., Grimm, R., Maier-Reimer, E., Schmiedl, G., Emeis, K.C., 2011. Upper ocean climate of the Eastern Mediterranean Sea during the Holocene Insolation Maximum - a model study. *Clim Past* 7, 1103-1122.

Antonsson, K., Brooks, S.J., Seppa, H., Telford, R.J., Birks, H.J.B., 2006. Quantitative palaeotemperature records inferred from fossil pollen and chironomid assemblages from Lake Gilltjarnen, northern central Sweden. *J Quaternary Sci* 21, 831-841.

Bard, E., Rostek, F., Turon, J.L., Gendreau, S., 2000. Hydrological impact of Heinrich events in the subtropical northeast Atlantic. Science 289, 1321-1324.

Bendle, J. A. and Rosell-Melé, A. 2004: Distributions of UK37 and UK37? in the surface waters and sediments of the Nordic Seas: implications for paleoceanography. *Geochemistry, Geophysics, Geosystems* 5, Q11013-Q11013,

Cheddadi, R., Lamb, H.F., Guiot, J., van der Kaars, S., 1998. Holocene climatic change in Morocco: a quantitative reconstruction from pollen data. *Clim Dynam* 14, 883-890.

Davis, B.A.S., Brewer, S., Stevenson, A.C., Guiot, J., 2003. Th e temperature of Europe during the Holocene reconstructed from pollen data. Quaternary Sci Rev 22, 1701-1716.

Davis, B.A.S., Collins, P.M., Kaplan, J.O., 2015. The age and post-glacial development of the modern European vegetation: a plant functional approach based on pollen data. Veg Hist Archaeobot 24, 303-317.

de Menocal, P., Ortiz, J., Guilderson, T., Adkins, J., Sarnthein, M., Baker, L., Yarusinsky, M. (2000) Abrupt onset and termination of the African Humid Period: Rapid climate responses to gradual insolation forcing. Quat. Sci. Rev. 19, 347–361.

Duplessy, J.C., Cortijo, E., Kallel, N., 2005. Marine records of Holocene climatic variations. *Cr Geosci* 337, 87-95.

Emeis, K.C., Struck, U., Schulz, H.M., Rosenberg, R., Bernasconi, S., Erlenkeuser, H., Sakamoto, T., Martinez-Ruiz, F., 2000. Temperature and salinity variations of Mediterranean Sea surface waters over the last 16,000 years from records of planktonic stable oxygen isotopes and alkenone unsaturation ratios. *Palaeogeogr Palaeocl* 158, 259-280.

Gallimore, R., Jacob, R., Kutzbach, J., 2005. Coupled atmosphere-ocean-vegetation simulations for modern and mid-Holocene climates: role of extratropical vegetation cover feedbacks. *Clim Dyn* 25, 755-776.

Garzon, M.B., de Dios, R.S., Ollero, H.S., 2007. Predictive modelling of tree species distributions on the Iberian Peninsula during the Last Glacial Maximum and Mid-Holocene. *Ecography* 30, 120-134.

Grauel, A.L., Leider, A., Goudeau, M.L.S., Muller, I.A., Bernasconi, S.M., Hinrichs, K.U., de Lange, G.J., Zonneveld, K.A.F., Versteegh, G.J.M., 2013. What do SST proxies really tell us? A high-resolution multiproxy (U-37(K '), TEX86H and foraminifera delta O-18) study in the Gulf of Taranto, central Mediterranean Sea. *Quaternary Sci Rev* 73, 115-131.

Harrison, S.P., Digerfeldt, G., 1993. European Lakes as Paleohydrological and Paleoclimatic Indicators. *Quaternary Sci Rev* 12, 233-248.

Harrison, S.P., Bartlein, P.J., Izumi, K., Li, G., Annan, J., Hargreaves, J., Braconnot, P., Kageyama, M., 2015. Evaluation of CMIP5 palaeo-simulations to improve climate projections. Nat Clim Change 5, 735-743.

Hessler, I., Harrison, S.P., Kucera, M., Waelbroeck, C., Chen, M.T., Anderson, C., de Vernal, A., Frechette, B., Cloke-Hayes, A., Leduc, G., Londeix, L., 2014. Implication of methodological uncertainties for mid-Holocene sea surface temperature reconstructions. *Clim Past* 10, 2237-2252.

Huntley, B., Prentice, I.C., 1988. July Temperatures in Europe from Pollen Data, 6000 Years before Present. *Science* 241, 687-690.

Kallel, N., Paterne, M., Labeyrie, L., Duplessy, J.C., Arnold, M., 1997. Temperature and salinity records of the Tyrrhenian Sea during the last 18,000 years. *Palaeogeogr Palaeocl* 135, 97-108.

Kleinen, T., Brovkin, V., von Bloh, W., Archer, D., Munhoven, G., 2010. Holocene carbon cycle dynamics. *Geophys Res Lett* 37, -.

Magny, M., Combourieu-Nebout, N., de Beaulieu, J.L., Bout-Roumazeilles, V., Colombaroli, D., Desprat, S., Francke, A., Joannin, S., Ortu, E., Peyron, O., Revel, M., Sadori, L., Siani, G., Sicre, M.A., Samartin, S., Simonneau, A., Tinner, W., Vanniere, B., Wagner, B., Zanchetta, G., Anselmetti, F., Brugiapaglia, E., Chapron, E., Debret, M., Desmet, M., Didier, J., Essallami, L., Galop, D., Gilli, A., Haas, J.N., Kallel, N., Millet, L., Stock, A., Turon, J.L., Wirth, S., 2013. North-south palaeohydrological contrasts in the central Mediterranean during the Holocene: tentative synthesis and working hypotheses. *Clim Past* 9, 2043-2071.

Mauri, A., Davis, B.A.S., Collins, P.M., Kaplan, J.O., 2014. Th e infl uence of atmospheric circulation on the mid-Holocene climate of Europe: a data-model comparison. Cimates of the Past 10, 1925–1938.

Melki, T., Kallel, N., Jorissen, F.J., Guichard, F., Dennielou, B., Berne, S., Labeyrie, L., Fontugne, M., 2009. Abrupt climate change, sea surface salinity and paleoproductivity in the western Mediterranean Sea (Gulf of Lion) during the last 28 kyr. *Palaeogeogr Palaeocl* 279, 96-113.

Prentice, I.C., Harrison, S.P., Jolly, D., Guiot, J., 1998. The climate and biomes of Europe at 6000 yr BP: Comparison of model simulations and pollen-based reconstructions. *Quaternary Sci Rev* 17, 659-668.

Renssen, H., Seppa, H., Heiri, O., Roche, D.M., Goosse, H., Fichefet, T., 2009. The spatial and temporal complexity of the Holocene thermal maximum. *Nat Geosci* 2, 410-413.

Rodrigues, T., Grimalt, J.O., Abrantes, F.G., Flores, J.A., Lebreiro, S.M., 2009. Holocene interdependences of changes in sea surface temperature, productivity, and fluvial inputs in the Iberian continental shelf (Tagus mud patch). Geochem Geophy Geosy 10.

Rohling, E.J., Marino, G., Grant, K.M., 2015. Mediterranean climate and oceanography, and the periodic development of anoxic events (sapropels). *Earth-Sci Rev* 143, 62-97.

Sawada, M., Viau, A.E., Vettoretti, G., Peltier, W.R., Gajewski, K., 2004. Comparison of North-American pollen-based temperature and global lake-status with CCCma AGCM2 output at 6 ka. Quaternary Sci Rev 23, 225-244.

Sbaffi, L., Wezel, F.C., Kallel, N., Paterne, M., Cacho, I., Ziveri, P., Shackleton, N., 2001. Response of the pelagic environment to palaeoclimatic changes in the central Mediterranean Sea during the Late Quaternary. *Marine Geology* 178, 39-62.

Siani, G., Paterne, M., Michel, E., Sulpizio, R., Sbrana, A., Arnold, M., Haddad, G., 2001. Mediterranean Sea surface radiocarbon reservoir age changes since the last glacial maximum. *Science* 294, 1917-1920.

Stocker, T.F., Qin, D., Plattner, G.-K., Tignor, M., Allen, S.K., Boschung, J., Nauels, A., Xia, Y., Bex, V., Midgley, P.M., 2013. Climate Change 2013: The Physical Science Basis. Contribution of Working Group I to the Fifth Assessment Report of the Intergovernmental Panel on Climate Change. Cambridge University Press, Cambridge, United Kingdom and New York, NY, USA.

Velle, G., Brodersen, K.P., Birks, H.J.B., Willassen, E., 2010. Midges as quantitative temperature indicator species: Lessons for palaeoecology. Holocene 20, 989-1002.

Versteegh, G.J.M., de Leeuw, J.W., Taricco, C., Romero, A., 2007. Temperature and productivity influences on U-37(K') and their possible relation to solar forcing of the Mediterranean winter. *Geochem Geophy Geosy* 8, Q09005.

Viau, A.E., Gajewski, K., Sawada, M.C., Fines, P., 2006. Millennial-scale temperature variations in North America during the Holocene. J Geophys Res-Atmos 111, D09102.

Wohlfahrt, J., Harrison, S.P., Braconnot, P., 2004. Synergistic feedbacks between ocean and vegetation on mid- and high-latitude climates during the mid-Holocene. Clim Dynam 22, 223-238.

Waelbroeck, C., Duplessy, J.C., Michel, E., Labeyrie, L., Paillard, D., Duprat, J., 2001. Th e timing of the last deglaciation in North Atlantic climate records. Nature 412, 724-727.

Wu, H.B., Guiot, J.L., Brewer, S., Guo, Z.T., 2007. Climatic changes in Eurasia and Africa at the last glacial maximum and mid-Holocene: reconstruction from pollen data using inverse vegetation modelling. Clim Dynam 29, 211-229.

---

## Author Comment (AC3) · 11 Apr 2016

Reply to
**1st Reviewer**
*Mid-to-late Holocene Temperature Evolution and Atmospheric Dynamics over Europe in Regional Model Simulations by Russo, Emmanuele; Cubasch, Ulrich cp-2016-10*

Dear reviewer,

Thank you very much for your effort in reviewing our paper.

Below we go point by point through your technical corrections, detailing how we dealt with your concerns reported in **bold**.

Thank you.

- **General Comments**

  **The paper can be broadly divided into three parts: i) validation of the simulation, ii) comparison with reconstructions iii) search for explanations of the disagreements. In all these three elements, I can identify caveats that in my opinion should be improved with different/complementary analyses. I have tried to review them in a comprehensive, yet constructive way, as detailed below. Besides the technical aspects, I think there is room in the manuscript for improvement regarding writing style. It was challenging for me to read and understand many parts of the paper. This is in part due to incomplete information in the captions and the main text, wrong labelling in the figures, and the misleading use of some concepts as "observation" or "validation". The internal structure in the paragraphs is confusing: paragraphs loosely connected, overly short, or in a misleading order respect to the panels in the figures. These issues add complexity that makes the lecture of the paper uncomfortable. Despite this rather negative view, I try to be constructive giving a list of points that develop the aspects that in my opinion can be improved in the manuscript. Note however that this list is not comprehensive.**

We propose to develop a clearer structure of the manuscript as suggested by the reviewer. We will divide the manuscript in three main parts: the first based on the validation of the model configuration for present-days, the second one based on the comparison against proxy-data, additionally providing analyses and discussion on the advantages of the use of highly resoluted simulations for the comparison against reconstructions, and the third one in which we will provide explanations for possible mismatches. We also provide improved/complementary analyses accordingly to the referee's comments. In addition, we propose to correct the manuscript, keeping in mind that, as mentioned by the referee, the comparison against proxy data is by no means a validation. We aim at providing such comparison in a clearer way, cautious on the use of proper terminology. Additionally, as detailed in the comments below, we will try to improve the grammar of the manuscript and its presentation, in order to better tie together the entire discussion.

1. **Abstract/Introduction:**
   **L1-3: In the first line, "is always been mentioned" is grammatically wrong. Despite that, it sounds a bit loose, almost sceptical. Is it an important factor or not? This ambiguous tone of the first sentence of the abstract is manifest through the whole manuscript. By the way, the authors do not make an attempt to demonstrate that this is indeed the case for these simulations. More on this below.**

   We reformulated the sentence according to additional analysis we aim to provide within the text. In the revised version of the manuscript we will present a section in which we conduct a detailed comparison between the models at different resolutions and proxy-data, elucidating possible advantages of Dynamical downscaling. Further details on such analysis are presented in the comments to the second referee. We aim at implementing our discussion and the manuscript consequently, following a common suggestion of both the authors.

   **L5-6: The paper is somewhat optimistic regarding the use of "for the first**

**time". It is true that, as far as I know, there are no other set of time slide simulations. However there are various high-resolution simulations for the last millennium for Europe. Actually there exists at least one transient simulations for the last two millennia, in fact driven by the same ECHO-G run used by the authors of the manuscript. The authors should not ignore such previous, yet scarce efforts in this topic in the intro, but also the discussion of the results.**

We agree with the referee and we aim at modifying the previous sentence. As the the referee mentioned, there are other paleo-simulations for Europe for mid-to-late Holocene. Nevertheless, such simulations often investigate only a time slice or do not cover the entire mid-to-late Holocene. Even if they do so, as the case of the ECHO-G simulation used for this study, their low resolution is often mentioned as one of the possible reasons for the disagreement between model results and reconstructions (Fischer $\&$ Jungclaus (2011);Bonfils et al. (2004)). With the previous sentence, we wanted to highlight the fact that no previous simulation exists for Europe, at such high resolution, covering different time-slices of mid-to-late Holocene. Our optimism, in this sense, regards the fact that these simulations could contribute in clarifying the debate on models and proxy disagreement. Additional discussion and more references (e.g. Strandberg et al. (2014),Schimanke et al. (2012),Braconnot et al. (2007a),Braconnot et al. (2007b)) will be added within the text.

**L6: In line 6, "validation" is used in a wrong context. The model is validated normally against observations. But you can not validate the model looking at a reconstruction. Neither you validate a reconstruction looking at a simulation. You can only compare them, and try to gain insight through the disagreements. The use of "validation" in this wrong context is spread through the manuscript and should be avoided.**

Accordingly to the referee's comment, we think that the terminology previously employed was incorrect. We aim at correcting the term "validation" with "comparison" here and throughout the manuscript, when referring to the comparison with proxy data .

**I think at least the first four paragraphs can be safely merged.**

We agree with the reviewer and propose merging such paragraphs, reformulating them in a more concise and clearer way.

**L39-41: it is argued that then changes in solar irradiation were "negligible", and latter than "we expect that such changes would imply relevant variations...". It sound contradictory.**

In this sentence our goal was to highlight the fact that during mid-to-late Holocene yearly variations in insolation, over northern latitudes in general, were neglicible when compared to the seasonal variations. The latest are expected to imply "relevant changes in the seasonal values of surface variables". We will re-formulate this period in a more comprehensive way.

**L42-45: The paragraph in lines 42 to 45 is made out of a single sentence, which is too long. Still, the paragraph itself is short and can be merged with the former. Further, such sentence demands references.**

We agree. We will reformulate the paragraph and join it with the former. We will also add further references (i.e. Cheddadi et al. (1997),
Bonfils et al. (2004),Braconnot et al. (2007a),Braconnot et al. (2007b)).

**In the paragraph starting in line 89, some examples of RCM simulations in palaeoclimate applications are outlined. It's strange to see that no simulation for Europe is referred. Examples of such simulations are: Gómez-Navarro et al. (2011, 2012, 2013, 2015a, 2015b) and Schimanke et al. (2012).**

We agree. Apart from the ones suggested by the referee, we will also take into consideration, in the revised version, other works in which high resoluted paleo-simulations for Europe were performed (i.e. Strandberg et al. (2014);Renssen et al. (2001)).

2. **Model Validation:**

   **It is not clear how long is the control period used in section 3.1. The only hint is the label in Figure 2, 1990-2000. Is that the case? It should be clearly stated, not only in the main text, but also in the caption of the figure. Actually, the length of this period is CRITICAL for the model evaluation, a fact that is not acknowledged in the discussion of the results. A 10-year period of a GCM simulation is strongly populated with internal variability. Under this scenario, a comparison with observations is tricky. The model could be "by chance" going through a cold or warm phase, which would have a strong impact in the validation, at least in the way it has been established in the paper, focused on mean values. In this sense, the validation does not look at important aspects such as the variability. How is the variance reproduced by the model? I'm not sure due to the short length of the simulation, but it could make sense to look at the variability modes of temperature and precipitation.**

   As indicated by the referee, the control run is 10 years long and covers the period 1991-2000. We propose to add more informations on the length of the simulation throughout the text. We will also add such specification in the caption of previous Fig.2, Fig.3 and Fig.4. We are aware that the length of this simulation is CRITICAL. Unfortunately, due to computational reasons, we were not able to cover a longer period. We will now acknowledge such choice within the text. Additionally, realizing that we have not been properly explicit in the description of our experiment, we want to clarify that the regional simulation was driven by the

ERA-Interim reanalysis dataset and not by the GCM, as mentioned by the referee. We will implement our text accordingly, with the main goal of better specifying such technical details. Since the main goal of the present-day experiment is to test whether the changes applied to the model routine, for this particular case of study, allow to obtain reliable results in comparison to the outcomes of other studies, we think that the validation focusing on mean values is a useful tool in this context. Nevertheless, we also conduct now an analysis of model and observations variability that we aim to provide as supplementary material in the revised manuscript. Additionally, we now also consider the E-OBS dataset (Haylock et al. (2008)) as a benchmark for the validation of our results, and present the mean climatology of temperature and precipitation, for both winter and summer, as reproduced in the three datasets.

The new analyses are shown in Fig.1, Fig.2., Fig.3, Fig.4 and Fig.5. A more detailed description is provided in the captions of such figures. In addition to previous conclusions based on the bias of seasonal mean, the new analyses show that, the model is able to reproduce, with a certain degree of accuracy, the climatology of the observations. Additionally, the analysis of the standard deviation (Fig.4 and Fig.5) shows that the area with the larger bias are the ones where the model is not able to correctly reproduce the variability of the observations, in particular for precipitation.

We will replace the previous analyses of precipitation and temperature with the one presented in Fig.1 and Fig.2 of this text, and develop our discussion accordingly.

**I do not think the choice of target for the validation is the best one. Using ERA Interim for the validation of precipitation in particular is a bad idea, since it is not constrained by precipitation observations, so there is no warranty that this dataset is bias free. I think it would be wiser to use the E-OBS dataset, which was developed specifically for the validation of RCMs**

**in Europe (Haylock et al. 2008).**

We agree with the author that the ERAInterim Reanalysis is not the best choice for model's validation, in particular for precipitation. For this reason, as mentioned above, we now compare the model's results against the CRU observational dataset and the E-OBS dataset.

**The way the similarity between the model and the observations is presented is a bit confusing to me. When the difference between two normally distributed variables is shown, the standard and intuitive approach, which steams from the application of the Central Limit Theorem, is to apply a t-test. The KS test is more suited for testing the shape of PDFs when the mean is know to be the same. For example if two dataset have the same mean, but different variance, the figure would show null bias (yellow colour here), but still the test would produce significant differences, which is misleading for the reader.**

For the comparison of mean values we agree with the referee that a T-test is better suitable. We now perform the Student's T-test for the validation of the considered variables (Fig.1; Fig.2 and Fig.3).

**I think the maps showing precipitation difference are not very useful. A difference of 5 mm/day might be huge or tiny depending on the mean precipitation. I think changes in precipitation are more meaningful when shown as perceptual deviations with respect to the mean.**

We agree with the referee on the fact that changes in precipitation are more meaningful when shown as percentual deviations with respect to the mean. In the new maps (Fig.2), we now present precipitation biases as the percentual deviations from the observations values.

**It is mentioned that the dots indicate grid where differences are "significantly not different". That's not exactly true. They indicate areas where**

**the null hypothesis of the data being sampled from the same underlying distribution could not be ruled out, i.e. where they are "not significantly different".**

We agree. The previous sentence was not totally correct. We propose now to reformulate the sentence as follows: "the dots show the points where the null hypothesis of a Student T-test, at a significance level of 0.05, assuming that the data being sampled could be drawn from the same underlying distribution, is true".

**I agree with the hypothesis used to explain the model deficiencies in Souther Europe regarding soil- atmosphere feedbacks. The particular role of these processes in RCM simulations in areas with strong water deficit was investigated in detail by Jerez et al. (2010, 2012).**

We aim at proposing additional references as the ones indicated by the reviewer and listed at the end of this text.

**L206: reads "These findings CONFIRM that... are MOST PROBABLY...". This is an example of doubtful and confusing sentence that should be avoided.**

We agree. The previous sentence was doubtful as indicated by the reviewer. We will correct our sentence accordingly. Our analysis in fact confirms that model performances are influenced by its scarce capacity to reproduce soil-atmosphere exchanges correctly. This has consequences on both temperature and precipitation (particularly in summer when the biases are more pronounced) presenting a similar pattern of anomalies.

**Maybe is worth to mention that generally the model skill resembles that identified in similar simulations for Europe (Schimanke et al. 2012, Gómez-Navarro et a. 2011, 2013).**

We agree. Although a few works that generally propose similar model skills have been already considered within the discussion paper, it is reasonable to include

additional bibliography, that would help in strengthening our conclusions.

3. **Comparison with Pollen Reconstructions:**

**As pointed out above, this comparison is by no means a model valida-tion. This should be made clear in the wording. As such, all sentences like "CCLM performs well" should be modified. The maps in Figure 5 are calculated as means with respect to which period?**

We are aware of the incorrect terminology employed, as already highlighted be-fore. We propose to correct this period substituting the term "validation" with "comparison", being the pollen reconstructions not an observational dataset. We also aim at using better expressions in order to indicate the good or the bad agreement of the two datasets. We now modified Figure 5 of the discussion paper (also accordingly to the comments of the 2nd reviewer). In the revised manuscript, we aim at presenting the maps of the anomalies as represented in the two datasets, calculated for every investigated period with respect to the pre-industrial times. We also propose to accompany them with the corresponding maps of the pollen-based reconstructions uncertainties. Please refer to the 2nd reviewer response for further details.

**In Figure 6, error bars are provided for the pollen data, but not for the simulation. I'm aware it is not easy to stablish them. However, such er-rors/uncertainties should not be neglected in the discussion of the results. The model has deficiencies that introduce systemic biases. But on top of then, there are non systematic biases introduced by unpredictable internal variability. This factor might lower or rise mean temperature in the simula-tion quite significantly, as pointed out by Gómez-Navarro et al. (2012) in a very similar scenario. Thus, this should be discussed at least qualitatively in this part of the text.**

It has been a choice of the authors not to include the model's uncertainty interval to the plots of Figure 6 of the discussion paper. Since the uncertainties of the 25 years CCLM simulations are way smaller than the ones of the proxy-reconstructions, even if their values need to be considered for the computation of regional means, we think that neglicting them, in this figure, was an appropriate choice.

Nevertheless, following the suggestion of the reviewer, we will add such considerations within the manuscript. Additionally, we will propose a new analysis of the trends of temperature in which the uncertainties are taken into consideration by means of a weighted least squares method. Further details are presented in the next point.

We also will add more discussion of model's uncertainties based on the results of Gómez-Navarro et al. (2012).

**Many conclusions are drawn from Figure 6 regarding matchings of trends. I'm not sure at what extent such conclusions have any statistical significance, since in almost all cases the simulation lies within the uncertainty of the reconstruction. Having an almost perfect match between the reconstruction and the simulation is still perfectly possible within the range of uncertainty of the reconstructions.**

Following the referee's comment, we realyzed that the computation of the mean and of the relative uncertainties presented in Fig.6 of the discussion paper should be re-performed. In particular, the plots we previously proposed and the conclusions we have drawn from them had no statistical significance. In fact, in a first place, we simply calculated the error as a mean of the provided uncertainty for every point. Realizing that this procedure is not correct, we tried to be more cautious with our analyses.

We present now new maps in Fig.6, representing the trends of seasonal means of 2 meters temperature calculated, for every grid box, by means of the weighted

least squares method. The points where the trends are not significant, according to a F-test at a significance level of 0.1, are additionally masked out. We provide again more details in the caption of the figure. We think that these maps are better suitable for our discussion. In fact, they are statistically more robust, allowing to consider trend and relative uncertainties for every grid box and time slice, resulting in a better suitable benchmark for the comparison against the pollen-based or other kinds of reconstructions. In the revised version of the paper we will replace Fig.6 and Fig.11 of the former manuscript with Fig.6 of this text.

**Something I miss in this analysis here is the GCM simulations used to drive COSMO. I wonder how the ECHO-G and later ECHAM5 compares also with reconstructions. Is the RCM adding anything relevant to these simulations? If the answer is "certainly yes", then the use of the RCM is fully justified and the paper would gain interest. If the answer is "mostly no", it would be still interesting, since it would imply that the many GCM simulations available for the last millennia are still relevant at rather regional scales. I'm sure the PMIP community would be very interested in answering this question.**

A similar comment has also been addressed by the 2nd reviewer. We refer to the answer to his comment as an exhaustive response to this point. As suggested by the referee, we think that answering this question would definitely strengthen the paper. In the revised manuscript we will add a section in which the possible advantages of highly resolved simulations for the comparison of change in 2 meters temperature against proxy reconstructions will be investigated. Also more analysis will be presented accordingly.

4. **Interpretation of Paleo Records:**

**Generally it was difficult to follow the arguments in this section. It would significantly help to label the maps as Fig 6b, Fig 6c, etc. and use such labels extensively through the manuscript. In this regard, the discussion of**

**the results starts with summer, whereas the first row shows winter. Small inconsitencies like this, although non critical for the scientific message, have a dramatic impact in the reading pace.**

We agree. In order to make the manuscript more easily readable we propose to label the maps accordingly to the referee's suggestion. We will also correct the order of the seasonal analyses within the text.

**The EOFs for MSLP are shown and used in the discussion. They are used to argue regarding NAO and SNAO, for instance. I'm not totally comfortable with that, since the NAO is defined as the leading pattern for a spatial window that is not that of the RCM. This explains in my opinion why the NAO pattern does not stand out as the leading mode in winter, and second mode in summer just "resembles" the SNAO pattern. I think a more orthodox approach would be to calculate the EOFs within the GCM, in a window that properly encompasses the North Atlantic. This is justified since the large scale circulation is fixed by the GCM, and thus the NAO simulated should be consistent with the climate variability within the RCM domain. Hence, such patterns could still be used to discuss about regional variability within the RCM domain.**

We agree with the referee's comment. We now conduct the EOF analysis of MSLP anomalies of the ECHAM5 simulations in order to properly consider a spatial window that encompasses the entire North Atlantic region. We select the region in between 90W and 40E and in between 20N and 80N, as defined in **?**. This would allow us to infer about changes in the NAO and other atmospheric circulation patterns characteristic of this region. The results are shown in the attached Fig.6 and Fig.7. Since the RCM large scale circulation is "dictated" by the GCM, we reasonably think that such results can be used to argue about regional variability within the RCM. We propose to modify the discussion within the revised manuscript accordingly to the new analysis.

[Figure]

**Line 255 reads "In summer the first EOF shows that the model reproduces similar conditions in atmospheric circulation between the mid-Holocene and pre-industrial times". I do not understand how that conclusion is drawn from the map in Figure 8.**

We propose to modify the previous sentence accordingly to the new analysis presented above. Since the investigation area is different now, the results of the EOF analysis changed. Nevertheless, the time expansion of the principle components of the previously evinced pattern and its structure (representing now the the second mode of atmospheric variability during summer), mainly driven by changes in insolation, seems to be a proper product of this particular case of study. Even if it implies changes in circulation, we do not see any particularly prominent dipole structure characteristic of other well-known circulation patterns for the region. We aim at modifying the discussion within the revised version of the paper, being more cautious about arising risky conclusions as the one spotted out by the referee.

**In page 8 the wording "observed" is used in various sentences, and it's not fully clear what is meant (most likely respect to the simulation, but it could also be the reconstruction). I think "simulated" is more appropriate and precise.**

As highlighted in previous points we agree with the referee and propose to correct the sentence accordingly in the revised manuscript .

**Some inferences about the "clearness" of the sky are made which are based in indirect evidence such as EOF analysis. I think it is not necessary to make such risky affirmations. We have direct information that can tell us exactly how cloudy the simulated climate was. After all, in the simulation we can check directly variables such as cloud cover, which give a direct measure of what is being argued. I would go for a direct measure whenever possible, as it is the case. Similarly, in the paragraph between**

**lines 277 and 279 (and Fig. 11) the more pronounced positive phase of the NAO can be directly tested within the GCMs, rather than indirectly inferred through a map of temperatures.**

We propose to modify the previous sentence within the revised manuscript, accordingly to the fact that, even if the SNAO shows a trend that in this case is positive troughout the mid-to-late Holocene, such trend is not significant and presents high variability. We propose to review our previous discussion and to avoid any conclusion on the trend of cloud cover due to the high variability of the emerged pattern throughout the investigation time, eventually presenting alternative analyses if necessary . As already mentioned, we also preferred to merge Fig.11 together with fig 6 of the discussion paper, considering also summer analysis.

**Finally, I think there are more powerful statistical tools than the one used here to study the co- variability between temperature and MSLP. Canonical Correlation Analysis could be used to derive relations between the variability of MSLP and temperature, and it would produce a picture of such co-variability more robust that the one provided by maps in Figure 10, for instance. An example of the application of such a tool in a very similar context is Gomez-Navarro et al. (2015b)**

We agree. We investigated the covariability of MSLP and temperature by means of Canonical Correlation Analysis and present the results of such analysis in Fig. 7 and Fig.8. We will also refer to the study of Gómez-Navarro et al. (2015b) as a good example of application of such method for the investigation of the relations between atmospheric variability and temperature.

5. **Comments regarding Figures Figure 1: The colour scale shows everything below 1000 meters as green. I think a palette with stronger contrast could be chosen.**

We improved the previous plot accordingly to the referee's suggestion. We add

the modified picture to this discussion and modify it within the paper.

**Figure 2: The reference period should be stated in the caption. I think the limits of the palette can be adjusted to better span the range of temperatures.**

We agree. We added the reference period within the caption and further details. We also provide additional analysis and improve the palette in order to better span the range of temperature.

**Figure 3: The colour palette provides barely any contrast all. Everything is yellow in the maps.**

We modified the plot accordingly to the previous point

**Figures 4 and 5: Same comments as in former figures**

Figure 4 has been adjusted accordingly to the referee's comment. Figure 5 of the discussion paper has now been modified. The new plots are presented in the response to referee number 2 (Fig. 2 and Fig. 3).

**Figure 6: Please label panels as 6a, 6b, etc. I do not think using colour in the caption is an orthodox approach. Note that the caption does not agree with the order of panels. First row does not show North, but it is the first column which does, etc.**

We agree. We now label the panels of the new map presented in Fig.6 as 6a,6b, etc., as suggested by the referee. We also avoid using colours in the caption. We also modified the order of the captions, accordingly to the figure.

**Figure 7: I can barely see the numbers and labels in the figures in the right.**

We enlarged this figure in order to make it more easily readable. Accordingly to a comment of the second referee, we propose to move this picture to section two of the revised paper.

**Figure 8: I think the label with the loading can be moved to inside the maps. This would allow to put the maps closer together, which would allow to make maps larger and more readable. The latter comment can be applied to almost all figures.**

We agree. We moved the loadings inside the maps. We also applied similar modifications to all the pictures in order to make them larger.

**Figure 9: Please label panels to indicate which represent EOF1 etc. Where are the units? Either the EOF or the PC carries the units, in this case pressure. I guess they are included in the EOF patterns in Figure 8. If so, please label the palette accordingly.**

We realized, following the referee's comment, that we were not precise in our previous discussion. Consequently, we propose to add further details in the caption of this figure. In fact, here, we do not indicate units since the analysis we conducted were based on values standardized with respect to the pre-industrial period.

GLDAS dataset, calculated for the refernce period 1991-2000. As in the previous figures, the area with a point represent the grid cells where the anomalies between the two datasets are not significant, according to a Student's T-test, at a significance level of 0.05. Winter results are presented in the first row, and Summer results in the second.

**Figure4**: Analysis of Winter variability of Temperature (left) and Precipitation (right) as simulated by the CCLM in respect to the observational datasets. Comparison against the CRU dataset is shown in the first row, while the one against the E-OBS dataset is presented in the second.

**Figure5**: As in Fig.4 but for summer Temperature and Precipitation.

**Figure6**: Mid-to-late Holocene temporal Evolution of 2 meters temperature seasonal mean. The maps show the slopes of the linear trends calculated, for every grid box, taking into consideration the uncertainties associated to the two datasets, by means of a weighted least squares method. The area masked out in grey, are the area where such trends are not significant, according to a F-test at a significance level of 0.1.

**Figure7**: Canonical correlation pattern pairs of MSLP (left) and T2M (right) in Winter. Each panel illustrates the percentage of variance explained by each pattern and the canonical correlation associated with the pair. The results are calculated for the mid-to-late Holocene, from 6000BP to Pre-industrial times. Note that the MSLP has been obtained directly from the driving GCM, since the window of interest lies outside the RCM domain. Both the variables are standardized with respect to the pre-industrial period, and the units represent the Variance explained by the different patterns.

**Figure8**: As in Fig.7 but for Summer.

**Figure9**: Time expansion of the principal components of the first and second EOFs of winter (**1st row**) and summer (**2nd row**) MSLP anomalies of the ECHAM5 (**lower row**) simulations, standardized to the pre-industrial period.

**Figure10**: Orography Map of the COSMO-CLM simulation domain in rotated

*coordinates.*

**Figure11**: *(Left) Anomalies of zonal mean insolation on top of the atmosphere between pre-industrial period **PI** and 6000 years BP. (Right) Trends of December and June incoming Radiation on top of the Atmosphere.*

**References**

1 Fischer, N., Jungclaus, J., 2011. *Evolution of the seasonal temperature cycle in a transient Holocene simulation: orbital forcing and sea-ice*, Climate of the Past, 7, 1139-1148.

Braconnot, P., Otto-Bliesner, B., Harrison, S., Joussaume, S., Peterchmitt, J.-Y., Abe-Ouchi, A., Crucifix, M., Driesschaert, E., Fichefet, Th., Hewitt, C.D., Kageyama, M., Kitoh, A., Loutre, M.F., Marti, O., Merkel, U., Ramstein, G., Valdes, P., Weber, L., Yu, Y. and Zhao, Y., 2007. *Results of PMIP2 coupled simulations of the Mid-Holocene and Last Glacial Maximum - Part 1: Experiments and large-scale features*, Climate of the Past, 3, 261-277.

Braconnot, P., Otto-Bliesner, B., Harrison, S., Joussaume, S., Peterchmitt, J.-Y., Abe-Ouchi, A., Crucifix, M., Driesschaert, E., Fichefet, Th., Hewitt, C.D., Kageyama, M., Kitoh, A., Loutre, M.F., Marti, O., Merkel, U., Ramstein, G., Valdes, P., Weber, L., Yu, Y. and Zhao, Y., 2007. *Results of PMIP2 coupled simulations of the Mid-Holocene and Last Glacial Maximum - Part 2: feedbacks with emphasis on the location of the ITCZ and mid- and high latitudes heat budget*, Climate of the Past, 3, 279-296.

Cheddadi, R., Guiot, J., Harrison, S., and Prentice, I., 1997. *The climate of Europe 6000 years ago*, Climate Dynamics, 13, 1-9.

Bonfils, C., de Noblet, N., Guiot, J., and Bartlein, P., 2004. *Some Mechanisms of mid-Holocene climate change in Europe, inferred from comparing PMIP models to data*, Climate Dynamics, 23, 79–98.

Strandberg, G., Kjellström, E., Poska, A., Wagner, S., Gaillard, M.J., Trondman, A.K., Mauri, A., Davis, B.A.S., Kaplan, J.O., Birks, H.J.B., Bjune, A.E., Fyfe, R., Giesecke, T., Kalnina, L., Kangur, M., van der Knaap, W.O., Kokfelt, U., Kunes, P., Latalowa, M., Marquer, L., Mazier, F., Nielsen, A.B., Smith, B., Seppa, H. and Sugita, S., 2014. *Regional Climate model simulations*

*for Europe at 6 and 0.2 k BP: sensitivity to changes in anthropogenic deforestation*, Climate of the Past, 10, 661-680.

Renssen, H., Isarin, R., Jacob, D., Podzun, R., and Vandenberghe, J., 2001. *Simulation of the Younger Dryas climate in Europe using a regional climate model nested in an AGCM: preliminary results*, Global and Planetary Changes, 30, 41-57.

Gómez-Navarro, J.J., Montávez, J.P., Jerez, S., Jiménez-Guerrero, P., Lorente-Plazas, R., González-Rouco, J.F., and Zorita, E., 2011. *A regional climate simulation over the Iberian Peninsula for the last millennium*, Climate of the Past, 7, 451-472.

Gómez-Navarro, J.J., Montávez, J.P., Jiménez-Guerrero, P., Jerez, S., Lorente-Plazas, R., González-Rouco, J.F., and Zorita, E., 2012. *Internal and external variability in regional simulations of the Iberian Peninsula climate over the last millennium*, Climate of the Past, 8, 25-36.

Gómez-Navarro, J.J., Montávez, J.P., Wagner, S., and Zorita, E., 2013. *A regional climate palaeosimulation for Europe in the period 1500-1990 - Part 1: Model validation*, Climate of the Past, 9, 1667-1682.

Gómez-Navarro, J.J., Werner, J., Wagner, S., Luterbacher, J., and Zorita, E., 2015. *Establishing the skill of climate field reconstruction techniques for precipitation with pseudoproxy experiments*, Climate Dynamics, 1-19.

Gómez-Navarro, J.J., Bothe, O., Wagner, S., Zorita, E., Werner, J.P., Luterbacher, J., Raible, C.C., and Montávez, J.P, 2015. *A regional climate palaeosimulation for Europe in the period 1500-1990 - Part 2: Shortcomings and strengths of models and reconstructions*, Climate of the Past, 11, 1077-1095.

Haylock, M.R., N. Hofstra, A.M.G. Klein Tank, E. J. Klok, P.D. Jones, and M. New, 2008. *A European daily high-resolution gridded data set of surface temperature and precipitation for 1950-2006*, J. Geophys. Res., 113, D20119.

Jerez, S., J. P. Montavez, J.J. Gomez-Navarro, P. Jimenez-Guerrero, J. Jimenez, and J.F. Gonzalez-Rouco, 2010. *Temperature sensitivity to the land-surface model in MM5 climate simulations over the Iberian Peninsula*, Meteorol. Z., 19(4), 363-374.

Jerez, S., J.P. Montavez, J.J. Gomez-Navarro, P.A. Jimenez, P. Jimenez-Guerrero, R. Lorente, and J.F. Gonzalez-Rouco, 2012. *The role of the land-surface model for climate change projections over the Iberian Peninsula*, J. Geophys. Res., 117, D01109.

Schimanke, S., Meier, H.E.M., Kjellström, E., Strandberg, G., and Hordoir, R., 2012. *The climate in the Baltic Sea region during the last millennium simulated with a regional climate*
*model*, Climate of the Past, 8, 1419-1433.

Hurrell, J.W., Kushnir, Y., Ottersen, G. and Visbeck, M., 2003. *An overview of the North Atlantic Oscillation.* In North Atlantic Oscillation: Climate sig- nificance and environmental impact, Hurrell, J. W., Kushnir, Y., Ottersen, G., and Visbeck, M., editors, volume 134. American Geophysical Union. Geophysical Monograph Series, 1-35.

With kind regards on behalf of the all authors,
Emmanuele Russo

**T 2M**

Clim.          CCLM Anomalies

**PREC**

Clim.          CCLM Anomalies

CCLM

CRU

E-OBS

$^{o}C$          $^{o}C$          $mm/day$          %

**Fig. 1.**

**T 2M**

**PREC**

Clim.    CCLM Anomalies

Clim.    CCLM Anomalies

$^oC$    $^oC$    $mm/day$    $\%$

**Fig. 2.**

**Fig. 3.**

**T 2M**

**PREC**

$\sigma_{CCLM}/\sigma_{CRU}$

$\sigma_{CCLM}/\sigma_{E-OBS}$

**Fig. 4.**

**T 2M** **PREC**

$\sigma_{CCLM}/\sigma_{CRU}$

$\sigma_{CCLM}/\sigma_{E-OBS}$

0  1  2  3  4

**Fig. 5.**

**CCLM**

**POLLEN**

DJF

JJA

−0.6    −0.4    −0.2    0.0    0.2    0.4    0.6

**Fig. 6.**

**MSLP**

**T2M**

R=0.62

37.94%

19.75%

R=0.65

14.74%

25.04%

**Fig. 7.**

MSLP        T2M

R=0.40

38.72%       11.78%

R=0.59

21.91%       22.48%

−0.03 −0.02 −0.01 0.00 0.01 0.02 0.03    −0.03 −0.02 −0.01 0.00 0.01 0.02 0.03

**Fig. 8.**

**1st PC**

**2nd PC**

DJF

JJA

Years BP

Variance

**Fig. 9.**

**Fig. 10.**

meters a.s.l.

[Figure]

**Fig. 11.**

---

## Author Response (AR1)

In the following text we go point by point through the technical corrections and comments of the two reviewers, additionally considering short comments received by other members of the scientific community, detailing how we dealt with these concerns reported in **Bolt**, and, when necessary, specifying the modifications applied within the revised manuscript in *italic*

Reply to
**1st Reviewer**
*Mid-to-late Holocene Temperature Evolution and Atmospheric Dynamics over Europe in Regional Model Simulations by Russo, Emmanuele; Cubasch, Ulrich cp-2016-10*

- **General Comments**

  **The paper can be broadly divided into three parts: i) validation of the simulation, ii) comparison with reconstructions iii) search for explanations of the disagreements. In all these three elements, I can identify caveats that in my opinion should be improved with different/complementary analyses. I have tried to review them in a comprehensive, yet constructive way, as detailed below. Besides the technical aspects, I think there is room in the manuscript for improvement regarding writing style. It was challenging for me to read and understand many parts of the paper. This is in part due to incomplete information in the captions and the main text, wrong labelling in the figures, and the misleading use of some concepts as observation or validation. The internal structure in the paragraphs is confusing: paragraphs loosely connected, overly short, or in a misleading order respect to the panels in the figures. These issues add complexity that makes the lecture of the paper uncomfortable. Despite this rather negative view, I try to be constructive giving a list of points that develop the aspects that in my opinion can be improved in the manuscript. Note however that this list is not comprehensive.**

We agree with the referee.

*We developed a clearer structure of the manuscript as suggested by the reviewer. We divided the discussion of the manuscript in three main parts: the first based on the validation of the model configuration for present-days, the second one based on the comparison against proxy-data, additionally providing analyses and discussion on the advantages of the use of highly resoluted simulations for the comparison against reconstructions, and the third one in which we will provide explanations for possible mismatches. We also provide improved/complementary analyses accordingly to the referee's comments. In addition, we corrected the manuscript, keeping in mind that, as mentioned by the referee, the comparison against proxy data is by no means a validation. We provide such comparison in a clearer way, cautious on the use of proper terminology. Additionally, as detailed in the comments below, we tried to improve the grammar of the manuscript and its presentation, in order to better tie together the entire discussion.*

1. **Abstract/Introduction:**
   **L1-3: In the first line, is always been mentioned is grammatically wrong. Despite that, it sounds a bit loose, almost sceptical. Is it an important factor or not? This ambiguous tone of the first sentence of the abstract is manifest through the whole manuscript. By the way, the authors do not make an attempt to demonstrate that this is indeed the case for these simulations. More on this below.**

   We agree with referee. Further details on additional analysis we conducted with respect to the previous comment are presented in the comments to the second referee.

   *We reformulated the sentence according to additional analysis we provided within the text. In the revised version of the manuscript we present a section in which we conduct a detailed comparison between the models at different resolutions and proxy-data, elucidating possible advantages of Dynamical downscaling. We implemented our discussion and the manuscript consequently, following a common suggestion of both the authors.*

   **L5-6: The paper is somewhat optimistic regarding the use of for the first time. It is true that, as far as I know, there are no other set of time slide simulations. However there are various high-resolution simulations for the last millennium for Europe. Actually there exists at least one transient simulations for the last two millennia, in fact driven by the same**

**ECHO-G run used by the authors of the manuscript. The authors should not ignore such previous, yet scarce efforts in this topic in the intro, but also the discussion of the results.**

We agree with the referee.

As the the referee mentioned, there are other paleo-simulations for Europe for mid-to-late Holocene. Nevertheless, such simulations often investigate only a time slice or do not cover the entire mid-to-late Holocene. Even if they do so, as the case of the ECHO-G simulation used for this study, their low resolution is often mentioned as one of the possible reasons for the disagreement between model results and reconstructions ([Fischer & Jungclaus 2011];[Bonfils et al. (2004)]). With the previous sentence, we wanted to highlight the fact that no previous simulation exists for Europe, at such high resolution, covering different time-slices of mid-to-late Holocene. Our optimism, in this sense, regards the fact that these simulations could contribute in clarifying the debate on models and proxy disagreement.

*We modified the previous sentence accordingly to the referee's comment. Additional discussion and more references*
*(e.g. [Strandberg et al. (2014)],[Schimanke et al. (2012)],[Braconnot et al. (2007a)],[Braconnot*
*have been added within the text.*

**L6: In line 6, validation is used in a wrong context. The model is validated normally against observations. But you can not validate the model looking at a reconstruction. Neither you validate a reconstruction looking at a simulation. You can only compare them, and try to gain insight through the disagreements. The use of validation in this wrong context is spread through the manuscript and should be avoided.**

Accordingly to the referee's comment, we think that the terminology previously employed was incorrect.

*We corrected the term "validation" with "comparison" here and throughout the manuscript, when referring to the comparison with proxy data.*

**I think at least the first four paragraphs can be safely merged.**

We agree with the reviewer.

*We merged such paragraphs reformulating them in a more concise and clearer way.*

**L39-41: it is argued that then changes in solar irradiation were negligible, and latter than we expect that such changes would imply relevant variations.... It sound contradictory.**

In this sentence our goal was to highlight the fact that during mid-to-late Holocene yearly variations in insolation, over northern latitudes in general, were neglicible when compared to the seasonal variations. The latest are expected to imply "relevant changes in the seasonal values of surface variables".

*We re-formulated this period in a more comprehensive way.*

**L42-45: The paragraph in lines 42 to 45 is made out of a single sentence, which is too long. Still, the paragraph itself is short and can be merged with the former. Further, such sentence demands references.**

We agree.

*We reformulate the paragraph and join it with the former. We also added further references (i.e. [Cheddadi et al. (1997)],*
*[Bonfils et al. (2004)],[Braconnot et al. (2007a)],[Braconnot et al. (2007b)]).*

**In the paragraph starting in line 89, some examples of RCM simulations in palaeoclimate applications are outlined. It's strange to see that no simulation for Europe is referred. Examples of such simulations are: Gmez-Navarro et al. (2011, 2012, 2013, 2015a, 2015b) and Schimanke et al. (2012).**

We agree.

*Apart from the ones suggested by the referee, we also take into consideration, in the revised version of the manuscript, other works in which high resoluted paleo-simulations for Europe were performed (i.e. [Strandberg et al. (2014)];[Renssen et al. (2001)]).*

2. **Model Validation:**

**It is not clear how long is the control period used in section 3.1. The only hint is the label in Figure 2, 1990-2000. Is that the case? It should be clearly stated, not only in the main text, but also in the caption of the figure. Actually, the length of this period is CRITICAL for the model evaluation, a fact**

**that is not acknowledged in the discussion of the results. A 10-year period of a GCM simulation is strongly populated with internal variability. Under this scenario, a comparison with observations is tricky. The model could be by chance going through a cold or warm phase, which would have a strong impact in the validation, at least in the way it has been established in the paper, focused on mean values. In this sense, the validation does not look at important aspects such as the variability. How is the variance reproduced by the model? I'm not sure due to the short length of the simulation, but it could make sense to look at the variability modes of temperature and precipitation.**

As indicated by the referee, the control run is 10 years long and covers the period 1991-2000. We are aware that the length of this simulation is "CRITICAL" for models' evaluation. Unfortunately, due to computational reasons, we were not able to cover a longer period. Additionally, realizing that we have not been properly explicit in the description of our experiment, we want to clarify that the regional simulation was driven by the ERA-Interim reanalysis dataset and not by the GCM, as mentioned by the referee. We will implement our text accordingly, with the main goal of better specifying such technical details. Since the main goal of the present-day experiment is to test whether the changes applied to the model routine, for this particular case of study, allow to obtain reliable results in comparison to the outcomes of other studies, we think that the validation focusing on mean values is a useful tool in this context. Nevertheless, we also conduct now an analysis of model and observations variability that we aim to provide as supplementary material in the revised manuscript. Additionally, we now also consider the E-OBS dataset ([Haylock et al. (2008)]) as a benchmark for the validation of our results, and present the mean climatology of temperature and precipitation, for both winter and summer, as reproduced in the three datasets.

In addition to previous conclusions based on the bias of seasonal mean, the new analyses show that, the model is able to reproduce, with a certain degree of accuracy, the climatology of the observations. Additionally, the analysis of the standard deviation (Fig.S1 and Fig.S2) shows that the area with the larger bias are the ones where the model is not able to correctly reproduce the variability of the observations, in particular for precipitation.

*In the revised version of the manuscript we added more informations on the length of the simulation throughout the text. We also added such specification in the caption of Fig.3, Fig.4 and Fig.5. We additionally acknowledge now the choice of performing a 10-years run within the text. We replaced the previous analyses of precipitation and temperature with the one presented in Fig.3 and Fig.4 of the new manuscript, and developed our discussion accordingly. A more detailed description is provided in the captions of such figures.*

**I do not think the choice of target for the validation is the best one. Using ERA Interim for the validation of precipitation in particular is a bad idea, since it is not constrained by precipitation observations, so there is no warranty that this dataset is bias free. I think it would be wiser to use the E-OBS dataset, which was developed specifically for the validation of RCMs in Europe (Haylock et al. 2008).**

We agree with the author that the ERAInterim Reanalysis is not the best choice for model's validation, in particular for precipitation. For this reason, as mentioned above, we now compared the model's results against the CRU observational dataset and the E-OBS dataset.

**The way the similarity between the model and the observations is presented is a bit confusing to me. When the difference between two normally distributed variables is shown, the standard and intuitive approach, which steams from the application of the Central Limit Theorem, is to apply a t-test. The KS test is more suited for testing the shape of PDFs when the mean is know to be the same. For example if two dataset have the same mean, but different variance, the figure would show null bias (yellow colour here), but still the test would produce significant differences, which is misleading for the reader.**

For the comparison of mean values we agree with the referee that a T-test is better suitable for our purposes.

*We now perform the Student's T-test for the validation of the considered variables (Fig.3; Fig.4 and Fig.5).*

**I think the maps showing precipitation difference are not very useful. A difference of 5 mm/day might be huge or tiny depending on the mean precipitation. I think changes in precipitation are more meaningful when shown as perceptual**

**deviations with respect to the mean.**

We agree with the referee on the fact that changes in precipitation are more meaningful when shown as percentual deviations with respect to the mean.

*In the new maps (Fig.3 and Fig.4), we now present precipitation biases as the percentual deviations from the observations values.*

**It is mentioned that the dots indicate grid where differences are significantly not different. That's not exactly true. They indicate areas where the null hypothesis of the data being sampled from the same underlying distribution could not be ruled out, i.e. where they are not significantly different.**

We agree. The previous sentence was not totally correct.

*We now reformulate the sentence as follows: "the dots show the points where the null hypothesis of a Student T-test, at a significance level of 5%, assuming that the data being sampled could be drawn from the same underlying distribution, is not rejected".*

**I agree with the hypothesis used to explain the model deficiencies in Souther Europe regarding soil- atmosphere feedbacks. The particular role of these processes in RCM simulations in areas with strong water deficit was investigated in detail by Jerez et al. (2010, 2012).**

*We proposed additional references within the new manuscript as the ones indicated by the reviewer and listed at the end of this text.*

**L206: reads These findings CONFIRM that... are MOST PROBABLY.... This is an example of doubtful and confusing sentence that should be avoided.**

We agree. The previous sentence was doubtful as indicated by the reviewer.

*We corrected our sentence accordingly. Our analysis in fact confirms that model performances are influenced by its scarce capacity to reproduce soil-atmosphere exchanges correctly. This has consequences on both temperature and precipitation (particularly in summer when the biases are more pronounced) presenting a similar pattern of anomalies.*

**Maybe is worth to mention that generally the model skill resembles that identified in similar simulations for Europe (Schimanke et al. 2012, Gmez-Navarro et a. 2011, 2013).**

We agree. Although a few works that generally propose similar model skills have been already considered within the discussion paper, it is reasonable to include additional bibliography, that would help in strengthening our conclusions.

3. **Comparison with Pollen Reconstructions:**

   **As pointed out above, this comparison is by no means a model validation. This should be made clear in the wording. As such, all sentences like CCLM performs well should be modified. The maps in Figure 5 are calculated as means with respect to which period?**

   We are aware of the incorrect terminology employed, as already highlighted before.

   *We corrected this period substituting the term "validation" with "comparison", being the pollen reconstructions not an observational dataset. We also aim at using better expressions in order to indicate the good or the bad agreement of the two datasets. We now modified Figure 5 of the discussion paper (also accordingly to the comments of the 2nd reviewer). In the revised manuscript, we present the maps of of the anomalies as represented in the the two datasets, calculated for every investigated period with respect to the pre-industrial times. We also propose to accompany them with the corresponding maps of the pollen-based reconstructions uncertainties. Please refer to the 2nd reviewer response for further details.*

   **In Figure 6, error bars are provided for the pollen data, but not for the simulation. I'm aware it is not easy to stablish them. However, such errors/uncertainties should not be neglected in the discussion of the results. The model has deficiencies that introduce systemic biases. But on top of then, there are non systematic biases introduced by unpredictable internal variability. This factor might lower or rise mean temperature in the simulation quite significantly, as pointed out by Gmez-Navarro et al. (2012) in a very similar scenario. Thus, this should be discussed at least qualitatively in this part of the text.**

   It has been a choice of the authors not to include the model's uncertainty interval to the plots of Figure 6 of the discussion paper.

Since the uncertainties of the 25 years CCLM simulations are considerably smaller than the ones of the proxy-reconstructions, we think that neglicting them, in this figure, was an appropriate choice.

*Nevertheless, following the suggestion of the reviewer, we added such considerations within the manuscript. Additionally, we proposed a new analysis of the trends of temperature in which the uncertainties are taken into consideration by means of a weighted least squares method.*

**Many conclusions are drawn from Figure 6 regarding matchings of trends. I'm not sure at what extent such conclusions have any statistical significance, since in almost all cases the simulation lies within the uncertainty of the reconstruction. Having an almost perfect match between the reconstruction and the simulation is still perfectly possible within the range of uncertainty of the reconstructions.**

Following the referee's comment, we realyzed that the computation of the mean and of the relative uncertainties presented in Fig.6 of the discussion paper should be re-performed. In particular, the plots we previously proposed and the conclusions we have drawn from them had no statistical significance. In fact, in a first place, we simply calculated the error as a mean of the provided uncertainty for every point. Realizing that this procedure is not correct, we tried to be more cautious with our analyses.

*We present now new maps in Fig.9, representing the trends of seasonal means of 2 meters temperature calculated, for every grid box, by means of the weighted least squares method. The points where the trends are not significant, according to a F-test at a significance level of 10%, are additionally masked out. We provide again more details in the caption of the figure. We think that these maps are better suitable for our discussion. In fact, they are statistically more robust, allowing to consider trends and relative uncertainties for every grid box and time slice, resulting in a better suitable benchmark for the comparison against the pollen-based reconstructions or other proxy datasets. In the revised version of the paper we replaced Fig.6 and Fig.11 of the former manuscript with Fig.9.*

**Something I miss in this analysis here is the GCM simulations used to drive COSMO. I wonder how the ECHO-G and later ECHAM5 compares also with reconstructions. Is the RCM adding anything relevant to these simulations? If the**

**answer is certainly yes, then the use of the RCM is fully justified and the paper would gain interest. If the answer is mostly no, it would be still interesting, since it would imply that the many GCM simulations available for the last millennia are still relevant at rather regional scales. I'm sure the PMIP community would be very interested in answering this question.**

A similar comment has also been addressed by the 2nd reviewer. We refer to the answer to his comment as an exhaustive response to this point. As suggested by the referee, we think that answering this question would definitely strengthen the paper.

*In the revised manuscript we added a section in which the possible advantages of highly resolved simulations for the comparison of change in 2 meters temperature against proxy reconstructions is investigated.*

4. **Interpretation of Paleo Records:**

**Generally it was difficult to follow the arguments in this section. It would significantly help to label the maps as Fig 6b, Fig 6c, etc. and use such labels extensively through the manuscript. In this regard, the discussion of the results starts with summer, whereas the first row shows winter. Small inconsitences like this, although non critical for the scientific message, have a dramatic impact in the reading pace.**

We agree.

*In order to make the manuscript more easibly readable we labeled the maps accordingly to the referee's suggestion. We also corrected the order of the seasonal analyses within the text.*

**The EOFs for MSLP are shown and used in the discussion. They are used to argue regarding NAO and SNAO, for instance. I'm not totally comfortable with that, since the NAO is defined as the leading pattern for a spatial window that is not that of the RCM. This explains in my opinion why the NAO pattern does not stand out as the leading mode in winter, and second mode in summer just resembles the SNAO pattern. I think a more orthodox approach would be to calculate the EOFs within the GCM, in a window that properly encompasses the North Atlantic. This is justified since the**

**large scale circulation is fixed by the GCM, and thus the NAO simulated should be consistent with the climate variability within the RCM domain. Hence, such patterns could still be used to discuss about regional variability within the RCM domain.**

We agree with the referee's comment.

*We now conduct the analysis of MSLP anomalies of the ECHAM5 simulations in order to properly consider a spatial window that encompasses the entire North Atlantic region. We select the region in between 90W and 40E and in between 20N and 80N, as defined in [Hurrell et al. (2003)]. This would allow us to infer about changes in the NAO and other atmospheric circulation patterns characteristic of this region. The results are shown in Fig.10 and Fig.11 of the new manuscript. Since the RCM large scale circulation is "dictated" by the GCM, we reasonably think that such results can be used to argue about regional variability within the RCM. We modified the discussion within the revised manuscript accordingly to the new analysis.*

**Line 255 reads In summer the first EOF shows that the model reproduces similar conditions in atmospheric circulation between the mid-Holocene and pre-industrial times. I do not understand how that conclusion is drawn from the map in Figure 8.**

We propose to modify the previous sentence accordingly to the new analysis presented above. Since the investigation area is different now, the results of the EOF analysis changed. Additionally, as will be elucidated in the next points, we conduct in the new version of the manuscript, in substitution of the EOF analysis, a canonical correlation analysis (CCA) of MSLP and T2M.

*We modified the discussion within the revised version of the paper accordingly to the new analysis, being more cautious about arising risky conclusions as the one spotted out by the referee.*

**In page 8 the wording observed is used in various sentences, and it's not fully clear what is meant (most likely respect to the simulation, but it could also be the reconstruction). I think simulated is more appropriate and precise.**

As highlighted in previous points we agree with the referee.

*We corrected the sentence accordingly in the revised manuscript.*

Some inferences about the clearness of the sky are made which are based in indirect evidence such as EOF analysis. I think it is not necessary to make such risky affirmations. We have direct information that can tell us exactly how cloudy the simulated climate was. After all, in the simulation we can check directly variables such as cloud cover, which give a direct measure of what is being argued. I would go for a direct measure whenever possible, as it is the case. Similarly, in the paragraph between lines 277 and 279 (and Fig. 11) the more pronounced positive phase of the NAO can be directly tested within the GCMs, rather than indirectly inferred through a map of temperatures.

*We modified the previous sentence within the revised manuscript, accordingly to the fact that, even if the SNAO shows a trend that in this case is positive troughout the mid-to-late Holocene, such trend is not significant and presents high variability. We propose to review our previous discussion and to avoid any conclusion on the trend of cloud cover due to the high variability of the emerged pattern throughout the investigation time . As already mentioned, we also preferred to merge Fig.11 together with fig 6 of the discussion paper, considering also summer analysis.*

Finally, I think there are more powerful statistical tools than the one used here to study the co- variability between temperature and MSLP. Canonical Correlation Analysis could be used to derive relations between the variability of MSLP and temperature, and it would produce a picture of such co-variability more robust that the one provided by maps in Figure 10, for instance. An example of the application of such a tool in a very similar context is Gomez-Navarro et al. (2015b)

We agree.

*We now investigated the covariability of MSLP and temperature by means of Canonical Correlation Analysis and present the results of such analysis in Fig. 10 and Fig.11 of the revised manuscript. We referred to the study of [Gmez-Navarro et al. (2015b)] as a good example of application of such method for the investigation of the relations between atmospheric variability and temperature. For our analysis we employed the method of Barnett and Preisendorfer 1987, for*

*which the data are pre-fildered by a EOF analysis before applying the CCA, retrieving only the principle components that explain most of the variability. We have to acknowledge the fact that, realizing that the computations relative to the CCA we provided in the public response to the referee presented some errors, we proceeded to new analyses, the results of which are shown in Fig.10 and Fig.11 of the revised manuscript. Such figures result different from the former ones. Aware of the mistake we apologize for such inconvenient.*

5. **Comments regarding Figures**

   **Figure 1: The colour scale shows everything below 1000 meters as green. I think a palette with stronger contrast could be chosen.**

   We improved the previous plot accordingly to the referee's suggestion.

   *We moved the modified figure to Fig.2 of the revised manuscript.*

   **Figure 2: The reference period should be stated in the caption. I think the limits of the palette can be adjusted to better span the range of temperatures.**

   We agree.

   *We added the reference period within the caption and further details. We also provided, in the same figure, the results of additional analysis and improved the palette in order to better span the range of temperature. For better developing the discussion within the paper, we preferred to present together, in the revised manuscript, the plots of the analyses of temperature and precipitation, for winter, in Fig.3 and, for summer, in Fig.4.*

   **Figure 3: The colour palette provides barely any contrast all. Everything is yellow in the maps.**

   *We modified the plot accordingly to the previous point.*

   **Figures 4 and 5: Same comments as in former figures**

   *Figure 4 has been adjusted accordingly to the referee's comment. Figure 5 of the discussion paper has now been modified accordingly to the comments of the 2nd referee. The new plots are presented in Fig.7 and Fig.8 of the revised manuscript.*

   **Figure 6: Please label panels as 6a, 6b, etc. I do not think using colour in the caption is an orthodox approach. Note that the caption does not agree with the order of panels.**

**First row does not show North, but it is the first column which does, etc.**

We agree.

*We now label the panels of the new map presented in Fig.10 of the revised version of the manuscript as 10a,10b, etc., as suggested by the referee. We also avoid using coulours in the caption. We also modified the order of the captions, accordingly to the figure.*

**Figure 7: I can barely see the numbers and labels in the figures in the right.**

*We modified this figure in order to make it clearer and more easily readable. Accordingly to a comment of the second referee, we propose to move this picture to section two of the revised paper (now Fig.1).*

**Figure 8: I think the label with the loading can be moved to inside the maps. This would allow to put the maps closer together, which would allow to make maps larger and more readable. The latter comment can be applied to almost all figures.**

We agree.

*We moved the loadings inside the maps (Fig.11 and Fig.12 of the revised manuscript). We also applied similar modifications to all the pictures in order to make them larger.*

**Figure 9: Please label panels to indicate which represent EOF1 etc. Where are the units? Either the EOF or the PC carries the units, in this case pressure. I guess they are included in the EOF patterns in Figure 8. If so, please label the palette accordingly.**

*Realizing, following the referee's comment, that we were not precise in our previous discussion, we propose to add further details in the caption of this figure, with more specification regarding the units and the analysis we conducted.*

Reply to
**2nd Reviewer**
*Mid-to-late Holocene Temperature Evolution and Atmospheric Dynamics over Europe in Regional Model Simulations by Russo, Emmanuele; Cubasch, Ulrich cp-2016-10*

1. **Main Comments**

   **The grammar and spelling can be much improved. There are many long sentences that are hard to read. I have indicated a few below. I strongly suggest to have the text thoroughly checked by a native English speaker.**

   We agree.

   *We improved the structure and the grammar of the paper in order to make it more easily readable. We also shortened long sentences and expressed complex periods in a more concise and robust way.*

   **I propose to compare the results of COSMO-CLM to the results of ECHAM5. The latter results have already a relatively high spatial resolution (T106 or 1.125x1.25 degr) compared to previous GCM studies. This resolution is actually close the resolution of the reconstructions (1x1 degr). In the manuscript, the authors have regridded (up scaled) their regional climate model results from 0.44x0.44 degree resolution to 1x1 degree to make the comparison in Fig 5. It would be interesting to see to what extent the COSMO-CLM produces a better match. Is it, from a paleoclimate perspective, worthwhile to make the considerable effort to nest the regional model in the high-resolution GCM results? Or do both models produces very similar results? In my view, addressing these questions would strengthen the paper. To make room for such a comparison, Figures 2, 3 and 4 could be moved to the supplementary information, as these figures do not directly concern the core topic of this study (mid-to-late Holocene temperatures and atmospheric dynamics).**

   According to the [IPCC(2007)] report: "Paleoclimate data are key to evaluating the ability of climate models to simulate realistic climate change". In particular, since the details added by high resolution models can help in the interpretation of proxy data that are often influenced by processes taking place on smaller scales than the ones

resolved in coarser models, they are considered a particularly suitable tool for paleoclimate studies.

Within this context, in our discussion we try to highlight the importance of using high resolution models, and in particular Regional Climate Models, for the simulation of past climate change. Aiming at investigating the value added by highly resoluted simulations for the comparison of near surface temperatures against proxy-reconstructions, we follow a two steps approach:

(a) Firstly, we conduct a qualitative analysis of the simulations performed with three models at different resolutions in order to detect visible differences in the reproduced signals.

(b) Secondly, we employ a quantitative approach in order to estimate the skills of the RCM, in comparison to the driving GCM, in reproducing the same changes in temperature during mid-to-late Holocene as derived from proxy-reconstructions.

As a benchmark for such comparison we use the pollen-based temperature reconstructions of [Mauri et al. 2014]. In this way we aim at establishing whether the representation of smaller scale processes and improved orographic features of the region of study, could lead to results that are in better agreement with the mentioned proxy-reconstructions.

In Fig. 6 of the revised manuscript we present the anomalies of summer and winter seasonal mean temperatures between 6000BP and the Pre-industrial period, as reproduced by the different models. From these maps we first notice as, in both the seasons, a similar signal of climate change is present for all the simulations. This is expected, beeing, in every case, the data constrained by the coarser resoluted models. Nevertheless, while the higher resoluted simulations allow to catch a warmer bias over Northern Europe in winter, also present in the proxy data, the ECHO-G does not show such behaviour. Additionally, the land-sea area in the ECHO-G is considerably different than the ones of the other models. Regions such as Southern Spain and the Black sea area, Italy and Scandinavia are partly or completely masked-out in this case.

Consequently, we reasonably suggest to focus further analyses on the comparison between the ECHAM5 and the CCLM results. In both seasons additional details are easily detectable in the CCLM pattern. The

coastline is also better reproduced in this case, resulting in more suitable informations for possible comparison with proxy-data. Nonetheless, the CCLM shows better defined patterns as a consequence of higher resolution, being able to discriminate higher spatial variability.

In the successive step, we try to quantify how better the CCLM reproduces the reconstructed temperatures in comparison to the ECHAM5. Under the mentioned considerations, we use a similar approach to the one employed by [Zhang et al. (2010)] and based on the work of [Goosse et al. (2006)]. After upscaling the RCMs results and interpolating the ECHAM5 ones on the reconstructions grid, we introduce a Cost Function defined as:

$$CF^k_{mod} = \sqrt{\frac{1}{n} \sum_{i=1}^{n} \omega_i{}^k (T^k_{rec,i} - T^k_{mod,i})^2} \tag{1}$$

where $CF^k_{mod}$ is the value of the cost function for each considered time slice $k$ of mid-to-late Holocene, and each model $mod$ . The parameter $n$ is the number of the reconstructions grid boxes, $T^k_{rec,i}$ the reconstructions temperature at every location $i$, while $T^k_{mod,i}$ is the correspondant temperature of the model simulation. The parameter $w^k_i$ is instead introduced for considering the uncertainties of the reconstructions at every location and time period. Its value is given by:

$$\omega_i^k = \frac{1}{(SE_i^k)^2 + 1} \tag{2}$$

where $SE_i^k$ represents the standard error of the pollen-based reconstructions at every grid point and every timestep k. In this way reconstructions with higher uncertainties will contribute less in the calculation of the Cost Function. We neglected models uncertainties since they are considerably small ($\sim 0.01^o C$) in comparison to the reconstructions ones, similarly to [Goosse et al. (2006)].

The values of the CF for the two models are provided in Tab.1 and in Tab.2.

As we can notice, even if not particularly large differences are present, the Cost Function computed for the CCLM is in almost all the cases lower than the ECHAM5 one. In particular the CCLM results are, in

some cases, closer by almost 10% to the reconstructions. It is important to mention that the scale considered in our analysis is closer to the resolution of the ECHAM5 than the one of the CCLM. As suggested by [Di Luca et al. (2015)], given that the main difference between the GCM and the RCM is related with their horizontal resolution, it seems natural that the results depend on spatial scale of the analysis.

Additionally, is key to state that the evinced results are relative to this case of study and other comparisons should be performed, considering different couples of RCM-GCM, in order to derive more robust conclusions on the suitability of higher resoluted models for the comparison against proxy-reconstructions.

Nonetheless, the motivation behind producing higher resolution climate simulations is not only related to scientific arguments of the type described above. From a different perspective, such results, due to the greater level of detail, could be preferable for applications in studies in which human adaptation or environmental response to past climatic changes would be investigated. The need for climate information at very fine scales, for application such as archaeology or vegetation reconstructions, hence constitutes a strong incentive to perform higher-resolution climate simulations ([Di Luca et al. (2015)], [Rummukainen (2016)]).

In conclusion, the evinced results and the proposed discussion, give us concrete motivations for the choice of conducting RCM simulations for this particular case of study. Nevertheless, we keep Fig.2, Fig.3 and Fig.4 of the discussion paper within the revised version of the manuscript, as representing a satisfactory test for the reliability of the chosen model setup, they could be suitable for other studies conducting paleoclimate simulations for the region.

*In the new version of the manuscript we added a section based on the presented analyses accompanied by detailed and pertinent discussion.*

**The left column of Fig 5 presents maps of the winter and summer temperature anomalies (model minus reconstructions), "averaged over all the mid-to-late Holocene time slices". It is not clear to me what the authors have actually done here. Have they first averaged the maps of the different time-slices for the model and the data, and then calculated the model-data anomaly? Or have they calculated the trend between 6000 and 200 BP in both model and data, and then made a**

map of the difference between the two methods? The caption suggests that they have applied the first method, but in my view this would only be meaningful if the anomalies are more less constant through time, which is clearly not the case (see Figure 6). Since the trends from 6000 to 200 BP seem approximately linear in both model and data, it would make more sense to compare maps of these trends or to show maps for different time slices. Figure 11 actually shows linear trend maps for both the model and the reconstructions, but only for DJF. It is unclear to me how to relate Figure 11 to Figure 6. Figure 11 seems to indicate a pollen-based linear warming in Southern Europe of mostly less than 0.4 C, while Figure 6 shows a warming trend for the pollen-based reconstructions of 1 C for Southern Europe. In addition, the pollen-based cooling trend in Figure 6 of more than 2 C does not match Figure 11 which shows a much smaller cooling trend. Is there an inconsistency between Figure 6 and 11, or have I missed something? Please clarify.

In the previous analysis, Fig.5 was obtained by simply averaging the anomalies over all the time-slices. The same procedure was also applied in order to obtain a map of the average uncertainties. Following the considerations of the referee, we realized that such approach was not totally correct and we re-performed our analysis consequently.

*In Fig.7 and Fig.8 of the revised manuscript, we computed the seasonal anomalies of 2 meters temperature between the CCLM and the pollen-based reconstructions for every single period of time. We additionally provided, together with the anomalies, the respective pollen-based reconstructions uncertainties. This choice is reasonable since the uncertainties maps could result useful in the interpretation of the mismatches arising between the two datasets. Additionally, we are now considering a new approach for the investigation of seasonal trends. We recomputed figure 6 of the old version of the manuscript taking into consideration, this time, the uncertainties in both the datasets (for more specifications please refer to the first referee response). Here the new plots (Fig.9) are similar to Fig. 11 of the discussion paper, showing this time both winter and summer trends. Only the area where the trends are significant, according to a F-test at a significance level of 10%, are shown. Additionally, such trends are calculated by mean of*

*a weighted least squares method, allowing to take into consideration, as said, the uncertainties of the two datasets. Since the changes in both the datasets are not homogeneous over the region, we think that these maps should be more appropriate than the previous ones based on regional means. We want to highlight, relatively to the referee's comment, that the new maps do not show values of changes in temperature. Rather they show the slope of the trend associated to every grid box.*

**The right column of Fig. 5 shows the uncertainties in the pollen-based temperature reconstruction. How were these maps constructed? According to Fig. 6, these uncer- tainties are not constant through time, so simply averaging the errors for the different time slices is not informative here either. Please clarify.**

Please refer to the previous point.

**For the summer in Southern Europe, the model and the reconstructions show opposite trends: cooling in the model and warming in the reconstructions. The authors provide an explanation for this model-data mismatch that is based on the warm bias of the model in S Europe due to the underestimation of evaporation in summer. However, the mismatch may also be explained by uncertainty in the pollen-based reconstructions in S Europe. Paleoclimate reconstructions based on pollen rely on the assumption that changes in the vegetation were driven by the parameter to be reconstructed (i.e.summer temperature). In the Mediterranean region, vegetation distribution is mainly limited by effective precipitation, rather than by summer temperature (e.g. Osborne et al. 2000). It would therefore be good to discuss the associated uncertainties in the methodology of the pollen-based reconstructions and to mention Holocene temperature reconstructions that are based on other proxies. For instance, summer temperature reconstructions from the S Europe domain based on Chironomids, show a clear Holocene cooling (Heiri et al. 2015; Toth et al. 2015) that actually support the presented modelling results. In addition, Holocene SST reconstructions from the Mediterranean Sea show a similar cooling trend (e.g. Marchal et al. 2002). The discussion section should be extended accordingly.**

The choice of the dataset of [Mauri et al. 2014] has been done for several reasons. First of all, it allows to perform a comparison against model results over most of the simulations domain, considering different variables (even if we only focus on temperature in our discussion). Then, it covers exactly the same time-slices of our model simulations. No other dataset has this temporal and spatial coverage at such high spatial resolution. Additionally, the robustness of the data has been thoroughly tested, in [Mauri et al. 2014], against other proxies (including chironomids, $\delta 18$ O from speleothems and lake ostracods, bog-oaks, glacio-lacustrine sediments, wood anatomy and other pollen reconstructions based on different reconstruction methods) leading to satisfactory results. Nonetheless, similar pollen-based climatic reconstructions have been extensively employed in other data-model comparisons, and, most recently, for the evaluation of the PMIP3/CMIP5 climate models included in the last IPCC report (Stocker et al. 2013, Harrison et al. 2015).

As the referee mentioned, different studies already criticized the use of pollen-based data for reconstruction of temperature over the Mediterranean region, claiming that the vegetation distribution is mainly limited by effective precipitation, rather than by summer temperature (e.g. [Osborn et al. 2000]; [Renssen et al. 2009]). In response to such critiques we want to refer to a detailed comment provided by Basil Davis, and attached to the discussion paper.

According to the aforementioned reasons, and additionally supported by the explanations given by B. Davis in his comments, we think that the employed pollen-based reconstructions can be considered a very reliable source for the main goals of our paper.

Nevertheless, in accordance to the referee's comments, in the new version of the manuscript we will provide further discussion on the uncertainties in the methodology of the pollen-based reconstructions and specify more details on the reliability tests conducted by [Mauri et al. 2014]. Since the comparison against indipendent and different proxies has already been performed by [Mauri et al. 2014], we feel that such analysis could be omitted from our manuscript. Additionally, the previous analyses of mid-to-late Holocene temperature evolution were misleading. In fact, simply considering regional means, they did not allow to have a proper overview of the trends at different locations, possibly resulting in a mismatch in the comparison against other proxies. The new maps presented in Fig. 4 show now a more heterogeneous behaviour, and are in better agreement with other indipendent reconstructions such as the one of [Heiri et al. 2015], mentioned by the referee, for which summer temperatures over the Alpine region were characterized by a decreasing trend during mid-to-late Holocene.

**In the discussion, the results should also be compared to other modelling studies that focus on the mid-to-late Holocene climate. Do the new results presented here confirm earlier findings? How do the seasonal trends and 6k-0k anomalies compare to that of other models (e.g., PMIP3)? What do other Holocene modelling studies say about changes in atmospheric circulation over Europe and the North Atlantic basin?**

We agree.

*We present, in the revised version of the manuscript, a section in which our results are compared against other studies. In particular, we focus our analysis on the anomalies between 6000BP and the pre-industrial period, performing a direct comparison against the outcomes of 12 models from the PMIP3 experiment. We compute the regional means for two regions over Northern and Southern Europe for al the datasets. We include such values in two tables, provided as supplementary material in the revised manuscript. The main features arising from such analysis are, a common positive bias over Southern Europe in summer, and the failure to properly represent winter anomalies in both the regions. We aim to implement and develop our discussion accordingly.*

**Conclusions: The conclusions should be made less descriptive / more quantitative. The paragraph starting on line 296 does not contain conclusions and can be removed. Please explain on Line 310 what atmospheric circulation configuration is meant here.**

We agree.

*We make our conclusions more quantitative. According to the new analysis presented here and as a response to the 1st referee, we aim at extending our discussion and develop our conclusions in a more*

*concise and robust way.*

2. **Minor Comments:**

**Line 26: I suggest providing a more accurate definition of climate models**

We agree.

*We tried to develop a more detailed description of the climate models.*

**Line 34: "orbital parameters". I propose to use astronomical parameters instead, since obliquity is not a parameter of the Earths orbit.**

We agree.

*We changed the term "Orbital" in "Astronomical".*

**Line 37: Please rephrase this sentence, as it is not easy to read**

We agree.

*We rephrased the highlighted sentence accordingly to the referee's comment*

**Line 43: "solar forcing". Usually, "solar forcing" is used to describe changes in solar activity as opposed to astronomical forcing that reflects changes in insolation due to changes astronomical parameters. To avoid confusion, I suggest using astronomical forcing here.**

We agree.

*Aware of the mistake, we corrected the term "solar forcing" with "astronomical forcing".*

**Line 46: In my view, this sentence does not introduce the reader to the paragraph, so I propose using a different topic sentence.**

We agree.

*We modified this part in order to better connect it with the following text.*

**Line 57: It is not clear to me what is meant by "hampered climate anomalies"**

We agree.

*We reformulated this sentence. With "hampered anomalies" we wanted to indicate that, the improvement in the reproduction of soil water storage and heat fluxes by climate models, as suggested by[Starz et al. 2013], could lead to a reduction of the biases arising from the comparison with observations. We agree with the referee that the former expression was somehow misleading and we reformulated it in a clearer way.*

**Line 60: typo, atmopshere**

*Corrected in atmosphere.*

**Line 60: "not being able to reproduce correctly the reconstructed data over the entire region". Please clarify. Was the model too cold or too warm? What was the bias?**

We agree.

*We extended the previous period with further details, referring to the results of [Fischer & Jungclaus 2011]. In particular, their results presented only a weak shift to a positive phase of the NAO at mid-Holocene in Winter, resulting in colder conditions over Northern Europe and warmer over Southern Europe with respect to the values of reconstructions. In summer, again, the signal seemed to be mainly driven by changes in insolation, resulting in homogenously warmer conditions at 6000 BP.*

**Line 63: Please rephrase the sentence starting at this line.**

We agree.

*We reformulated the sentence accordingly to the referee's comment.*

**Line 72: " In many cases" What cases, please elaborate. The objectives of the paper should be explained more clearly. On page 3, two objectives are provided. The first objective is to "obtain a better interpretation of the new pollen database..."**

**Why better? What problems have been encountered in the interpretation?**

We agree. The objectives of the paper should be better explained. We try to do so also based on the referee comments and the additional analyses provided in the revision.

[Mauri et al. 2014] presented a possible interpretation of the anomalies evinced from their reconstructions between 6000BP and the pre-industrial period, mainly based on changes in atmospheric circulations. Supported by previous findings, we use our results and the entire mid-to-late Holocene time slices reconstructions of [Mauri et al. 2014], in order to arise plausible interpretations. In particular, while for winter we agree with their interpretation of a more pronounced positive phase of the NAO at mid-Holocene, our findings support different interpretations for summer temperature behaviour.

*We tried to improve our discussion accordingly.*

**Line 105. This first sentence of Section 2 does not provide information on the applied methods. I suggest moving this sentence to Section 1 and to replace it with a topic sentence that introduces the methodology used.**

We agree.

*We moved this sentence to section 1 and modified it in order to better introduce the reader to the employed methodology.*

**Line 128: Berger and Loutre (2002) do not calculate astronomical parameters and is not the appropriate reference here. In their figure they show the values of such parameters, but these are based on Berger (1978), so I suggest to use this reference here.**

We agree.

*We changed the reference accordingly to the referee's comment.*

**Line 133: "only the latest ones". I am not sure what is referred to here. The latter effects?**

In the previous sentence we referred to the changes in insolation due to astronomical forcings.

*We tried to express the period in a clearer way within the revised manuscript.*

**Line 175: "while coloured are the anomalies". Please rephrase and clarify.**

We agree. We wanted to indicate that biases between the two datasets are represented by a chromoghraphic gradient, from blue (when negative), to red (when positive).

*We reformulate the sentence accordingly.*

**Line 194: I propose to use "anomalously warm conditions" here.**

We agree.

*We corrected the sentence accordingly to the referee's suggestion.*

**Line 195: " as a consequence of a wrong conversion of energy towards latent heat." This suggests to me that there is an error in the model code that described this con- version. Is that the case, or is the conversion in principle correct and does the model have a bias in S Europe?**

Being our results consistant with the ones of previous studies investigating present-day conditions (Kotlarski et al. 2014; Jacob et al. 2014; Hollweg et al. 2008), we suggest that the model code describing soil-atmosphere interactions should be reliable. Some biases are present, particularly over Southern Europe, most presumably due to difficulties in properly reproducing soil water storage capacity for this complex orographic area.

**Line 205: typo "teperature"**

*Corrected in Temperature*

**Line 213: I suggest replacing "Pollen" by "pollen-based temperatures"**

We agree.

*We replaced "Pollen" with "pollen-based temperatures" accordingly to the referee's comment, here and throughout the text.*

**Line 214: Please rephrase, as this sentence is confusing. The sentence suggests that Section 3.2 will discuss the results after the validation against Mauri et als data has taken place, while in fact the next paragraph deals with this validation. Besides, I would prefer using evaluation instead of validation here.**

We agree.

*We used "Comparison" as a better suitable word in this case.*

**Line 216: I suggest referring to Figure 1, as this figure shows the boundaries of the two domains.**

We agree.

*We modified Figure 1 accordingly to the new analysis we presented.*

**Line 220: I assume that the model results are up-scaled and regridded on a 1x1 degree grid before the anomalies are calculated. Please clarify this here**

The model results are up-scaled to the observations'grid as hypothesized by the referee.

*We provided further details within the revised manuscript when necessary.*

**Line 231: I propose replacing "Paleo-Results" by Paleoclimate results.**

We agree.

*We modified the sentence accordingly to the referee's suggestion.*

**Line 237: Figure 7 shows the insolation changes over the mid-to-late Holocene. This is the main radiative forcing for the model experiments, so I suggest to show it already in Section 2 where the experimental design is discussed.**

We agree.

*We moved the mentioned picture to the second chapter accordingly to the editor's suggestion.*

**Line 250: what other cases?**

We realized that the previous sentence was misleading.

*We replaced it with "other regions".*

**Caption Figures 8 and 9: The captions are not consistent with the figures. Are summer results plotted at the upper or the lower row?**

As the referee noticed, in Figures 8 and 9 of the discussion paper the upper row represented winter while the lower summer. The captions, instead, were previously inverted.

*We changed the caption accordingly.*

**Figure 8: How is Figure 8 constructed? On what timeslice is it based, or is it based on results from several time slices?**

Figure 8 of the discussion paper represented the first two EOFs of winter and summer seasonal mean of mean sea level pressure, standardized to the preindustrial period. We now used a different analysis in the revised version of the manuscript, accordingly to the suggestions of the 1st referee.

*We replaced the maps of the EOFs of MSLP with the ones of a Canonical Correlation Analysis conducted on MSLP and T2M and presented in Fig10 and Fig.11. Nevertheless we added more details in the caption of these figures, being the previous ones not very precise.*

**Line 268: "scarce ability" Replace by poor ability?**

We agree.

*We modified "scarce ability" with "poor ability" following the referee's suggestion.*

**Line 276: "showing instead low correlation over the South". This is a confusing state- ment. Figure 10 shows that over most of the Mediterranean, the correlation in winter is strongly negative for the 1st EOF and strongly positive for summer.**

We realized that the previous period was not really clear. In fact, with the term SNAO we wanted to refer here to the Summer NAO.

The conclusions we were proposing, were definetely the same as the ones suggested by the referee.

*For this reason we better expressed this period in order make it more easily readable.*

**Line 284: "the model simulates a lower weight of the NAO ($\sim 40\%$) for mid-to-late Holocene in comparison to present-days conditions ($\sim 55\%$)". How can we reconcile this with the notion of a "more pronounced positive phase of the NAO during the mid- Holocene" as stated on line 277?**

We agree. Nevertheless, we want to highlight the fact that, according to different comments of both the authors, we deeply modified the previous analysis of atmospheric circulation.

*Based on the new analyses, we corrected the previous sentence on line 284.*

Reply to
**A. Strandberg**
*Mid-to-late Holocene Temperature Evolution and Atmospheric Dynamics over Europe in Regional Model Simulations by Russo, Emmanuele; Cubasch, Ulrich cp-2016-10*

1. **I would like to draw the authors attention to a study (Strandberg et al., 2014) that simulates 6k BP and 0.2k BP climate in Europe with a RCM. Although it only consists of two time slices I think it qualifies as high resolution simulations for different time slices of mid-to-late Holocene performed over Europe using a Regional Climate Model (perhaps the first such simulations). Furthermore, since Strandberg et al. (2014) use boundary data from ECHO-G and compare the results with the reconstructions from Mauri et al. (2014) it should be of interest for Russo and Cubash.**

Thanks for suggesting the work of Strandberg et al. 2014. It is interesting and gave us the opportunity to consider new proxy-reconstructions

for our discussion. Additionally, the paper structure, and in particular the paragraph on the comparison against other PMIP results, makes it a good reference to consider in order to further improve the first draft of our manuscript.

2. **I know that it is a characteristic of modellers to exaggerate the uncertainties in the models and downplay the uncertainties in the reconstructions, but I would be careful to validate the model against one set of reconstructions alone since they may be of equally good/poor quality as the model simulations. When considering astronomical forcing alone (see Fig. 2 in Wagner et al., 2007), we would expect 6k to be warmer than 0.2k and the temperature difference to be largest in summer in northern Europe. This is the signature we see in the model simulations of Strandberg et al. (2014). The non-pollen proxy based palaeoclimatic data presented in Strandberg et al. (2014) and the pollen based reconstruction of Peyron et al. (2013) rather support the differences in summer temperatures simulated by Strandberg et al. (2014) than the reconstruction of Mauri et al. (2014), in particular for southern and eastern Europe.**

The choice of the dataset of Mauri et al. 2014 has been done for many reasons. First of all it allows to perform a comparison with model results over most of the simulations domain, considering different variables (even if we only focus on temperature in our discussion). Then, it covers exactly the same time-slices of the model simulations. No other dataset has this temporal and spatial coverage. Additionally, the robustness of the data has been already tested, in Mauri et al. 2014, against other proxies (including chironomids, $\delta^1 8O$ from speleothems and lake ostracods, bog-oaks, glacio-lacustrine sediments, wood anatomy and other pollen reconstructions based on different reconstruction methods). For such reasons we think that the reconstructions of Mauri et al. 2014 are a reliable source for the comparison of model results. Nevertheless, considering other proxies for our analyses could be an important point. Preliminary qualitative analysis against other reconstructions, such as the ones of Hairi et al. 2014 and Peyron et al. 2013, confirm that the data used in our discussion present a similar behaviour. In our former analysis this was not evident since we considered regional means for the investigation of mid-to-late Holocene temperature evolution. For this reason, we now performed additional

analysis accordingly to this point.

[revised manuscript text omitted]

---

## Author Response (AR2)

Reply to
**Referees' Comments**
*Mid-to-late Holocene Temperature Evolution and Atmospheric Dynamics over Europe in Regional Model Simulations by Russo, Emmanuele; Cubasch, Ulrich cp-2016-10*

Dear editor,

Thank you very much, again, for your effort in reviewing our paper.

Below we go point by point through the technical corrections suggested by the two referees, detailing how we dealt with their concerns reported in **Bold**.

Thank you.

Reply to
**1st Reviewer**

- **Minor Comments**

  **As in the former version of the manuscript, there is an excess of too short paragraphs that could be merged to produce a more cohesive text. Examples of this are the first three paragraphs (that have not been merged despite my recommendation), lines 82-83, 143-145, 227-228, 258-259, 312-322, the four paragraphs between lines 347 and 366, and the two between 411 and 415, and finally even in the acknowledgements, which I'd merge into one single paragraph.**

  We agree.

  We merged together the mentioned paragraphs and tried to make the text more cohesive.

  **In section 2 I think it would be clearer if method and different dataset are described in separate subsections. Something like 2.1 Models, 2.2 Observations, 2.3 Reconstructions, or something similar.**

  We agree.

We structured the section accordingly to the referee comment.

**In the discussion of the model performance, in page 7, I think the discussion should be more quantitative, providing more numbers drwn from the corresponding figures that quantify the disagreements that are being discussed. In this regard, it would be interesting to provide also numbers drawn from similar exercises in the literature, so it can be quantitatively argued how the errors are indeed within the margins "acceptable".**

We agree.

Also based on the comment of the second refereee we made the discussion more quantitative, providing numbers quantifying the disagreements arising in the comparison between the different datasets.

**In Fig. 6, the anomalies are calculated over which period? Only one year, or are they averages over 10 years or longer?**

The anomalies are calculated over a 25-year period.

We added such information within the caption of the considered figure.

**In the discussion of Figure 6, it us argued that the warm bias in the North is present in the reconstructions, which happen in all models but in ECHO-G. However the figure is not shown. I think this is a quite important finding, and I'd advise the authors to reconsider showing the figure that is mentioned, because it's the only that can supports that rather important claim.**

We agree.

We added the figure relative to the pollen reconstructions to the previous plot.

**I think Tables 2 and 3 should be merged.**

We agree.

We merged together the mentioned tables accordingly to the referee comment.

**Although the text is clearer, I found several mistakes or expressions that I do not think are very standard. However, I didn't review this aspect of the manuscript extensively since as I understand this will be eventually tackled at a latter stage by professional text reviewers prior final publication, which**

**surely will amend these small mistakes and many other I could not possibly spot.**

We agree.

As also suggested by the second reviewer, we had the text proofread by an english native speaker and we modified it accordingly to his corrections.

Reply to
**2nd Reviewer**

- **General Comments**

**Compared to the previous version, this version has much improved. For instance, in my view, the comparison of the CCLM results with ECHAM5, including the application of the cost function, has strengthened the manuscript considerably. However, there are still several points that must be addressed before this manuscript can be considered for publication. Concerning my first main comment in the previous review, I find that the manuscript is still rather poorly written. It still includes many language mistakes, so I am quite sure that the text has not been checked by a native speaker as I suggested. Such a check would significantly improve the readability of the manuscript, so I stress this issue once more here.**

We agree with the second referee that the text was still requiring additional corrections and needed to be improved. This time the manuscript has been reviewed and corrected by an english native speaker, significantly improving its readability.

**Furthermore, regarding my 5th previous comment, the authors have not really provided a more in-depth discussion of the uncertainty of the summer temperature reconstructions for Southern Europe. In addition, their reply to my comment is not clear to me on this point. The authors note that "the previous analyses of mid-to-late Holocene temperature evolution were misleading. In fact, simply considering regional means, they did not allow to have a proper overview of the trends at different locations,**

**possibly resulting in a mismatch in the comparison against other proxies. The new maps presented in Fig. 4 show now a more heterogeneous behaviour, and are in better agreement with other indipendent reconstructions". From the reply, it is not clear who "they" are, or what "previous analyses" the authors are referring to. Furthermore, Figure 4 presents maps of summer temperatures for the period 1991-2000, so not for the mid-to-late Holocene.**

First of all we would like to clarify our previous response to the referee's 5th comment. With "the previous analyses of mid-to-late Holocene temperature evolution" we referred to the first analysis of the trends of temperature that we conducted. We think that the former analysis was misleading because it was obtained by simply averaging the data over two wide regions, not allowing to have a proper overview of the trends at different locations, possibly leading to a mismatch with other proxy-based reconstructions. The new maps (Fig. 9 of the Final Version of the manuscript) permit the comparison with indipendent reconstructions for different areas. In particular they present now a more heterogeneous behaviour, being in agreement this time with other reconstructions such as the one of Heiri et al. 2015 suggested by the referee, that showed that over the alps summer temperature was characterized by a decreasing trend during mid-to-late Holocene. Nevertheless, the new plots have been designed in order to carefully take into account the uncertainties of the reconstructions (as suggested by the referee) not only in summer but in both the seasons, for every single point of the domain (North and South), by means of a weighted least squares method. In addition, a detailed discussion on the pollen uncertainties and mismatches with some other reconstructiobns, as the ones based on chironomids, has been provided by Basil Davis in response to the second referee comment. However, despite all the efforts we made in order to carefully take into consideration the referee comment, we realized that our previous response was probably not accurate enough. Aware of this, considering the issue of reconstruction uncertainties a fundamental factor, we agree with the referee on the fact that a more in-depth discussion of the uncertainty of the summer temperature reconstructions for Southern Europe should be provided.

Based on these considerations we modfied the text accordingly, considering a series of different references.

**A major outcome of the present paper is the mismatch between the CCLM model results in Southern Europe and the pollen-based**

**reconstructions of Mauri et al. (2014), showing an opposite trends in summer. In Mauri et al. (2014), the pollen-based reconstruction is compared against independent reconstructions (their Figure 8), but this was not done for summer temperatures in Southern Europe, except for the Alps. In explaining the noted mismatch, Russo and Cubasch only consider a model bias in surface heat fluxes, but do not reflect on the possibility that the pollen-based reconstruction does not capture the right temperature trend in the Mediterranean area during the Holocene. The latter possibility has been discussed in the literature before as I mentioned in my review of the CPD manuscript, and I urge the authors again to make the readers aware of this discussion. As mentioned in Section 3.3.1., all PMIP3 models suggest warmer conditions in Southern Europe during the mid-Holocene (see also Table S2), so if the reconstruction of Mauri et al. (2014) is correctly indicating cooler conditions here, this implies that all these models are not capable of capturing the right climate response in Southern Europe under influence of astronomical forcing. An alternative to consider is that the reconstructions are incorrect. This important issue is still under debate, and in my opinion it is crucial to provide the arguments of both sides of this debate.**

We agree with the referee that our discussion should be possibly expanded, taking into account his suggestions and trying to make the reader more aware of the possibility that the reconstructions could be incorrect, providing the arguments of both sides of the debate.

Bearing this in mind, taking into account the bibliography suggested by the referee and considering additional one, we developed the text accordingly.

Below we report in *italic* the text included into the discussion and developed according to the previous two points:

*It is important to mention that the behaviour of mid-to-late Holocene's summer temperature over Europe has been highly debated during recent years. While a dipole behaviour has been suggested by several studies based on pollen analyses ([1, 9, 2, ?, 10]) and others relying on a combination of different proxies, such as the one of Magny et al. 2013 [3], which suggested a North-South paleohydrological contrast in the central Mediterranean during the Holocene, other studies argued against such hypothesis. In particular Osborne et al. 2000 [4] proposed that reconstructions of summer tempera-*

*ture based on pollen could be erroneous for the Mediterranean region, since here the vegetation distribution is mainly limited by effective precipitation, rather than by summer temperature. The latest hypothesis should be taken into account for the comparison between pollen-based reconstructions and model simulations. Nevertheless, additional investigations have shown that, when directly compared to the pollen record, the mid-Holocene vegetation simulated from the output of climate models is way too dry over Southern Europe, with an expansion of Mediterranean and steppe/desert vegetation and contraction in forest cover, a direct consequence of simulated warmer conditions ([2, 5, 6, 7, 8]). Based on these considerations, recognizing the dataset of Mauri et al. 2015 as a valuable source for the investigation of European temperature evolution during mid-to-late Holocene, we acknowledge the fact that joint efforts from specialists of different disciplines are still required in order to further clarify possible uncertainties.*

- Minor Comments:

  **L157: It is mentioned here that effects of greenhouse gas forcing is not considered in the analyses. Does this mean that the trace gas concentrations were kept fixed at their preindustrial values in all experiments? Or are the concentrations adjusted in the experiments based on ice-core measurements, but is the effect of this forcing not separated from the impact of insolation changes? Please clarify.**

  In our analysis of temperature evolution during mid-to-late Holocene, since the impact on the radiative balance of changes in the concentration of greenhouse gases was considerably small compared to the effects of changes in insolation, in our discussion we did not separate them from the latter ones. Nonetheless, the values of GHGs were set in the experiment based on ice-cores measurements.

  **L195: Please explain what you mean by 'satisfactory results'.**

  With "satisfactory results" we referred to the fact that the data of Mauri et al. 2015 proved to be in good agreement with indipendent reconstructions derived from different proxies. The details of their analysis have already been enunciated in Mauri et al. 2015.

  **L213 and L219: Why do you use "instead" in these sentences?**

  We agree with the referee that "instead" in this context is not necessary.

We decided to take it out from the text.

**L**219 to L228: I suggest to make this section more quantitative, as it is very descriptive.

We agree.

We modified this part of the text accordingly.

**S**ection 3.3.1: This new section on the results of PMIP3 experiments is a useful addition to the manuscript. However, it is very descriptive and it could be made more informative by making it more quantitative.

We agree.

We modified this subsection in order to make it more quantitative, accordingly to the referee suggestion.

**L270: I suggest to use "proxy-based reconstructions", as the proxies themselves are not reconstructed.**

We agree.

Corrected in proxy-based reconstructions.

**L281: "The coastline is also better reproduced in this case, resulting in more suitable informations for possible comparison against proxy-data". Please clarify how a more detailed coastline would improve the model-data comparison. See also line 448 in the Conclusions.**

We modified the text accordingly to the referee's comment, stressing out the fact that a better reproduced coastline allows for a more robust and precise representation of the surface and its physical processes. This would consequently lead to more suitable and robust information for the comparison against proxy-data.

**L283: "CCLM shows better defined patterns as a consequence of higher resolution". What kind of patterns, please clarify. See also line 449 in the conclusions.**

Although we took into consideration the referee's comment we preferred not to change the former sentence since we think it was already clear. With "CCLM shows better defined patterns as a consequence of higher resolution" we mainly referred to the fact that the spatial patterns of temperature deriving from the CCLM simulations are better defined in comparison to the ECHAM5's ones, since the RCM's spatial resolution allows to better discriminate higher spatial variability. Hence this could constitute added value.

**L287: Zhang et al. (2010) is missing in the reference list.**

We checked the previous version of the text and the mentioned reference was already present.

**L306: "It is important to mention that the scale considered in our analysis is closer to the resolution of ECHAM5 than the one of the CCLM". What scale are you referring to here? The scale of the reconstructions?**

In the previous sentence we referred to the resolution of the reconstructions. We tried to improve the text in order to make it clearer.

**L347: In Fig 9 the winter map for CCLM is gray (i.e. the trend is not significant) in Scandinavia, where the reconstructions show a strong, significant cooling trend. So here the model results do not match the proxy data, and I suggest to mention this here.**

We agree.

We modified the text accordingly to the referee's comment.

**L387: "the first CCA pair (Fig. 11 a,b)". Is this correct? The MSLP pattern explaining most of the variance is shown in Fig. 11c, so should Fig 11c,d not be the first CCA pair?**

Yes. It is correct. In CCA the order of the canonical pairs follows the values of correlation. The first pair of spatial patterns is the one with the highest correlation, followed by the second one and so on in a decreasing order.

**L409: "to capture this trend". What trend? In soil moisture during Summer? Or in winter or spring precipitation. Please clarify.**

We refer to the trend in temperature over Southern Europe.

We tried to make it clearer within the text.

**Figure1: In the caption, a reference for the presented data should be added.**

We agree.

We modified the caption accordingly.

**Caption Figure 5: Typo, "refernce period".**

Corrected.

**Figure 6: I suggest to use the same projection in all six panels to facilitate the inter-model comparison.**

Even if we agree with the referee on the fact that the use of the same projection would facilitate the inter-model comparison, we think that the best way to achieve this task would be by means of interpolation. We think that this could alter the results leading us away from the original goal of the analysis. For this reason we preferred to leave the figure as it is. However we tried to be more accurate in the caption of the figure, adding further details.

**Figure 9: Please clarify what the unit of the slopes. Is it in C per 6kyr? Or C per kyr?**

We agree. The unit is $^{o}$C/kyr. We modified the figure accordingly.

**Figure 12: In my view, this figure does not add much to the manuscript, so it could be described only, or it could be moved to the supplementary information.**

We do not agree with the referee and we do think that the figure should be included in the manuscript, as it gives important informations of the temporal expansion of the two variables into consideration and allows a substantial contribution to the developed discussion and conclusions.

Below we propose some additional bibliography that we will provide, in the revised version of the manuscript, accordingly to the referee's comments. Also some references already included in the manuscript and mentioned in these comments are reported.

[revised manuscript text omitted]

---

## Author Response (AR3)

Reply to
**Editor's Comments**
*Mid-to-late Holocene Temperature Evolution and Atmospheric Dynamics over Europe in Regional Model Simulations by Russo, Emmanuele; Cubasch, Ulrich cp-2016-10*

Dear editor,

Thank you very much, again, for your effort in reviewing our paper.

Below we go point by point through your suggested technical corrections, detailing how we dealt with your concerns reported in **Bold**. All the line numbers correspond to the versions with track changes appended to the response to reviewers comments.

Thank you.

**Abstract. Line 20. You have modified the text to mention the uncertainties in the reconstructions but the abstract has not been changed (nor the conclusions).**

We agree.

We modified both the abstract and the conclusions in order to include more details on the uncertainties of the reconstructions. We tried to be concise in the abstract, providing additional considerations in the conclusion.

**Abstract. Line 22 .same as what ?**

Corrected.

We modified the sentence referring to the reconstructions.

**Line 43 case study instead of case of study (see also line 128)**

We agree.

Corrected.

**Line 56. Their signal refer to reconstructions?**

Yes.

We tried to reformulate the sentence more clearly.

**Line 159, Fig ?? (same problem later)**

We realized that the same error was present throughout the entire manuscript, and it is due to the version of latexdiff we used for the track-changes version of the manuscript.

In the final revised version such errors are not present.

**Line 161 60 and 30 latitudes North $60^o$ and $30^o$ North (same caption of Fig. 1).**

Corrected.

**Line 161 60 and 30 latitudes North 60 and 30 North (same caption of Fig. 1).**

We agree.

**Line 246. mentioned area  refer to what ?**

It refers to Southern Europe and the Mediterranean region.

We corrected the sentence trying to be more specific.

**Line 292. Is data the appropriate term ?**

We agree with the editor that in this case "data" is not the most appropriate term.

We corrected it with "simulations".

**Line 294.  Is bias' the appropriate term?  It usually refers to an error not a just a change?**

We agree.

Corrected.

**Line 306. step or steps ?**

We think that step is the right word in this case, since this is the second (and final) step of our analysis of the added value of the RCM.

**Line 307-308. I would suppress Under the mentioned considerations**

We agree.

Corrected. **Line 345 Would bias be better than anomalies (as used in line 350)?**

We agree.

Corrected.

**Line 355. South mean Southern Europe?**

Yes.

Corrected it in order to make it clearer.

**Line 371 their instead of thei**

Corrected.

**Line 424 explain instead of justify?**

We agree.

Corrected.

**Line 436 scarce ability does not seem to be the appropriate wording to me.**

We agree.

We modified the text accordingly to the editor's suggestion.

**Line 438 I would join the two paragraphs.**

Again it seems that it was an error related to the latexdiff version we employed for the track-changes version of the manuscript.

In the final version such paragraphs are merged together.

**Line 442 Specify a dipole between which regions.**

Between Northern and Southern Europe.

We corrected the text accordingly to the editor's suggestion.

**Line 446. Why paleohydrological while the paper focuses on temperatures ?**

Magny et al. 2013 found a contrasting behaviour between Northern and Southern Mediterranean regions for temperature and precipitation.

We corrected the previous sentence employing the term "paleoclimate".

**Line 471. Again, I would not use bias here, rather anomaly (same line 480).**

We agree.

Corrected.

**Line 478-481. This paragraph corresponds to which season? Should it be merged with the previous paragraph?**

It corresponds to summer season.

We merged this paragraph with the previous one.

With kind regards on behalf of the all authors,
Emmanuele Russo

[revised manuscript text omitted]